# TRAINING-FREE MESSAGE PASSING FOR LEARNING ON HYPERGRAPHS

**Bohan Tang**[1]   **Zexi Liu**[2*]   **Keyue Jiang**[3*]   **Siheng Chen**[2,4]   **Xiaowen Dong**[1]
[1]University of Oxford   [2]Shanghai Jiao Tong University   [3]University College London
[4]Shanghai AI Laboratory
`bohan.tang@eng.ox.ac.uk`

## ABSTRACT

Hypergraphs are crucial for modelling higher-order interactions in real-world data. Hypergraph neural networks (HNNs) effectively utilise these structures by message passing to generate informative node features for various downstream tasks like node classification. However, the message passing module in existing HNNs typically requires a computationally intensive training process, which limits their practical use. To tackle this challenge, we propose an alternative approach by decoupling the usage of hypergraph structural information from the model training stage. This leads to a novel training-free message passing module, named TF-MP-Module, which can be precomputed in the data preprocessing stage, thereby reducing the computational burden. We refer to the hypergraph neural network equipped with our TF-MP-Module as TF-HNN. We theoretically support the efficiency and effectiveness of TF-HNN by showing that: 1) It is more training-efficient compared to existing HNNs; 2) It utilises as much information as existing HNNs for node feature generation; and 3) It is robust against the oversmoothing issue while using long-range interactions. Experiments based on seven real-world hypergraph benchmarks in node classification and hyperlink prediction show that, compared to state-of-the-art HNNs, TF-HNN exhibits both competitive performance and superior training efficiency. Specifically, on the large-scale benchmark, Trivago, TF-HNN outperforms the node classification accuracy of the best baseline by $10\%$ with just $1\%$ of the training time of that baseline.

## 1 INTRODUCTION

Higher-order interactions involving more than two entities exist in various domains, such as co-authorships in social science (Han et al., 2009). Hypergraphs extend graphs by allowing hyperedges to connect more than two nodes, making them suited to capture these complex relationships (Bick et al., 2023; Jhun, 2021; Tang et al., 2023b;a). To utilise such structures for downstream tasks, learning algorithms on hypergraphs have garnered increasing attention.

Inspired by the success of graph neural networks (GNNs) (Wu et al., 2020), current research focuses on developing hypergraph neural networks (HNNs) with a message passing module (MP-Module) that can be compatible with various task-specific modules. The MP-Module enables information exchange between connected nodes to generate informative node features for specific tasks (Feng et al., 2019; Wang et al., 2023b; Telyatnikov et al., 2023). However, similar to other message passing neural networks (Frasca et al., 2020; Wu et al., 2022), training the MP-Module makes loss computation interdependent for connected nodes, resulting in a computationally intensive training process for HNNs. This limits their practical applications, especially in the process of large-scale hypergraphs.

To address this challenge, our key solution is to develop a training-free MP-Module that shifts the processing of hypergraph structural information from the training stage to the data pre-processing phase. This approach is inspired by recent advancements in efficient GNNs (Wu et al., 2019a; Gasteiger et al., 2019; Frasca et al., 2020), where training-free MP-Modules are typically implemented as graph filters (Ortega et al., 2018). However, directly using existing hypergraph filters (Zhang et al.,

---

*Equal contribution.

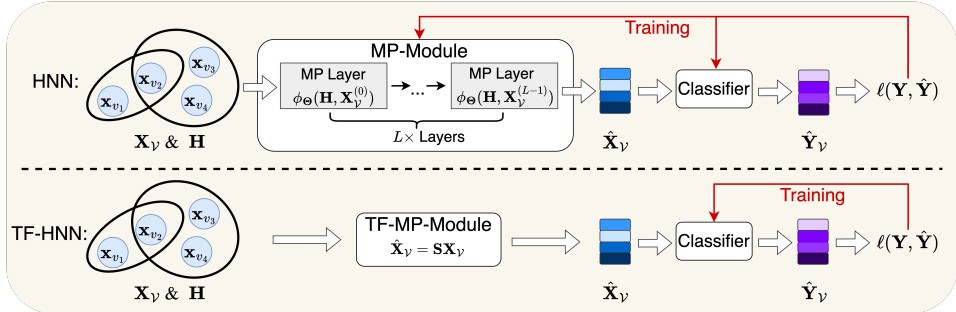

Figure 1: Training pipeline of HNN vs. TF-HNN for node classification. *Top row*: HNN uses a hypergraph structure to generate node features by a learnable MP-Module, which are then used by a classifier, with the MP-Module and the classifier being trained together. For brevity, we omit the MLP in HNN for the input node features. *Bottom row*: TF-HNN comprises only the classifier trained for node classification and the TF-MP-Module that can be recomputed before the classifier training.

2019; Qu et al., 2022) presents two major obstacles. Firstly, these filters are usually designed for $k$-uniform hypergraphs, where all hyperedges must have the same size $k$, limiting their applicability to more general hypergraph structures with varying hyperedge sizes. Secondly, the reliance on an incidence tensor to represent the hypergraph presents significant challenges in practical usage. As the tensor dimension grows exponentially with $n^k$, existing computational resources struggle to perform multiplications involving such high-dimensional tensors. Therefore, instead of adopting existing hypergraph filters, we introduce a novel training-free MP-Module for hypergraphs.

In this work, we develop a novel, training-free message passing module, TF-MP-Module, for hypergraphs to decouple the usage of hypergraph structure information from the model training stage. To achieve this, we first construct a theoretical framework that provides a unified view of existing HNNs (Huang & Yang, 2021; Chen et al., 2022; Chien et al., 2022; Wang et al., 2023a). Specifically, this framework identifies the feature aggregation function as the core operator of MP-Modules in existing HNNs. Based on these insights, we design the TF-MP-Module in two stages: 1) We make the feature aggregation functions within the MP-Modules of four state-of-the-art HNNs (Huang & Yang, 2021; Chen et al., 2022; Chien et al., 2022; Wang et al., 2023a) training-free by removing their learnable parameters; and 2) To further enhance efficiency, we remove the non-linear activation functions and consolidate the feature aggregation across $L$ layers into a single propagation step. Remarkably, this two-stage process unifies the chosen MP-Modules into a single formulation, despite their different design philosophies. We refer to this unified formulation as the TF-MP-Module and denote the hypergraph neural network equipped with it as TF-HNN. To demonstrate the efficiency and effectiveness of TF-HNN, we provide a theoretical analysis showing three key advantages: 1) TF-HNN is more training-efficient compared to existing HNNs; 2) TF-HNN can utilise as much information as existing HNNs for generating node features; and 3) TF-HNN is robust against the oversmoothness issue while taking into account the long-range information.

The main contributions of this work are summarised as follows:

• We present an original theoretical framework that identifies the feature aggregation function as the core component of MP-Modules in existing HNNs, which processes the hypergraph structural information. This insight provides a deeper understanding of existing HNNs.

• We develop TF-HNN, an efficient and effective model for hypergraph-structured data. To our knowledge, TF-HNN is the first model that decouples the processing of hypergraph structural information from model training, significantly enhancing training efficiency.

• We theoretically support the efficiency and effectiveness of TF-HNN by showing that: 1) It leads to remarkably low training complexity when solving hypergraph-related downstream tasks; 2) It utilises the same amount of information as existing HNNs for generating node features; and 3) It is robust against the oversmoothing issue when utilising long-range information.

• We conduct extensive experiments in both node classification and hyperedge prediciton tasks to compare TF-HNN with nine state-of-the-art HNNs. The empirical results show that the proposed TF-HNN exhibits both competitive learning performance and superior training efficiency.

## 2 NOTATION

**Hypergraph.** Let $\mathcal{H} = \{\mathcal{V}, \mathcal{E}, \mathbf{H}\}$ be a hypergraph, where $\mathcal{V} = \{v_1, v_2, \cdots, v_n\}$ is the node set, $\mathcal{E} = \{e_1, e_2, \cdots, e_m\}$ is the hyperedge set, and $\mathbf{H} \in \{0, 1\}^{n \times m}$ is an incidence matrix in which $\mathbf{H}_{ik} = 1$ means that $e_k$ contains node $v_i$ and $\mathbf{H}_{ik} = 0$ otherwise. Define $\mathbf{D}_{\mathcal{H}\mathcal{V}} \in \mathbb{R}_{\geq 0}^{n \times n}$ as a diagonal matrix of node degrees and $\mathbf{D}_{\mathcal{H}\mathcal{E}} \in \mathbb{R}_{\geq 0}^{m \times m}$ as a diagonal matrix of hyperedge degrees, where $\mathbf{D}_{\mathcal{H}_{ii}^{\mathcal{V}}}$ and $\mathbf{D}_{\mathcal{H}_{kk}^{\mathcal{E}}}$ are the number of hyperedges with $v_i$ and the number of nodes in $e_k$, respectively. We assume hypergraphs do not have isolated nodes or empty hyperedges; details of this assumption are in Appendix Q. Moreover, we define: 1) The distance between two nodes on a hypergraph is the number of hyperedges in the shortest path between them, e.g., the distance is one if they are directly connected by a hyperedge; and 2) The $k$-hop neighbours of a node $v_i$ are all nodes with a distance of $k$ or less.

**Graph and clique expansion.** Let $\mathcal{G} = \{\mathcal{V}, \mathbf{W}\}$ be a graph, where $\mathcal{V} = \{v_1, v_2, \cdots, v_n\}$ is the node set, and $\mathbf{W} \in \mathbb{R}_{\geq 0}^{n \times n}$ is the adjacency matrix of $\mathcal{G}$ in which $\mathbf{W}_{ij} > 0$ means that $v_i$ and $v_j$ are connected and $\mathbf{W}_{ij} = 0$ otherwise. We set $\mathbf{D} \in \mathbb{R}_{\geq 0}^{n \times n}$ as a diagonal node degree matrix for $\mathcal{G}$, where $\mathbf{D}_{ii}$ is the sum of the $i$-th row of $\mathbf{W}$. Moreover, we denote the graph Laplacian of $\mathcal{G}$ as $\mathbf{L} = \mathbf{D} - \mathbf{W}$. Given a hypergraph $\mathcal{H} = \{\mathcal{V}, \mathcal{E}, \mathbf{H}\}$, its clique expansion is defined as a graph $\mathcal{G} = \{\mathcal{V}, \mathbf{W}\}$, where $\mathcal{V}$ remains unchanged, and $\mathbf{W}_{ij} > 0$ if and only if $v_i$ and $v_j$ are connected on $\mathcal{H}$ and $\mathbf{W}_{ij} = 0$ otherwise.

**Other notations.** We set node features as $\mathbf{X}_{\mathcal{V}} = [\mathbf{x}_{v_1}^{\top}, \mathbf{x}_{v_2}^{\top}, \cdots, \mathbf{x}_{v_n}^{\top}]^{\top} \in \mathbb{R}^{n \times d}$, which is a matrix that contains $d$-dimensional features. We denote functions or variables at the $l$-th layer of a model using the superscript $(l)$ and use $\oplus$ for concatenation. We use $\mathbf{\Theta}$ to represent the learnable weight matrix in a model, $\mathrm{MLP}(\cdot)$ for a multilayer perceptron (MLP), and $\sigma(\cdot)$ for non-linear functions (e.g., ReLU).

## 3 METHODOLOGY

In this section, we first show an overview of the proposed TF-HNN approach (Subsec. 3.1). Moreover, we present a theoretical framework that offers a unified view on existing HNNs (Subsec. 3.2). Finally, we use the insights gained from the theoretical framework to design our TF-MP-Module (Subsec. 3.3).

### 3.1 OVERVIEW

To enhance the GNN efficiency, Gasteiger et al. (2019) design APPNP by using the connection between GCN (Kipf & Welling, 2017) and PageRank (Page et al., 1998), shifting learnable parameters to an MLP prior to the message passing. Wu et al. (2019a) and Frasca et al. (2020) further improve training efficiency by removing the MLP before the message passing, completing message passing in preprocessing and making it training-free. Inspired by them, we design the TF-HNN that removes the reliance on message passing at training. Generally, TF-HNN is described as the following framework:

$$\hat{\mathbf{Y}} = \varphi_{\mathbf{\Theta}}(\hat{\mathbf{X}}_{\mathcal{V}}), \quad \hat{\mathbf{X}}_{\mathcal{V}} = S(\mathbf{X}_{\mathcal{V}}, \mathbf{H}), \tag{1}$$

where $\varphi_{\mathbf{\Theta}}(\cdot)$ is a task-specific module, $\hat{\mathbf{Y}}$ denotes the task-specific output whose dimension is task-dependent[1], $S(\cdot)$ denotes an MP-Module that contains only pre-defined parameters allowing it to be precomputed in the pre-processing phase, and $\hat{\mathbf{X}}_{\mathcal{V}} \in \mathbb{R}^{n \times d}$ denote the features generated by $S(\cdot)$. In the GNN literature, $S(\cdot)$ is implemented as the graph filter (Ortega et al., 2018). However, we cannot directly use existing hypergraph filters (Zhang et al., 2019; Qu et al., 2022) for two main reasons. Firstly, existing hypergraph filters are usually designed for $k$-uniform hypergraphs, where all hyperedges have size $k$. Secondly, these filters require the use of an incidence tensor, which presents challenges in handling high-dimensional matrix multiplication. Hence, rather than relying on existing methods, we design a novel function $S(\cdot)$ for hypergraphs, the TF-MP-Module. Since our design is based on existing MP-Modules for hypergraphs, the next section introduces an original framework that offers a unified perspective on current HNNs.

### 3.2 REVISITING HYPERGRAPH NEURAL NETWORKS

A traditional HNN for a certain downstream task is formulated as the following equation:

$$\hat{\mathbf{Y}} = \varphi_{\mathbf{\Theta}}(\hat{\mathbf{X}}_{\mathcal{V}}), \quad \hat{\mathbf{X}}_{\mathcal{V}} = \Phi_{\mathbf{\Theta}}(\mathbf{X}_{\mathcal{V}}, \mathbf{H}), \tag{2}$$

---

[1]For instance, in the node classification task, $\hat{\mathbf{Y}} \in \mathbb{R}^{n \times c}$ contains the logits for $c$ categories; and in the hyperlink prediction task, $\hat{\mathbf{Y}} \in \mathbb{R}^{m_p}$ contains the probability of $m_p$ potential hyperedges.

Table 1: Overview for four state-of-the-art HNNs and our TF-HNN. In this table, $\gamma_U, \gamma_E, \gamma_D, \gamma'_l \in (0,1)$ are hyperparameters and $\mathbf{I} \in \mathbb{R}^{d \times d}$ denotes an identity matrix. Moreover, $M$ is the training computational complexity of the task-specific module, $n$ is the node count, $m$ is the hyperedge count, $m'$ is the number of edges in the clique expansion, $\|\mathbf{H}\|_0$ is the number of non-zero values in $\mathbf{H}$, $T$ is the number of training epochs, $L$ is the number of layers, and $d$ is the feature dimension.

| Name | Type | Hypergraph-Wise Feature Aggregation Function | Training Computational Complexity |
|---|---|---|---|
| UniGCNII (Huang & Yang, 2021) | Direct | $\mathbf{X}_{\mathcal{V}}^{(l)} = \sigma\Big( \big( (1-\gamma_U)\mathbf{D}_{\mathcal{H}\mathcal{V}}^{-1/2}\mathbf{H}\tilde{\mathbf{D}}_{\mathcal{H}\varepsilon}^{-1/2}\mathbf{D}_{\mathcal{H}\varepsilon}^{-1}\mathbf{H}^{\top}\mathbf{X}_{\mathcal{V}}^{(l-1)} + \gamma_U\mathbf{X}_{\mathcal{V}}^{(0)} \big)\boldsymbol{\Theta}^{(l)} \Big)$ | $\mathcal{O}(M + TL(n + m + \|\mathbf{H}\|_0)d + TLnd^2)$ |
| Deep-HGNN (Chen et al., 2022) | Direct | $\mathbf{X}_{\mathcal{V}}^{(l)} = \sigma\Big( \big( (1-\gamma_D)\mathbf{D}_{\mathcal{H}\mathcal{V}}^{-1/2}\mathbf{H}\mathbf{D}_{\mathcal{H}\varepsilon}^{-1}\mathbf{H}^{\top}\mathbf{D}_{\mathcal{H}\mathcal{V}}^{-1}\mathbf{X}_{\mathcal{V}}^{(l-1)} + \gamma_D\mathbf{X}_{\mathcal{V}}^{(0)} \big)\big( (1-\gamma'_D)\mathbf{I} + \gamma'_D\boldsymbol{\Theta}^{(l-1)} \big) \Big)$ | $\mathcal{O}(M + TLm'd + TLnd^2)$ |
| AllDeepSets (Chien et al., 2022) | Indirect | $\mathbf{X}_{\mathcal{V}}^{(l)} = \mathrm{MLP}\Big( \mathbf{D}_{\mathcal{H}\mathcal{V}}^{-1}\mathbf{H}\,\mathrm{MLP}\big( \mathbf{D}_{\mathcal{H}\varepsilon}^{-1}\mathbf{H}^{\top}\mathrm{MLP}(\mathbf{X}_{\mathcal{V}}^{(l-1)}) \big) \Big)$ | $\mathcal{O}(M + TL\|\mathbf{H}\|_0 d + TL(n+m)d^2)$ |
| ED-HNN (Wang et al., 2023a) | Indirect | $\mathbf{X}_{\mathcal{V}}^{(l)} = \mathrm{MLP}\left[ (1-\gamma_E)\mathrm{MLP}\Big( \mathbf{D}_{\mathcal{H}\mathcal{V}}^{-1}\mathbf{H}\,\mathrm{MLP}\big( \mathbf{D}_{\mathcal{H}\varepsilon}^{-1}\mathbf{H}^{\top}\mathrm{MLP}(\mathbf{X}_{\mathcal{V}}^{(l-1)}) \big) \Big) + \gamma_E\mathbf{X}_{\mathcal{V}}^{(0)} \right]$ | $\mathcal{O}(M + TL\|\mathbf{H}\|_0 d + TL(n+m)d^2)$ |
| TF-HNN (Ours) | Direct | $\hat{\mathbf{X}}_{\mathcal{V}} = \mathbf{S}\mathbf{X}_{\mathcal{V}}$ | $\mathcal{O}(M)$ |

where $\varphi_{\boldsymbol{\Theta}}(\cdot)$ is a task-specific module, $\hat{\mathbf{Y}}$ denotes the task-specific output whose dimension is task-dependent, $\Phi_{\boldsymbol{\Theta}}(\cdot)$ is a learnable message passing module, and $\hat{\mathbf{X}}_{\mathcal{V}} \in \mathbb{R}^{n \times d}$ denote the features generated by $\Phi_{\boldsymbol{\Theta}}(\cdot)$. The modules $\varphi_{\boldsymbol{\Theta}}(\cdot)$ and $\Phi_{\boldsymbol{\Theta}}(\cdot)$ are trained together with the supervision given by the downstream tasks. See Figure 1 for the comparison between the training pipelines of traditional HNNs and TF-HNN in the context of node classification. Existing HNNs use the hypergraph structural information by the message passing module $\Phi_{\boldsymbol{\Theta}}(\cdot)$ (MP-Module). The MP-Module in existing HNNs is typically based on one of four mechanisms[2]: clique-expansion, star-expansion, line-expansion, or incidence-tensor (Huang & Yang, 2021; Wang et al., 2023b; Chen et al., 2022; Chien et al., 2022; Wang et al., 2023a; Yang et al., 2022; Antelmi et al., 2023; Kim et al., 2024). We summarize the MP-Module using these mechanisms in Proposition 3.1, with the proof provided in Appendix C.

**Proposition 3.1.** *Let $\mathbf{x}_{v_i}^{(l)}$ be features of node $i$ in the $l$-th message passing layer of a HNN based on clique-expansion/star-expansion/line-expansion/incidence-tensor, $\phi_{\boldsymbol{\Theta}}(\cdot)$ denotes a learnable node-wise feature aggregation function, and $\mathcal{N}_{\mathcal{H}_{v_i}}$ is the set of neighbours of $v_i$ on the hypergraph. For $i = 0$, we have $\mathbf{x}_{v_i}^{(0)} = \mathrm{MLP}(\mathbf{x}_{v_i})$. For $i \in \mathbb{Z}^+$, we have $\mathbf{x}_{v_i}^{(l)} = \phi_{\boldsymbol{\Theta}}\big( \mathbf{x}_{v_i}^{(0)}, \mathbf{x}_{v_i}^{(l-1)}, \oplus_{v_j \in \mathcal{N}_{\mathcal{H}_{v_i}}} \mathbf{x}_{v_j}^{(l-1)} \big)$.*

Specifically, $\phi_{\boldsymbol{\Theta}}(\cdot)$ defined in Proposition 3.1 can be categorized into two types: direct and indirect feature aggregation functions. The direct approaches aggregate features of neighbouring nodes to that of the target node directly (Huang & Yang, 2021; Wang et al., 2023b), while the indirect methods aggregate features of neighbouring nodes to that of the target node via some virtual nodes (Chien et al., 2022; Wang et al., 2023a; Yang et al., 2022). These approaches can be formulated as:

$$\mathbf{x}_{v_i}^{(l)} = f_3\Big( f_0(\mathbf{x}_{v_i}^{(0)}) + f_1(\mathbf{x}_{v_i}^{(l-1)}) + \sum_{v_j \in \mathcal{N}_{\mathcal{H}_{v_i}}} f_2(\mathbf{x}_{v_j}^{(l-1)}) \Big), \tag{3a}$$

$$\mathbf{x}_{v_i}^{(l)} = g_2\Big( g_0(\mathbf{x}_{v_i}^{(0)}) + \mathbf{p}^{\top} g_1(\mathbf{x}_{v_i}^{(l-1)}, \oplus_{v_j \in \mathcal{N}_{\mathcal{H}_{v_i}}} \mathbf{x}_{v_j}^{(l-1)}) \Big), \tag{3b}$$

where $f_0(\cdot)$, $f_1(\cdot)$, $f_2(\cdot)$, $f_3(\cdot)$, $g_0(\cdot)$, $g_1(\cdot)$, $g_2(\cdot)$ represent seven learnable functions, and $\mathbf{p} \in \mathbb{R}^{n_v}$ is a vector used to aggregate the features of virtual nodes generated by $g_1(\cdot)$. Here, $n_v$ is a hyperparameter corresponding to the number of the virtual nodes, and Eq. (3a) and Eq. (3b) represent the direct and indirect approaches, respectively. Based on Propostion 3.1, the hypergraph structure is only used to construct the neighbourhood set $\mathcal{N}_{v_i}$ within the MP-Module of HNNs. Consequently, the feature aggregation function $\phi_{\boldsymbol{\Theta}}(\cdot)$ is the core component of the MP-Module in processing hypergraph structural information. Moreover, based on Eq. (3a) and Eq. (3b), the usage of the hypergraph structural information is not directly related to the learnable parameters and the non-linear function. Therefore, in the next subsection, we design our TF-MP-Module by first removing the learnable parameters for the feature aggregation function from existing HNNs to make it training-free. To further improve its efficiency, we eliminate the non-linear function from the feature aggregation process. Specifically, we design our model based on HNNs shown in Table 1.

### 3.3 TRAINING-FREE MESSAGE PASSING

We introduce the TF-MP-Module by removing the learnable parameters and the non-linear function from four state-of-the-art HNNs shown in Table 1, which includes two methods with the direct feature aggregation function and two methods with the indirect feature aggregation function.

---

[2]Details on these mechanisms can be found in Appendix B

**Learnable parameters removal.** Based on the insights from Subsec. 3.2, to develop a training-free MP-Module, we first remove learnable parameters from the feature aggregation functions of the selected HNNs. Specifically, we replace the learnable matrices in these functions with identity matrices. This removal allows us to reformulate the feature aggregation functions as follows:

$$\mathbf{X}_{\mathcal{V}}^{(l)} = \sigma\Big((1-\gamma_U)\mathbf{D}_{\mathcal{H}\mathcal{V}}^{-1/2}\mathbf{H}\tilde{\mathbf{D}}_{\mathcal{H}\mathcal{E}}^{-1/2}\mathbf{D}_{\mathcal{H}\mathcal{E}}^{-1}\mathbf{H}^{\top}\mathbf{X}_{\mathcal{V}}^{(l-1)} + \gamma_U\mathbf{X}_{\mathcal{V}}^{(0)}\Big), \tag{4a}$$

$$\mathbf{X}_{\mathcal{V}}^{(l)} = \sigma\Big((1-\gamma_D)\mathbf{D}_{\mathcal{H}\mathcal{V}}^{-1/2}\mathbf{H}\mathbf{D}_{\mathcal{H}\mathcal{E}}^{-1}\mathbf{H}^{\top}\mathbf{D}_{\mathcal{H}\mathcal{V}}^{-1/2}\mathbf{X}_{\mathcal{V}}^{(l-1)} + \gamma_D\mathbf{X}_{\mathcal{V}}^{(0)}\Big), \tag{4b}$$

$$\mathbf{X}_{\mathcal{V}}^{(l)} = \sigma\left(\mathbf{D}_{\mathcal{H}\mathcal{V}}^{-1}\mathbf{H}\sigma\Big(\mathbf{D}_{\mathcal{H}\mathcal{E}}^{-1}\mathbf{H}^{\top}\sigma(\mathbf{X}_{\mathcal{V}}^{(l-1)})\Big)\right), \tag{4c}$$

$$\mathbf{X}_{\mathcal{V}}^{(l)} = \sigma\left[(1-\gamma_E)\sigma\left(\mathbf{D}_{\mathcal{H}\mathcal{V}}^{-1}\mathbf{H}\sigma\Big(\mathbf{D}_{\mathcal{H}\mathcal{E}}^{-1}\mathbf{H}^{\top}\sigma(\mathbf{X}_{\mathcal{V}}^{(l-1)})\Big)\right) + \gamma_E\mathbf{X}_{\mathcal{V}}^{(0)}\right], \tag{4d}$$

where Eq. (4a), Eq. (4b), Eq. (4c) and Eq. (4d) represent the simplified formulation of UniGCNII, Deep-HGNN, AllDeepSets and ED-HNN, respectively. Notably, $\sigma(\cdot)$ only denotes a general non-linear function without parameters, not implying that these formulas use the same function.

**Linearisation.** Inspired by the insights from Subsec. 3.2, we further remove the non-linearity from the functions represented by Eqs. (4a~4d), thereby transforming an MP-Module with $L$ feature aggregation functions into a more efficient single propagation step. For the sake of brevity, we present the simplification results as Proposition 3.2 and provide its mathematical proof in Appendix D.

**Proposition 3.2.** *Let $\mathbf{X}_{\mathcal{V}}$ be input node features, $\hat{\mathbf{X}}_{\mathcal{V}}$ be the output of the MP-Module of an UniGCNII/AllDeepSets/ED-HNN/Deep-HGNN with $L$ MP layers, and $\alpha \in [0,1)$. Assume that learnable parameters and the non-linearity are removed from the module. For each model, given a hypergraph $\mathcal{H} = \{\mathcal{V}, \mathcal{E}, \mathbf{H}\}$, there exists a clique expansion $\mathcal{G} = \{\mathcal{V}, \mathbf{W}\}$ to unify its output as the following formula $\hat{\mathbf{X}}_{\mathcal{V}} = \Big((1-\alpha)^L\mathbf{W}^L + \alpha\sum_{l=0}^{L-1}(1-\alpha)^l\mathbf{W}^l\Big)\mathbf{X}_{\mathcal{V}}$.*

This proposition shows that, via our simplification, the selected models can be unified into a single formula. Based on this observation, our TF-MP-Module is designed as the following equation:

$$\hat{\mathbf{X}} = \mathbf{S}\mathbf{X}_{\mathcal{V}},$$

where $\mathbf{S} = (1-\alpha)^L\mathbf{W}^L + \alpha\sum_{l=0}^{L-1}(1-\alpha)^l\mathbf{W}^l$. Intuitively, in this formula, a larger $\alpha$ value emphasizes the retention of the node's inherent information, while a smaller $\alpha$ value increases the influence of information from neighbouring nodes. We define that the $L$ used to compute $\mathbf{S}$ is the number of layers of an TF-MP-Module. To understand the behaviour of an $L$-layer TF-MP-Module, we present the Proposition 3.3. For conciseness, we prove it in Appendix E.

**Proposition 3.3.** *Let $\mathcal{H} = (\mathcal{V}, \mathcal{E}, \mathbf{H})$ be a hypergraph and $\mathcal{G} = (\mathcal{V}, \mathbf{W})$ be its clique expansion with self-loops. Assume that $\alpha \in [0,1)$, and $\mathbf{S} = (1-\alpha)^L\mathbf{W}^L + \alpha\sum_{l=0}^{L-1}(1-\alpha)^l\mathbf{W}^l$. Then, for $i \neq j$, we have $\mathbf{S}_{ij} > 0$ if and only if $v_i$ is an $L$-hop neighbour of $v_j$ on $\mathcal{H}$, and $\mathbf{S}_{ij} = 0$ otherwise.*

By Proposition 3.3, an $L$-layer TF-MP-Module enables the information exchange between any node $v_i$ and its $L$-hop neighbours on the hypergraph. In the next paragraph, we detail the generation of $\mathbf{S}$.

**Operator design.** To use the TF-MP-Module in our TF-HNN, the key is designing $\mathbf{W}$ to generate $\mathbf{S}$. We achieve this using a hyperedge-size-based edge weight that is defined as:

$$\mathbf{W}_{H_{ij}} = \sum_{k=1}^{m} \frac{\delta(v_i, v_j, e_k)}{\mathbf{D}_{\mathcal{H}\mathcal{E}_{kk}}}, \tag{5}$$

where $\delta(\cdot)$ is a function that returns 1 if $e_k$ connects $v_i$ and $v_j$ and returns 0 otherwise. We discuss the connection between the clique expansion mentioned in Proposition 3.2 and the one defined in Eq (5) in Appendix T. The main intuition behind the design of this edge weight is that, under certain conditions, the value of $\mathbf{W}_{H_{ij}}$ is positively correlated with the probability of nodes $v_i$ and $v_j$ having the same label. Consequently, this edge weight design can make the weighted clique expansion fit the homophily assumption, which is that connected nodes tend to be similar to each other (McPherson et al., 2001). Appendix F elaborates on this intuition. Furthermore, inspired by previous works in the graph domain (Wu et al., 2020), we generate $\mathbf{S}$ based on a symmetrically normalised $\mathbf{W}_H$ with the self-loop. Specifically, we generate the $\mathbf{S}$ by:

$$\mathbf{S} = (1-\alpha)^L(\tilde{\mathbf{D}}_H^{-1/2}\tilde{\mathbf{W}}_H\tilde{\mathbf{D}}_H^{-1/2})^L + \alpha\sum_{l=0}^{L-1}(1-\alpha)^l(\tilde{\mathbf{D}}_H^{-1/2}\tilde{\mathbf{W}}_H\tilde{\mathbf{D}}_H^{-1/2})^l, \tag{6}$$

where $\tilde{\mathbf{W}}_H = \mathbf{W}_H + \mathbf{I}_n$ and $\tilde{\mathbf{D}}_H \in \mathbb{R}^{n \times n}$ is the diagonal node degree matrix of $\tilde{\mathbf{W}}_H$. In the next section, we present the theoretical analysis to support the efficiency and effectiveness of our TF-HNN.

# 4 THEORETICAL ANALYSIS

In this section, we begin by showing the efficiency of the proposed TF-HNN by comparing its training complexity with the traditional HNNs. Additionally, we demonstrate the effectiveness of our TF-HNN by analyzing its information utilisation and its robustness against the oversmoothing issue.

**Training complexity.** As illustrated by Eq. (1), the TF-HNN in downstream tasks requires training only the task-specific module. Additionally, since the TF-HNN relies on a pre-defined propagation operator, its computations can be handled during data pre-processing. Consequently, the training complexity of the TF-HNN is determined solely by the task-specific module. In contrast, on the basis of Eq. (2), the HNNs involve training both the MP-Module and the task-specific module. The MP-Module in HNNs includes complex learnable operations that facilitate message passing between connected nodes on a hypergraph, inherently making the training of the HNNs more complex than the TF-HNN. See Figure 1 for an example that qualitatively compares the training pipelines of the TF-HNN and HNN in node classification. To provide a quantitative comparison, we summarize the training computational complexity of the TF-HNN and the four state-of-the-art HNNs in Table 1. This table shows that the training complexity of the TF-HNN is consistently lower than that of the HNN, theoretically confirming the training efficiency of the proposed TF-HNN.

**Information utilisation.** The core strength of existing HNNs lies in their ability to utilise the information embedded within hypergraph structures to generate powerful node features for downstream tasks. Consequently, we analyse the effectiveness of our proposed TF-HNN by comparing its information utilisation capabilities with those of existing HNNs and conclude that our methods utilise the same amount of information as exising HNNs without any loss, which is summarised in Proposition 4.1. For conciseness, we show the detailed proof for Proposition 4.1 in Appendix G.

**Proposition 4.1.** *Let $H_0^{v_i^L}$ be the entropy of information[3] used by an HNN with $L$ feature aggregation functions in generating the features of a node $v_i$, $H_1^{v_i^L}$ be the entropy of information used by an TF-HNN with an $L$-layer TF-MP-Module for the same purpose, and $H_2^{v_i^L}$ denote the entropy of information within node $v_i$ and its $L$-hop neighbours on the hypergraph. Then, $H_0^{v_i^L} = H_1^{v_i^L} = H_2^{v_i^L}$.*

The proposition above theoretically demonstrates that TF-HNN is as effective as existing HNNs in utilising information from pre-defined hypergraphs to generate node features. Specifically, both TF-HNN and existing HNNs achieve this by aggregating neighbourhood information.

**Robustness against the oversmoothing issue.** According to Proposition 3.3, TF-HNN can use global interactions within the given hypergraph to generate node features by deepening the TF-MP-Module. However, as shown in a recent study (Chen et al., 2022), the message-passing-based models on hypergraphs may suffer from the oversmoothing issue. This issue refers to the tendency of a model to produce indistinguishable features for nodes in different classes as the model depth increases, which might degrade the performance of the framework in downstream tasks. Consequently, to further support the effectiveness of TF-HNN, we analyse its robustness against the oversmoothing issue. Inspired by works in the graph domain (Yang et al., 2021; Zhang et al., 2020; Singh et al., 2025), we show Proposition 4.2, with the proof provided in Appendix H.

**Proposition 4.2.** *Let $\mathcal{H} = \{\mathcal{V}, \mathcal{E}, \mathbf{H}\}$ denote a hypergraph, $\mathcal{G} = \{\mathcal{V}, \mathbf{W}_H\}$ be its clique expansion with edge weights computed by Eq. (5), $\mathbf{L}$ be the graph Laplacian matrix of $\mathcal{G}$ computed by a symmetrically normalised and self-loops added $\mathbf{W}_H$, $\mathbf{X}_\mathcal{V}$ represent the input node features, and $\alpha \in [0, 1)$ be a hyperparameter. We define that $F(\mathbf{X}) = \mathrm{tr}(\mathbf{X}^\top \mathbf{L} \mathbf{X}) + \frac{\alpha}{1-\alpha} \mathrm{tr}[(\mathbf{X} - \mathbf{X}_\mathcal{V})^\top (\mathbf{X} - \mathbf{X}_\mathcal{V})]$, and $F_{min}$ as the global minimal value of $F(\mathbf{X})$ for $\mathbf{X} \in \mathbb{R}^{n \times d}$. Assume that: 1) $\hat{\mathbf{X}}_\mathcal{V} = \mathbf{S}\mathbf{X}_\mathcal{V}$; 2) $\mathbf{S}$ is computed by Eq. (6); 3) $\alpha > 0$; and 4) $L \to +\infty$. Then, $F(\hat{\mathbf{X}}_\mathcal{V}) = F_{min}$.*

Notably, minimising the first term of $F(\mathbf{X})$ enhances feature similarity among connected nodes on $\mathcal{H}$, and minimising the second term of $F(\mathbf{X})$ encourages the generated features of each node to retain

---

[3]We use "entropy of information" in a conceptual way - A detailed discussion is in Appendix U.

Table 2: Dataset statistics. More details of these datasets are in Appendix I.

| Name | # Nodes | # Hyperedges | # Features | # Classes | Node Definition | Hyperedge Definition |
|------|---------|--------------|------------|-----------|-----------------|----------------------|
| Cora-CA | 2708 | 1072 | 1433 | 7 | Paper | Co-authorship |
| DBLP-CA | 41302 | 22363 | 1425 | 6 | Paper | Co-authorship |
| Citeseer | 3312 | 1079 | 3703 | 6 | Paper | Co-citation |
| Congress | 1718 | 83105 | 100 | 2 | Congressperson | Legislative-Bills-Sponsorship |
| House | 1290 | 340 | 100 | 2 | Representative | Committee |
| Senate | 282 | 315 | 100 | 2 | Congressperson | Legislative-Bills-Sponsorship |
| Trivago | 172738 | 233202 | 300 | 160 | Accommodation | Browsing Session |

the distinct information from its input features. Therefore, node features that can minimise $F(\mathbf{X})$ capture two key properties: 1) neighbouring nodes on $\mathcal{H}$ have similar features; and 2) each node contains unique information reflecting its individual input characteristics.

Based on Proposition 4.2, a deep TF-MP-Module tends to assign similar features to connected nodes on a hypergraph while maintaining the unique input information of each node. Hence, even with an extremely large number of layers, the TF-MP-Module can preserve the unique input information of each node, making it robust against the oversmoothing issue. This property further ensures the effectiveness of our TF-HNN. Sec. 6 empirically supports the analysis presented in this section.

## 5 RELATED WORK

Recent efforts have leveraged hypergraphs to improve node feature learning for downstream tasks (Bick et al., 2023; Liu et al., 2024; Xu et al., 2024). The predominant models in this area are hypergraph neural networks (HNNs) with a message passing module that enables information exchange between connected nodes (Antelmi et al., 2023). However, as noted for their graph counterparts (Wu et al., 2019a; Frasca et al., 2020), HNNs suffer from low learning efficiency. Two recent works attempt to mitigate this limitation (Feng et al., 2024; Tang et al., 2024). In (Feng et al., 2024), an HNN named HGNN (Feng et al., 2019) is distilled into an MLP during training; while in (Tang et al., 2024), hypergraph structural information is integrated into an MLP via a loss function computed by the sum of the maximum node feature distance in each hyperedge. These approaches primarily focus on reducing model inference complexity via a dedicated training process. However, these designs still lead to hypergraph machine learning models with computationally intensive training procedures. For instance, the distillation process in (Feng et al., 2024) necessitates training a HNN with a computational complexity of $\mathcal{O}(M + TLm'd + TLnd^2)$, where with $M$ is the training computational complexity of the task-specific module, $n$ is the node count, $m'$ is the number of edges in the clique expansion, $L$ is the number of layers, and $d$ is the feature dimension. Given that many existing models are applied to semi-supervised hypergraph node classification settings, which involve the simultaneous use of training and test data (Huang & Yang, 2021; Chien et al., 2022; Wang et al., 2023a;b; Joachims et al., 1999; Chapelle et al., 2009), reducing the training complexity of hypergraph-based models remains a critical yet underexplored challenge. We address this challenge with a novel model named TF-HNN, which is the first to decouple the processing of hypergraph structural information from model training.

## 6 EXPERIMENTS

### 6.1 EXPERIMENTAL SETUP

**Task description.** We conduct experiments in two tasks: hypergraph node classification and hyperlink prediction. In hypergraph node classification (Wang et al., 2023a;b; Duta et al., 2023), we are given the hypergraph structure $\mathbf{H}$, node features $\mathbf{X}_{\mathcal{V}}$, and a set of labelled nodes with ground truth labels $\mathbf{Y}_{lab} = \{\mathbf{y}_v\}_{v \in \mathcal{V}_{lab}}$, where $\mathbf{y}_{v_i} \in \{0,1\}^c$ be a one-hot label. Our objective is to classify the unlabeled nodes within the hypergraph. In hyperlink prediction (Chen & Liu, 2023), we are given node features $\mathbf{X}_{\mathcal{V}}$, a set of observed real and fake hyperedges $\mathcal{E}_{ob}$, and a set of potential hyperedges $\mathcal{E}_p$. The task requires the model to distinguish between real and fake hyperedges in $\mathcal{E}_p$ using $\mathbf{X}_{\mathcal{V}}$ and $\mathcal{E}_{ob}$.

**Dataset and baseline.** We conduct experiments on seven real-world hypergraphs: Cora-CA, DBLP-CA, Citeseer, Congress, House, Senate, which are from (Chien et al., 2022), and Trivago from (Kim et al., 2023). We use nine HNNs as baselines. Five of these baselines utilize direct feature aggregation, including HGNN (Feng et al., 2019), HCHA (Bai et al., 2021), UniGCNII (Huang & Yang, 2021), PhenomNN (Wang et al., 2023b), and Deep-HGNN (Chen et al., 2022). The remaining four baselines

Table 3: Node classification accuracy (%) for HNNs and TF-HNN. The best result on each dataset is highlighted in **bold font**. The second and third highest accuracies are marked with an underline.

| | Cora-CA | DBLP-CA | Citeseer | Congress | House | Senate | Avg. Mean |
|---|---|---|---|---|---|---|---|
| HGNN | 82.64 ± 1.65 | 91.03 ± 0.20 | 72.45 ± 1.16 | 91.26 ± 1.15 | 61.39 ± 2.96 | 48.59 ± 4.52 | 74.56 |
| HCHA | 82.55 ± 0.97 | 90.92 ± 0.22 | 72.42 ± 1.42 | 90.43 ± 1.20 | 61.36 ± 2.53 | 48.62 ± 4.41 | 74.38 |
| HNHN | 77.19 ± 1.49 | 86.78 ± 0.29 | 72.64 ± 1.57 | 53.35 ± 1.45 | 67.80 ± 2.59 | 50.93 ± 6.33 | 68.12 |
| UniGCNII | 83.60 ± 1.14 | 91.69 ± 0.19 | 73.05 ± 2.21 | 94.81 ± 0.81 | 67.25 ± 2.57 | 49.30 ± 4.25 | 76.62 |
| AllDeepSets | 81.97 ± 1.50 | 91.27 ± 0.27 | 70.83 ± 1.63 | 91.80 ± 1.53 | 67.82 ± 2.40 | 48.17 ± 5.67 | 75.31 |
| AllSetTransformer | 83.63 ± 1.47 | 91.53 ± 0.23 | 73.08 ± 1.20 | 92.16 ± 1.05 | 51.83 ± 5.22 | 69.33 ± 2.20 | 76.93 |
| PhenomNN | 85.81 ± 0.90 | **91.91 ± 0.21** | **75.10 ± 1.59** | 88.24 ± 1.47 | 70.71 ± 2.35 | 67.70 ± 5.24 | 79.91 |
| ED-HNN | 83.97 ± 1.55 | 91.90 ± 0.19 | 73.70 ± 1.38 | 95.00 ± 0.99 | 72.45 ± 2.28 | 64.79 ± 5.14 | 80.30 |
| Deep-HGNN | 84.89 ± 0.88 | 91.76 ± 0.28 | 74.07 ± 1.64 | 93.91 ± 1.18 | 75.26 ± 1.76 | 68.39 ± 4.79 | 81.38 |
| TF-HNN (ours) | **86.54 ± 1.32** | 91.80 ± 0.30 | 74.82 ± 1.67 | **95.09 ± 0.89** | **76.29 ± 1.99** | 70.42 ± 2.74 | **82.50** |

Figure 2: The relative training time required for HNNs and TF-HNN to achieve optimal performance.

employ indirect feature aggregation, consisting of HNHN (Dong et al., 2020), AllDeepSets (Chien et al., 2022), AllSetTransformer (Chien et al., 2022), and ED-HNN (Wang et al., 2023a). The dataset statistics are summarized in Table 2, with detailed descriptions of datasets available in Appendix I.

**Metric.** Following previous works (Wang et al., 2023a; Chen & Liu, 2023), we evaluate models for node classification and hyperlink prediction with classification accuracy and the Area Under the ROC Curve (AUC), respectively. Similar to Wu et al., we employ the relative training time $r_f$, defined as $r_f = t_f/t_s$ to evaluate the efficiency of the models, where $t_f$ is the training time to achieve optimal performance for the evaluated model, and $t_s$ is the training time for optimal performance of TF-HNN.

**Implementation.** For node classification, we follow previous works (Wang et al., 2023a; Duta et al., 2023) to use a 50%/25%/25% train/validation/test data split and adapt the baseline classification accuracy from them[4]. Additionally, similar to these works, we implement the classifier based on MLP for our TF-HNN and report the results from ten runs. For hyperlink prediction, existing works (Chen & Liu, 2023) primarily focus on the design of the prediction head, with no reported results for applying our baseline HNNs to this task, so we report all results from five runs conducted by ourselves. We employ the deep set function implemented by (Chien et al., 2022) as the prediction head for our method and baseline methods due to its simplicity. We use a 50%/25%/25% train/validation/test data split and ensure each split contains five times as many fake hyperedges as real hyperedges. TF-HNN and HNNs only use real hyperedges in training and validation sets for message passing. Experiments were on RTX 3090 GPUs by PyTorch. See our code here. More details are in Appendix L and S.

## 6.2 COMPARISON WITH BASELINES

**Node classification.** We summarize the classification accuracy and relative training time of the TF-HNN and HNNs in Table 3 and Figure 2, respectively. Table 3 demonstrates that TF-HNN not only leads to the best results across four datasets (Cora-CA, Congress, House, and Senate) but also results in the highest average mean accuracy overall. These findings confirm the effectiveness of TF-HNN in generating powerful node features for node classification. As discussed in Sec. 4, we attribute this effectiveness to the ability of TF-HNN to match the power of existing HNNs in utilising neighbourhood information to generate node features, while requiring the optimization of fewer parameters. This makes the TF-HNN more efficient in utilising training data and reduces the risk of overfitting, thereby outperforming more complex HNN counterparts. On the other hand, as shown in Figure 2, the TF-HNN consistently needs less training time to achieve superior performance compared to the HNNs. This highlights the high training efficiency of TF-HNN in downstream tasks.

**Hyperlink prediction.** Figure 3 shows the results for hyperlink prediction, which exhibit a pattern similar to that observed in the hypergraph node classification task: the TF-HNN outperforms the

---

[4]Since there are no reported results for Deep-HGNN under the chosen data split, we use our own results.

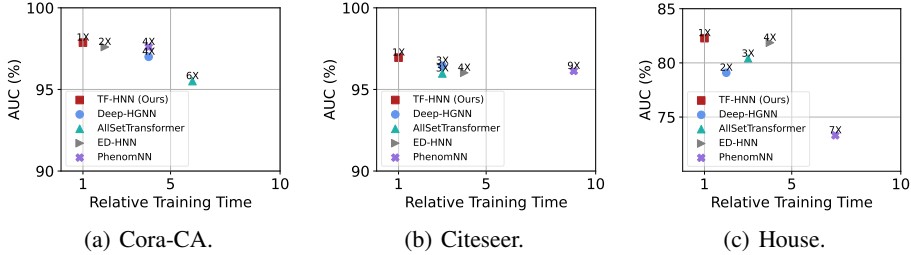

Figure 3: Hyperlink prediction AUC (%) and relative training time for HNNs and TF-HNN.

Table 4: Node classification accuracy (%) and training time for models on Trivago.

| Metric / Methods | Deep-HGNN | ED-HNN | TF-HNN (Ours) |
|---|---|---|---|
| Accuracy (%) | 84.06 ± 1.70 | 48.38 ± 1.35 | **94.03 ± 0.51** |
| Training Time (s) | 3352.35 ± 1039.83 | 143.92 ± 34.64 | **38.09 ± 6.65** |

Table 5: Hyperlink prediction AUC (%) and training time for models on Trivago.

| Metric / Methods | Deep-HGNN | ED-HNN | TF-HNN (Ours) |
|---|---|---|---|
| AUC (%) | 84.26 ± 1.62 | 69.44 ± 7.09 | **95.71 ± 1.00** |
| Training Time (s) | 1873.67 ± 559.55 | 1311.01 ± 627.99 | **484.00 ± 110.99** |

Figure 4: The impact from the hyperparameters of TF-MP-Module in node classification.

HNNs while requiring less training time. An interesting observation is that the training efficiency improvement from TF-HNN is smaller in the hyperlink prediction task compared to the node classification task. The intuition is that the task-specific module used for hyperlink prediction is more computationally expensive than the one used for hypergraph node classification. Specifically, the hyperlink predictor needs to generate a prediction for each hyperedge by aggregating the features of the nodes connected by the hyperedge, whereas the node classifier only needs to perform forward propagation for each individual node. Appendix M discusses that the efficiency improvement provided by TF-HNN is inversely correlated with the complexity of the task-specific module.

**Scalability of state-of-the-art models.** We compare the scalability of our TF-HNN against the top two baseline methods from previous experiments, Deep-HGNN and ED-HNN. Deep-HGNN uses a direct feature aggregation function, while ED-HNN uses an indirect feature aggregation function. We evaluate the performance of all methods on node classification and hyperlink prediction using the large-scale hypergraph benchmark, Trivago, with the results detailed in Table 4 and Table 5, respectively. The results show that TF-HNN not only surpasses its competitors in performance but also dramatically reduces training time. Remarkably, in node classification, TF-HNN improves the accuracy by around $10\%$ over the best baseline, Deep-HGNN, while using only about $1\%$ of its training time. These findings underscore the value of TF-HNN for large-scale hypergraphs.[5] Due to the space limit, additional discussions and experiments are in Appendix N, O, P, R, and V.

## 6.3 ANALYSIS

We conduct a series of analyses on the components within the proposed TF-HNN model in node classification on the Cora-CA, Citeseer, House, and Senate datasets.

**Impacts from the TF-MP-Module.** In this study, we evaluate the effectiveness of our proposed TF-MP-Module within the TF-HNN framework by comparing it to two training-free message passing methods developed for graphs: SGC (Wu et al., 2019a) and SIGN (Frasca et al., 2020). To adapt

---

[5] On Trivago, Deep-HGNN's training time is much higher for node classification than hyperlink prediction. This is because a 64-layer model largely improves node classification accuracy, while a 2-layer model suffices for hyperlink prediction, as deeper models offer no benefit. The extra layers account for the longer training time.

Table 6: The impact from the TF-MP-Module in node classification. The best results are in **bold font**.

|  | Cora-CA | Citeseer | Senate | House | Avg. Mean |
|---|---|---|---|---|---|
| SGC with CE | $81.60 \pm 1.68$ | $71.42 \pm 2.15$ | $47.18 \pm 6.48$ | $56.16 \pm 3.21$ | 64.09 |
| SIGN with CE | $86.04 \pm 1.53$ | $73.62 \pm 1.79$ | $64.93 \pm 3.15$ | $73.75 \pm 1.91$ | 74.59 |
| TF-MP-Module (ours) | $\mathbf{86.54 \pm 1.32}$ | $\mathbf{74.82 \pm 1.67}$ | $\mathbf{70.42 \pm 2.74}$ | $\mathbf{76.29 \pm 1.99}$ | **77.02** |

Table 7: The impact from the weighted $\mathbf{S}$ in node classification. The best results are in **bold font**.

|  | Cora-CA | Citeseer | Senate | House | Avg. Mean |
|---|---|---|---|---|---|
| TF-HNN without Weighted $\mathbf{S}$ | $84.90 \pm 1.55$ | $74.75 \pm 1.62$ | $68.87 \pm 4.57$ | $75.57 \pm 1.81$ | 76.02 |
| TF-HNN with Weighted $\mathbf{S}$ | $\mathbf{86.54 \pm 1.32}$ | $\mathbf{74.82 \pm 1.67}$ | $\mathbf{70.42 \pm 2.74}$ | $\mathbf{76.29 \pm 1.99}$ | **77.02** |

these methods to hypergraphs, we first project the given hypergraph into its clique expansion (CE), as introduced in Sec. 2, and then apply SGC and SIGN to this projected structure. Notably, we only replace the TF-MP-Module with SGC using CE and SIGN with CE, while keeping all other components of the TF-HNN unchanged. Table 6 show that our TF-MP-Module consistently outperforms the alternatives across all datasets. The superior performance of our method over SGC with CE is due to SGC with CE being a special case of our TF-MP-Module with $\alpha = 0$, which, as discussed in Proposition 4.2, can cause node features to become overly similar to their neighbours, potentially reducing distinctiveness. For SIGN with CE, the method concatenates features from different neighbor hops with the initial node features and processes them through a classifier, forcing it to handle both feature merging and classification, which increases complexity and can hinder convergence. In contrast, our TF-MP-Module directly merges neighbouring node information using the operator in Eq. (6), allowing the classifier to focus solely on classification, thereby simplifying the learning process and improving performance. These findings highlight the greater capacity of our TF-MP-Module compared to existing graph-based training-free modules in processing hypergraph structures, further underscoring its unique value for hypergraph machine learning.

**Impacts from hyperparameters of TF-MP-Module.** We focus on two key hyperparameters of the TF-MP-Module: the number of layers and the $\alpha$. As illustrated in Figure 4, models with $\alpha > 0$ exhibit more consistent accuracy as the number of layers increases compared to models with $\alpha = 0$. We attribute this consistency to the robustness discussed in Proposition 4.2, where a positive $\alpha$ helps retain the distinct information of each node, thereby mitigating the oversmoothing issue. Additionally, we observe that a positive $\alpha$ significantly enhances performance in the House and Senate datasets, which are more heterophilic, compared to the Cora-CA and Citeseer datasets. This suggests that retaining the distinct information of each node is more beneficial for heterophilic datasets.

**Impacts from the weighted S.** We define $\mathbf{S}$ based on Eq. (5) as weighted $\mathbf{S}$. In contrast, in this experiment, for the framework without weighted $\mathbf{S}$, $\mathbf{S}$ is generated using an adjacency matrix $\mathbf{W}$ of a clique expansion with all positive values set to one. As shown in Table 7, adding weighted $\mathbf{S}$ further enhances performance, highlighting its informativeness.

## 7 CONCLUSION

In this paper, we propose a novel model called TF-HNN. The key innovation of TF-HNN is an original training-free message passing module (TF-MP-Module) tailored for data on hypergraphs. We present both theoretical and empirical evidence demonstrating that TF-HNN can efficiently and effectively address hypergraph-related downstream tasks. To our knowledge, TF-HNN is the first model to shift the integration of hypergraph structural information from the model training stage to the data pre-processing stage, significantly enhancing training efficiency. The proposed TF-HNN can advance the field of hypergraph machine learning research by not only improving the efficiency of using observed hypergraph structures to solve downstream tasks, but also serving as a simple starting point for the development of future hypergraph machine learning models.

**Limitation and future work.** Learning tasks on hypergraphs can be defined at the node, hyperedge, and hypergraph levels. In this paper, we test our TF-HNN on node-level and hyperedge-level tasks, with the main limitation being the lack of results on hypergraph-level tasks. Designing task-specific modules to generate hypergraph-level features based on existing node features remains a challenging topic in the literature. Therefore, we consider the application of our TF-HNN to hypergraph-level tasks as future work. Finally, developing an automated and efficient hyperparameter search method for hypergraph neural networks presents a promising avenue for further enhancing model efficiency.

ACKNOWLEDGEMENT

Keyue Jiang was supported by the UKRI Engineering and Physical Sciences Research Council (EPSRC) [grant number EP/R513143/1]. Siheng Chen gratefully acknowledges support from the National Natural Science Foundation of China under Grant 62171276. Xiaowen Dong acknowledges support from the Oxford-Man Institute of Quantitative Finance, the EPSRC (EP/T023333/1), and the Royal Society (IEC \NSFC \211188).

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

## A    IMPACT STATEMENTS

This paper presents work that aims to advance the field of Machine Learning. There are many potential societal consequences of our work, none of which we feel must be specifically highlighted.

## B    BACKGROUND OF HYPERGRAPH MESSAGE PASSING

**Clique expansion.** Given a hypergraph $\mathcal{H} = \{\mathcal{V}, \mathcal{E}, \mathbf{H}\}$, its clique expansion is defined as a graph $\mathcal{G} = \{\mathcal{V}, \mathbf{W}\}$, where $\mathcal{V}$ remains unchanged, and $\mathbf{W}_{ij} > 0$ if and only if $v_i$ and $v_j$ are connected by a hyperedge on $\mathcal{H}$ and $\mathbf{W}_{ij} = 0$ otherwise. Hence, each hyperedge in $\mathcal{H}$ is a clique in $\mathcal{G}$.

**Star expansion.** Given a hypergraph $\mathcal{H} = \{\mathcal{V}, \mathcal{E}, \mathbf{H}\}$, its star expansion is defined as a bipartite graph $\mathcal{G} = \{\mathcal{V} \bigcup \mathcal{V}', \mathbf{W}\}$, where $v_i \in \mathcal{V}$ is a node in $\mathcal{H}$, hypernode $v_{e_j} \in \mathcal{V}'$ corresponds to the hyperedge $e_j$ in $\mathcal{H}$, and there is an edge between $v_i$ and $v_{e_j}$ if and only if $e_j$ contains $v_i$ in $\mathcal{H}$.

**Line expansion.** Given a hypergraph $\mathcal{H} = \{\mathcal{V}, \mathcal{E}, \mathbf{H}\}$, its line expansion is defined as a graph $\mathcal{G} = (\mathcal{V}_l, \mathbf{W})$. The node set $\mathcal{V}_l$ of $\mathcal{G}$ is defined by vertex-hyperedge pair $\{(v, e) \mid v \in e, v \in \mathcal{V}, e \in \mathcal{E}\}$ from the original hypergraph. The adjacency matrix $\mathbf{W} \in \{0, 1\}^{|\mathcal{V}_l| \times |\mathcal{V}_l|}$ is defined by pairwise relation with $\mathbf{W}(u_l, v_l) = 1$ if either $v = v'$ or $e = e'$ for $u_l = (v, e), v_l = (v', e') \in \mathcal{V}_l$. In the following sections, we refer the nodes on a line expansion as line-nodes. Notably, a node $v_i$ on the original hypergraph corresponds to multiple line-nodes, each representing a vertex-hyperedge pair involving $v_i$. These line-nodes are interconnected and also connect to line-nodes corresponding to the 1-hop neighbors of $v_i$ on the given hypergraph.

**Incidence tensor.** A $k$-uniform hypergraph can be represented by a $k$-dimensional supersymmetric tensor such that for all distinct node sets $\{v_1, \ldots, v_k\} \in \mathcal{V}$, $\mathbf{T}_{i_1, \ldots, i_d} = \frac{1}{(k-2)!}$ if hyperedge $e = \{v_1, \ldots, v_k\} \in \mathcal{E}$, and $\mathbf{T}_{i_1, \ldots, i_d} = 0$ otherwise.

## C    PROOF OF PROPOSITION 3.1

*Proof.* In this section, we provide proofs for the clique-based, star-based, tensor-based and line-based methods, respectively.

For the clique-based approaches, there are two specific designs for this type of approaches. The first design is based on the hypergraph convolution operator (Feng et al., 2019; Yadati et al., 2019; Bai et al., 2021; Duta et al., 2023; Chen et al., 2022), which is derived from the hypergraph Laplacian (Agarwal et al., 2006). The second design (Wang et al., 2023b) relies on an optimization algorithm that minimizes a handcrafted hypergraph energy function. Despite the differences in their designs, these methods start by using an MLP to project the given node features to the latent space, namely, for $i = 0$, we have $\mathbf{x}_{v_i}^{(0)} = \mathrm{MLP}(\mathbf{x}_{v_i})$. Moreover, these methods are analogous to using matrix multiplication between the adjacency matrix of a clique expansion, denoted as $\mathbf{W}$, and the node features $\mathbf{X}_{\mathcal{V}}$ to facilitate information exchange among connected nodes on hypergraphs. Therefore, for $i \in \mathbb{Z}^+$, the node-wise form of this type of methods can be summarized as:

$$\mathbf{x}_{v_i}^{(l)} = f_3\Big(f_0(\mathbf{x}_{v_i}^{(0)}) + f_1(\mathbf{x}_{v_i}^{(l-1)}) + \sum_{v_j \in \mathcal{N}_{\mathcal{G}_{v_i}}} f_2(\mathbf{x}_{v_j}^{(l-1)})\Big), \tag{7}$$

where $f_0(\cdot)$, $f_1(\cdot)$, $f_2(\cdot)$, $f_3(\cdot)$ represent three learnable functions, and $\mathcal{N}_{\mathcal{H}_{v_i}}$ is the set of 1-hop neighbours of $v_i$ on the hypergraph. Here the right-hand side of Eq. (7) is a special case of $\phi_{\boldsymbol{\Theta}}(\mathbf{x}_{v_i}^{(0)}, \mathbf{x}_{v_i}^{(l-1)}, \oplus_{v_j \in \mathcal{N}_{\mathcal{H}_{v_i}}} \mathbf{x}_{v_j}^{(l-1)})$.

For the star-based approaches, three primary designs exist for this type of approach. Despite the differences in their designs, these methods start by using an MLP to project the given node features to the latent space, namely, for $i = 0$, we have $\mathbf{x}_{v_i}^{(0)} = \mathrm{MLP}(\mathbf{x}_{v_i})$. Moreover, the first type of methods, as presented in (Huang & Yang, 2021), directly applies existing GNNs to a star expansion graph, initializing the features of hypernode $v_{e_j}$ as the mean of the features of nodes connected by the hyperedge $e_j$. Due to the linear generation of hypernode features, for $i \in \mathbb{Z}^+$, the node-wise form of this method can be expressed as Eq.(7). Additionally, (Chien et al., 2022) integrates set functions with star expansion to create HNNs, while (Wang et al., 2023a) develops a model by studying the information diffusion process on a hypergraph using star expansion. The node-wise forms of the

Table 8: Overview for four state-of-the-art HNNs. In this table, $\gamma_U, \gamma_E, \gamma_D, \gamma_l' \in (0, 1)$ are hyperparameters and $\mathbf{I} \in \mathbb{R}^{d \times d}$ denotes an identity matrix.

| Name | Type | Message Passing Function |
|---|---|---|
| UniGCNII (Huang & Yang, 2021) | Direct Message Passing | $\mathbf{X}_{\mathcal{V}}^{(l)} = \sigma\left(\left((1-\gamma_U)\mathbf{D}_{\mathcal{H}^{\mathcal{V}}}^{-1/2}\mathbf{H}\tilde{\mathbf{D}}_{\mathcal{H}^{\varepsilon}}^{-1/2}\mathbf{D}_{\mathcal{H}^{\varepsilon}}^{-1}\mathbf{H}^{\top}\mathbf{X}_{\mathcal{V}}^{(l-1)} + \gamma_U\mathbf{X}_{\mathcal{V}}^{(0)}\right)\mathbf{\Theta}^{(l)}\right)$ |
| Deep-HGNN (Chen et al., 2022) | Direct Message Passing | $\mathbf{X}_{\mathcal{V}}^{(l)} = \sigma\left(\left((1-\gamma_D)\mathbf{D}_{\mathcal{H}^{\mathcal{V}}}^{-1/2}\mathbf{H}\mathbf{D}_{\mathcal{H}^{\varepsilon}}^{-1}\mathbf{H}^{\top}\mathbf{D}_{\mathcal{H}^{\mathcal{V}}}^{-1/2}\mathbf{X}_{\mathcal{V}}^{(l)} + \gamma_D\mathbf{X}_{\mathcal{V}}^{(0)}\right)\left((1-\gamma_D')\mathbf{I} + \gamma_D'\mathbf{\Theta}^{(l)}\right)\right)$ |
| AllDeepSets (Chien et al., 2022) | Indirect Message Passing | $\mathbf{X}_{\mathcal{V}}^{(l)} = \text{MLP}\left(\mathbf{D}_{\mathcal{H}^{\mathcal{V}}}^{-1}\mathbf{H}\,\text{MLP}\left(\mathbf{D}_{\mathcal{H}^{\varepsilon}}^{-1}\mathbf{H}^{\top}\text{MLP}(\mathbf{X}_{\mathcal{V}}^{(l-1)})\right)\right)$ |
| ED-HNN (Wang et al., 2023a) | Indirect Message Passing | $\mathbf{X}_{\mathcal{V}}^{(l)} = \text{MLP}\left[(1-\gamma_E)\,\text{MLP}\left(\mathbf{D}_{\mathcal{H}^{\mathcal{V}}}^{-1}\mathbf{H}\,\text{MLP}\left(\mathbf{D}_{\mathcal{H}^{\varepsilon}}^{-1}\mathbf{H}^{\top}\text{MLP}(\mathbf{X}_{\mathcal{V}}^{(l-1)})\right)\right) + \gamma_E\mathbf{X}_{\mathcal{V}}^{(0)}\right]$ |

models in (Chien et al., 2022) and (Wang et al., 2023a) can be summarized as:

$$\mathbf{x}_{v_i}^{(l)} = g_2\Big(g_0(\mathbf{x}_{v_i}^{(0)}) + \mathbf{p}^{\top}g_1\big(\mathbf{x}_{v_i}^{(l-1)}, \oplus_{v_j \in \mathcal{N}_{\mathcal{H}_{v_i}}}\mathbf{x}_{v_j}^{(l-1)}\big)\Big), \tag{8}$$

$g_0(\cdot), g_1(\cdot), g_2(\cdot)$ represent three learnable functions, $\mathcal{N}_{\mathcal{H}_{v_i}}$ is the set of 1-hop neighbours of $v_i$ on the hypergraph, and $\mathbf{p} \in \mathbb{R}^{m_{v_i}}$ is a vector used to aggregate the features of hypernodes generated by $g_1(\cdot)$. Here, $m_{v_i}$ is the number of hyperedges containing $v_i$. Here the right-hand side of Eq. (8) is a special case of $\phi_{\mathbf{\Theta}}(\mathbf{x}_{v_i}^{(0)}, \mathbf{x}_{v_i}^{(l-1)}, \oplus_{v_j \in \mathcal{N}_{\mathcal{H}_{v_i}}}\mathbf{x}_{v_j}^{(l-1)})$.

For tensor-based approaches, these approaches facilitate message passing among connected nodes using the incidence tensor. These methods start by using an MLP to project the given node features to the latent space, namely, for $i = 0$, we have $\mathbf{x}_{v_i}^{(0)} = \text{MLP}(\mathbf{x}_{v_i})$. Moreover, as noted by the Theorem 3.3 of (Chien et al., 2022), practically, they are implemented by combining set functions with a star expansion. Consequently, for $i \in \mathbb{Z}^+$, the node-wise forms of these methods are specific instances of Eq. (8), where the right-hand side is a special case of $\phi_{\mathbf{\Theta}}(\mathbf{x}_{v_i}^{(0)}, \mathbf{x}_{v_i}^{(l-1)}, \oplus_{v_j \in \mathcal{N}_{\mathcal{H}_{v_i}}}\mathbf{x}_{v_j}^{(l-1)})$.

For line-based approaches, these approaches (Yang et al., 2022) generate node features using hypergraph structural information in four steps. Firstly, these methods start by using an MLP to project the given node features to the latent space, namely, for $i = 0$, we have $\mathbf{x}_{v_i}^{(0)} = \text{MLP}(\mathbf{x}_{v_i})$. Then, the hypergraph is projected onto its corresponding line expansion graph, where the features of each line-node are initialised as the features of its corresponding node on the original hypergraph. Next, a GNN is applied to this line expansion graph to facilitate message passing among connected line-nodes. Finally, the features of each node $v_i$ on the hypergraph are generated by aggregating the features of the line-nodes corresponding to $v_i$. This method can be summarized as Eq. (8), with three key differences from star-based approaches. Firstly, $\mathcal{N}_{\mathcal{H}_{v_i}}$ contains the $m$-hop neighbors of $v_i$, which includes all nodes with a path to $v_i$. Secondly, the line-based method utilises only one layer for generating node features, with the number of feature aggregation functions usually referring to the number of GNNs used to implement $g_1(\cdot)$. As a result, for $i \in \mathbb{Z}^+$, we still have $\mathbf{x}_{v_i}^{(l)} = \phi_{\mathbf{\Theta}}(\mathbf{x}_{v_i}^{(0)}, \mathbf{x}_{v_i}^{(l-1)}, \oplus_{v_j \in \mathcal{N}_{\mathcal{H}_{v_i}}}\mathbf{x}_{v_j}^{(l-1)})$.

$\square$

# D    PROOF OF PROPOSITION 3.2

To recap, the overview of UniGCNII, Deep-HGNN, AllDeepSets, and ED-HNN is in Table 8.

Before proving Proposition 3.2, we first prove the following Lemmas.

**Lemma D.1.** *Let $\mathbf{H} \in \{0, 1\}^{n \times m}$ be an incidence matrix of a hypergraph, $\mathbf{D}_{\mathcal{H}^{\mathcal{V}}} \in \mathbb{R}^{n \times n}$ is a diagonal matrix with node degrees, $\mathbf{D}_{\mathcal{H}^{\varepsilon}} \in \mathbb{R}^{m \times m}$ is a diagonal matrix with hyperedge degrees, $\tilde{\mathbf{D}}_{\mathcal{H}^{\varepsilon}_{jj}} \in \mathbb{R}^{m \times m}$ is a diagonal matrix with $\tilde{\mathbf{D}}_{\mathcal{H}^{\varepsilon}_{jj}} = \frac{\sum_{i=1}^{n}\mathbf{H}_{ij}\mathbf{D}_{\mathcal{H}^{\mathcal{V}}_{ii}}}{\mathbf{D}_{\mathcal{H}^{\varepsilon}_{jj}}}$, $\gamma_U \in (0, 1)$ and $\mathbf{W}_U = (1 - \gamma_U)\mathbf{D}_{\mathcal{H}^{\mathcal{V}}}^{-1/2}\mathbf{H}\tilde{\mathbf{D}}_{\mathcal{H}^{\varepsilon}}^{-1/2}\mathbf{D}_{\mathcal{H}^{\varepsilon}}^{-1}\mathbf{H}^{\top}$. We have: $\mathbf{W}_U$ is the adjacency matrix of a clique expansion.*

*Proof.* We have:

$$\mathbf{W}_{U_{ij}} = (1 - \gamma_U)\sum_{k=1}^{m}\frac{\mathbf{H}_{ik}\mathbf{H}_{jk}}{\mathbf{D}_{\mathcal{H}^{\mathcal{V}}_{ii}}^{1/2}\mathbf{D}_{\mathcal{H}^{\varepsilon}_{kk}}\tilde{\mathbf{D}}_{\mathcal{H}^{\varepsilon}_{kk}}^{1/2}}.$$

Here $\mathbf{W}_{U_{ij}} > 0$ if and only if $v_i$ and $v_j$ are connected by a hyperedge $e_k$ on $\mathcal{H}$, otherwise $\mathbf{W}_{U_{ij}} = 0$. Therefore, $\mathbf{W}_U$ is the adjacency matrix of a clique expansion. $\square$

**Lemma D.2.** *Let $\mathbf{H} \in \{0,1\}^{n \times m}$ be an incidence matrix of a hypergraph, $\mathbf{D}_{\mathcal{H}^{\mathcal{V}}} \in \mathbb{R}^{n \times n}$ is a diagonal matrix with node degrees, $\mathbf{D}_{\mathcal{H}\varepsilon} \in \mathbb{R}^{m \times m}$ is a diagonal matrix with hyperedge degrees, $\gamma_D \in (0,1)$ and $\mathbf{W}_D = (1-\gamma_D)\mathbf{D}_{\mathcal{H}^{\mathcal{V}}}^{-1/2}\mathbf{H}\mathbf{D}_{\mathcal{H}\varepsilon}^{-1}\mathbf{H}^\top\mathbf{D}_{\mathcal{H}^{\mathcal{V}}}^{-1/2}$. We have: $\mathbf{W}_D$ is the adjacency matrix of a clique expansion.*

*Proof.* We have:
$$\mathbf{W}_{D_{ij}} = (1-\gamma_D)\sum_{k=1}^{m} \frac{\mathbf{H}_{ik}\mathbf{H}_{jk}}{\mathbf{D}_{\mathcal{H}_{ii}^{\mathcal{V}}}^{1/2}\mathbf{D}_{\mathcal{H}_{jj}^{\mathcal{V}}}^{1/2}\mathbf{D}_{\mathcal{H}_{kk}^{\varepsilon}}}.$$

Here $\mathbf{W}_{D_{ij}} > 0$ if and only if $v_i$ and $v_j$ are connected by a hyperedge $e_k$ on $\mathcal{H}$, otherwise $\mathbf{W}_{D_{ij}} = 0$. Therefore, $\mathbf{W}_D$ is the adjacency matrix of a clique expansion. $\square$

**Lemma D.3.** *Let $\mathbf{H} \in \{0,1\}^{n \times m}$ be an incidence matrix of a hypergraph, $\mathbf{D}_{\mathcal{H}^{\mathcal{V}}} \in \mathbb{R}^{n \times n}$ be a diagonal matrix with node degrees, and $\mathbf{D}_{\mathcal{H}\varepsilon} \in \mathbb{R}^{m \times m}$ be a diagonal matrix with hyperedge degrees. We have: $\mathbf{W}_S = \mathbf{D}_{\mathcal{H}^{\mathcal{V}}}^{-1}\mathbf{H}\mathbf{D}_{\mathcal{H}\varepsilon}^{-1}\mathbf{H}^\top$ is the adjacency matrix of a clique expansion.*

*Proof.* We have:
$$\mathbf{W}_{S_{ij}} = \sum_{k=1}^{m} \frac{\mathbf{H}_{ik}\mathbf{H}_{jk}}{\mathbf{D}_{\mathcal{H}_{ii}^{\mathcal{V}}}\mathbf{D}_{\mathcal{H}_{kk}^{\varepsilon}}}.$$

Here $\mathbf{W}_{S_{ij}} > 0$ if and only if $v_i$ and $v_j$ are connected by a hyperedge on $\mathcal{H}$, otherwise $\mathbf{W}_{S_{ij}} = 0$. Therefore, $\mathbf{W}_S$ is the adjacency matrix of a clique expansion. $\square$

With the Lemmas above, we present the proof of Proposition 3.2 as follows.

*Proof.* For an UniGNN (Huang & Yang, 2021) with $L$ feature aggregation functions, devoid non-linear activation and by setting the learnable transformation matrix to the identity matrix, its MP-Module can be reformulated as:
$$\mathbf{X}_{\mathcal{V}}^{(L)} = \left((1-\gamma_U)^L(\mathbf{D}_{\mathcal{H}^{\mathcal{V}}}^{-1/2}\mathbf{H}\tilde{\mathbf{D}}_{\mathcal{H}\varepsilon}^{-1/2}\mathbf{D}_{\mathcal{H}\varepsilon}^{-1}\mathbf{H}^\top)^L + \gamma_U\sum_{l=0}^{L-1}(1-\gamma_U)^l(\mathbf{D}_{\mathcal{H}^{\mathcal{V}}}^{-1/2}\mathbf{H}\tilde{\mathbf{D}}_{\mathcal{H}\varepsilon}^{-1/2}\mathbf{D}_{\mathcal{H}\varepsilon}^{-1}\mathbf{H}^\top)^l\right)\mathbf{X}_{\mathcal{V}}.$$

Based on Lemma D.1, this equation is a special case of $\hat{\mathbf{X}}_{\mathcal{V}} = \left((1-\alpha)^L\mathbf{W}^L + \alpha\sum_{l=0}^{L-1}(1-\alpha)^l\mathbf{W}^l\right)\mathbf{X}_{\mathcal{V}}$ with $\hat{\mathbf{X}}_{\mathcal{V}} = \mathbf{X}_{\mathcal{V}}^{(L)}$, $\mathbf{W} = \mathbf{D}_{\mathcal{H}^{\mathcal{V}}}^{-1/2}\mathbf{H}\tilde{\mathbf{D}}_{\mathcal{H}\varepsilon}^{-1/2}\mathbf{D}_{\mathcal{H}\varepsilon}^{-1}\mathbf{H}^\top$ and $\alpha = \gamma_U$.

For a Deep-HGNN (Chen et al., 2022) with $L$ feature aggregation functions, devoid non-linear activation and by setting the learnable transformation matrix to the identity matrix, its MP-Module can be reformulated as:
$$\mathbf{X}_{\mathcal{V}}^{(L)} = \left((1-\gamma_D)^L(\mathbf{D}_{\mathcal{H}^{\mathcal{V}}}^{-1/2}\mathbf{H}\mathbf{D}_{\mathcal{H}\varepsilon}^{-1}\mathbf{H}^\top\mathbf{D}_{\mathcal{H}^{\mathcal{V}}}^{-1/2})^L + \gamma_D\sum_{l=0}^{L-1}(1-\gamma_D)^l(\mathbf{D}_{\mathcal{H}^{\mathcal{V}}}^{-1/2}\mathbf{H}\mathbf{D}_{\mathcal{H}\varepsilon}^{-1}\mathbf{H}^\top\mathbf{D}_{\mathcal{H}^{\mathcal{V}}}^{-1/2})^l\right)\mathbf{X}_{\mathcal{V}}.$$

Based on Lemma D.2, this equation is a special case of $\hat{\mathbf{X}}_{\mathcal{V}} = \left((1-\alpha)^L\mathbf{W}^L + \alpha\sum_{l=0}^{L-1}(1-\alpha)^l\mathbf{W}^l\right)\mathbf{X}_{\mathcal{V}}$ with $\hat{\mathbf{X}}_{\mathcal{V}} = \mathbf{X}_{\mathcal{V}}^{(L)}$, $\mathbf{W} = \mathbf{D}_{\mathcal{H}^{\mathcal{V}}}^{-1/2}\mathbf{H}\mathbf{D}_{\mathcal{H}\varepsilon}^{-1}\mathbf{H}^\top\mathbf{D}_{\mathcal{H}^{\mathcal{V}}}^{-1/2}$ and $\alpha = \gamma_D$.

For an AllDeepSets (Chien et al., 2022) with $L$ feature aggregation functions, devoid non-linear activation and by setting the learnable transformation matrix to the identity matrix, its MP-Module can be reformulated as:
$$\mathbf{X}_{\mathcal{V}}^{(L)} = (\mathbf{D}_{\mathcal{H}^{\mathcal{V}}}^{-1}\mathbf{H}\mathbf{D}_{\mathcal{H}\varepsilon}^{-1}\mathbf{H}^\top)^L\mathbf{X}_{\mathcal{V}}.$$

Based on Lemma D.3, this equation is a special case of $\hat{\mathbf{X}}_{\mathcal{V}} = \left((1-\alpha)^L\mathbf{W}^L + \alpha\sum_{l=0}^{L-1}(1-\alpha)^l\mathbf{W}^l\right)\mathbf{X}_{\mathcal{V}}$ with $\hat{\mathbf{X}}_{\mathcal{V}} = \mathbf{X}_{\mathcal{V}}^{(L)}$, $\mathbf{W} = \mathbf{D}_{\mathcal{H}^{\mathcal{V}}}^{-1}\mathbf{H}\mathbf{D}_{\mathcal{H}\varepsilon}^{-1}\mathbf{H}^\top$ and $\alpha = 0$.

For an ED-HNN (Wang et al., 2023a) with $L$ feature aggregation functions, devoid non-linear activation and by setting the learnable transformation matrix to the identity matrix, its MP-Module can be reformulated as:
$$\mathbf{X}_{\mathcal{V}}^{(L)} = \left((1-\gamma_E)^L(\mathbf{D}_{\mathcal{H}^{\mathcal{V}}}^{-1}\mathbf{H}\mathbf{D}_{\mathcal{H}\varepsilon}^{-1}\mathbf{H}^\top)^L + \gamma_E\sum_{l=0}^{L-1}(1-\gamma_E)^l(\mathbf{D}_{\mathcal{H}^{\mathcal{V}}}^{-1}\mathbf{H}\mathbf{D}_{\mathcal{H}\varepsilon}^{-1}\mathbf{H}^\top)^l\right)\mathbf{X}_{\mathcal{V}}.$$

Based on Lemma D.3, this equation is a special case of $\hat{\mathbf{X}}_{\mathcal{V}} = \left((1-\alpha)^L \mathbf{W}^L + \alpha \sum_{l=0}^{L-1}(1-\alpha)^l \mathbf{W}^l\right)\mathbf{X}_{\mathcal{V}}$ with $\hat{\mathbf{X}}_{\mathcal{V}} = \mathbf{X}_{\mathcal{V}}^{(L)}$, $\mathbf{W} = \mathbf{D}_{\mathcal{HV}}^{-1}\mathbf{H}\mathbf{D}_{\mathcal{H\varepsilon}}^{-1}\mathbf{H}^{\top}$ and $\alpha = \gamma_E$.

$\square$

# E    PROOF OF PROPOSITION 3.3

We first introduce some concepts about neighbours on hypergraphs and graphs: 1) The $k$-th hop neighbours of a node $v_i$ on a hypergraph are exactly $k$ hyperedges away from $v_i$; and 2) The $k$-hop neighbours of a node $v_i$ on a graph are all nodes with a distance of $k$ or less, while the $k$-th hop neighbours are exactly $k$ edges away from $v_i$. We first introduce some Lemmas.

Based on the definition of clique expansion, we present the following Lemma without a proof:

**Lemma E.1.** *Let $\mathcal{H} = (\mathcal{V}, \mathcal{E}, \mathbf{H})$ be a hypergraph and $\mathcal{G} = (\mathcal{V}, \mathbf{W})$ be its clique expansion with self-loops. For any node $v_i$, let $\mathcal{N}_{\mathcal{H}_{v_i}}^1$ be the node set containing the 1-hop neighbours of $v_i$ on $\mathcal{H}$, and $\mathcal{N}_{\mathcal{G}_{v_i}}^1$ be the node set containing the 1-hop neighbours of $v_i$ on $\mathcal{G}$. Then, $\mathcal{N}_{\mathcal{G}_{v_i}}^1 = \mathcal{N}_{\mathcal{H}_{v_i}}^1 \bigcup\{v_i\}$.*

Moreover, we present the following Lemma about $\mathbf{S}$:

**Lemma E.2.** *Let $\mathcal{H} = (\mathcal{V}, \mathcal{E}, \mathbf{H})$ be a hypergraph and $\mathcal{G} = (\mathcal{V}, \mathbf{W})$ be its clique expansion with self-loops. Assume that $\alpha \in [0, 1)$, and $\mathbf{S} = (1-\alpha)^L\mathbf{W}^L + \alpha\sum_{l=0}^{L-1}(1-\alpha)^l\mathbf{W}^l$. Then, for $i \neq j$, we have $\mathbf{S}_{ij} > 0$ if and only if $v_i$ is a $L$-hop neighbour of $v_j$ on $\mathcal{G}$, and $\mathbf{S}_{ij} = 0$ otherwise.*

*Proof.* According to previous works in graph theory (West et al., 2001), the element $\mathbf{W}_{ij}^l$ represents the number of walks of length $l$ from node $v_i$ to node $v_j$ on the graph $\mathcal{G}$. Since $\mathbf{W}$ includes self-loops, $\mathbf{W}_{ij}^l$ also accounts for paths where nodes can revisit themselves. Hence, for $i \neq j$, we have $\mathbf{W}_{ij}^l > 0$ if and only if $v_i$ is a $l$-hop neighbour of $v_j$ on $\mathcal{G}$, and $\mathbf{S}_{ij} = 0$ otherwise. As a result, for $\mathbf{S} = (1-\alpha)^L\mathbf{W}^L + \alpha\sum_{l=0}^{L-1}(1-\alpha)^l\mathbf{W}^l$ and $i \neq j$, we have $\mathbf{S}_{ij} > 0$ if and only if $v_i$ is a $L$-hop neighbour of $v_j$ on $\mathcal{G}$, and $\mathbf{S}_{ij} = 0$ otherwise. $\square$

On the basis of Lemmas E.1 and E.2, the proof of Proposition 3.3 can be transformed into the proof of the following Lemma:

**Lemma E.3.** *Let $\mathcal{H} = (\mathcal{V}, \mathcal{E}, \mathbf{H})$ be a hypergraph and $\mathcal{G} = (\mathcal{V}, \mathbf{W})$ be its clique expansion with self-loops. For any node $v_i$, let $\mathcal{N}_{\mathcal{H}_{v_i}}^l$ be the node set containing the $l$-hop neighbours of $v_i$ on $\mathcal{H}$, and $\mathcal{N}_{\mathcal{G}_{v_i}}^l$ be the node set containing the $l$-hop neighbours of $v_i$ on $\mathcal{G}$. Then, for $l \in \mathbb{Z}_+$, $\mathcal{N}_{\mathcal{G}_{v_i}}^l = \mathcal{N}_{\mathcal{H}_{v_i}}^l \bigcup\{v_i\}$.*

*Proof.* We prove this Lemma based on the mathematical induction.

Firstly, for the base case with $l = 1$, according to Lemma E.1, this case can be proved by the definition of the clique expansion.

Secondly, for the inductive step, we assume that $\mathcal{N}_{\mathcal{G}_{v_i}}^k = \mathcal{N}_{\mathcal{H}_{v_i}}^k \bigcup\{v_i\}$. Let $\widehat{\mathcal{N}}_{\mathcal{G}_{v_i}}^k$ be the 1-hop neighbours of the $k$-hop neighbours of $v_i$ on the graph $\mathcal{G}$, and $\widehat{\mathcal{N}}_{\mathcal{H}_{v_i}}^k$ be the 1-hop neighbours of the $k$-hop neighbours of $v_i$ on the hypergraph $\mathcal{H}$. Then, we have $\mathcal{N}_{\mathcal{G}_{v_i}}^{k+1} = \widehat{\mathcal{N}}_{\mathcal{G}_{v_i}}^k \bigcup\mathcal{N}_{\mathcal{G}_{v_i}}^k$ and $\mathcal{N}_{\mathcal{H}_{v_i}}^{k+1} = \widehat{\mathcal{N}}_{\mathcal{H}_{v_i}}^k \bigcup\mathcal{N}_{\mathcal{H}_{v_i}}^k$. According to the result in the base case, we have $\mathcal{N}_{\mathcal{H}_{v_i}}^{k+1} = \mathcal{N}_{\mathcal{H}_{v_i}}^{k+1} \bigcup\{v_i\}$.

By induction, for $l \in \mathbb{Z}_+$, $\mathcal{N}_{\mathcal{G}_{v_i}}^l = \mathcal{N}_{\mathcal{H}_{v_i}}^l \bigcup\{v_i\}$. $\square$

# F    DISCUSSION ABOUT EQ. (5)

To recap, for a given hypergraph $\mathcal{H} = \{\mathcal{V}, \mathcal{E}, \mathbf{H}\}$, the edge weight between $v_i$ and $v_j$ in our clique expansion is defined as:

$$\mathbf{W}_{H_{ij}} = \sum_{k=1}^m \frac{\delta(v_i, v_j, e_k)}{\mathbf{D}_{\mathcal{H}\varepsilon_{kk}}},$$

where $\delta(\cdot)$ is a function that returns 1 if $e_k$ connects $v_i$ and $v_j$ and returns 0 otherwise. In this section, we discuss the relationship between this edge weight and the probability of nodes $v_i$ and $v_j$ being in the same category.

We summarise the relationship between this edge weight and the probability of nodes $v_i$ and $v_j$ being in the same category as the following Lemma:

**Lemma F.1.** *Let $p_{i,j}$ denote the probability of $v_i$ and $v_j$ having the same label, $p'_{i,j}$ represent the probability of $v_i$ and $v_j$ being connected by a hyperedge that contains only nodes with the same label, and $\hat{p}_{e_k}$ be the probability of $e_k$ containing nodes with different labels. Then, $p_{i,j}$ is positively correlated with $\mathbf{W}_{H_{ij}}$ if the following conditions hold:*

*a) Hyperedges with higher degrees are more likely to connect nodes with different labels, and $\hat{p}_{e_k} = g_1(\frac{1}{\mathbf{D}_{\mathcal{H}^{\mathcal{E}}_{kk}}})$, where $g_1(\cdot)$ is a function, for any $x, y > 0$, with $g_1(x) \cdot g_1(y) = g_1(x+y)$, $\frac{\mathrm{d}\,g_1}{\mathrm{d}\,x} < 0$, and $g_1(x) \in (0,1)$.*

*b) $p_{i,j}$ is positively correlated with $p'_{i,j}$, namely, $p_{i,j} = g_2(p'_{i,j})$, where $g_2(\cdot)$ is a function for any $x > 0$ with $\frac{\mathrm{d}\,g_2}{\mathrm{d}\,x} > 0$ and $g_2(x) \in (0,1)$.*

*Proof.* According to the pre-defined condition a), we can have:

$$p'_{i,j} = 1 - \prod_{e_k \in \hat{\mathcal{E}}_{v_i,v_j}} \hat{p}_{e_k} = 1 - \prod_{e_k \in \hat{\mathcal{E}}_{v_i,v_j}} g_1(\frac{1}{\mathbf{D}_{\mathcal{H}^{\mathcal{E}}_{kk}}}) = 1 - g_1(\sum_{e_k \in \hat{\mathcal{E}}_{v_i,v_j}} \frac{1}{\mathbf{D}_{\mathcal{H}^{\mathcal{E}}_{kk}}}) = 1 - g_1(\sum_{k=1}^{m} \frac{\delta(v_i,v_j,e_k)}{\mathbf{D}_{\mathcal{H}^{\mathcal{E}}_{kk}}}) = 1 - g_1(\mathbf{W}_{H_{ij}}), \quad (9)$$

where $\hat{\mathcal{E}}_{v_i,v_j}$ is a set containing hyperedges connecting $v_i$ and $v_j$. Based on Eq. (9) and the pre-defined condition b) we can have:

$$p_{i,j} = g_2(p'_{i,j}) = g_2(1 - g_1(\mathbf{W}_{H_{ij}})). \quad (10)$$

The derivative of Eq. (10) with respect to $\mathbf{W}_{H_{ij}}$ is:

$$\frac{\mathrm{d}\,p_{i,j}}{\mathrm{d}\,\mathbf{W}_{H_{ij}}} = \frac{\mathrm{d}\,g_2}{\mathrm{d}\,\mathbf{W}_{H_{ij}}} = \frac{\mathrm{d}\,g_2}{\mathrm{d}\,p'_{i,j}} \cdot \frac{\mathrm{d}\,p'_{i,j}}{\mathrm{d}\,g_1} \cdot \frac{\mathrm{d}\,g_1}{\mathrm{d}\,\mathbf{W}_{H_{ij}}} = -1 \cdot \frac{\mathrm{d}\,g_2}{\mathrm{d}\,p'_{i,j}} \cdot \frac{\mathrm{d}\,g_1}{\mathrm{d}\,\mathbf{W}_{H_{ij}}}.$$

On the basis of our pre-defined conditions a) and b), we have $\frac{\mathrm{d}\,g_2}{\mathrm{d}\,p'_{i,j}} > 0$ and $\frac{\mathrm{d}\,g_1}{\mathrm{d}\,\mathbf{W}_{H_{ij}}} < 0$, thereby we have:

$$\frac{\mathrm{d}\,p_{i,j}}{\mathrm{d}\,\mathbf{W}_{H_{ij}}} = -1 \cdot \frac{\mathrm{d}\,g_2}{\mathrm{d}\,p'_{i,j}} \cdot \frac{\mathrm{d}\,g_1}{\mathrm{d}\,\mathbf{W}_{H_{ij}}} > 0.$$

As a result, $p_{i,j}$ is positively correlated with $\mathbf{W}_{H_{ij}}$. $\qquad\square$

## G PROOF OF PROPOSITION 4.1

Before proving Proposition 4.1, we first present two Lemmas.

**Lemma G.1.** *Let $H_0$ denote the entropy of the information that an HNN utilises by $L$ feature aggregation function $\phi_\Theta(\cdot)$ introduced in Proposition 3.1 for generating features of $v_i$, and $H_2^{v_i^L}$ denote the entropy of information within node $v_i$ and its $L$-hop neighbours on the hypergraph. Then, $H_0^{v_i^L} = H_2^{v_i^L}$.*

*Proof.* Based on the mathematical induction, we start by proving this Using mathematical induction, we first prove this lemma for the case where $\mathcal{N}_{\mathcal{H}_{v_i}}$ includes only the 1-hop neighbours of $v_i$, which applies to all models except the line-based models discussed in Appendix B.

Firstly, for the base case with $L = 1$, the message passing layer for generating the features of $v_i$ can be reformulated as:

$$\mathbf{x}_{v_i}^{(1)} = \phi_\Theta(\mathbf{x}_{v_i}^{(0)}, \mathbf{x}_{v_i}^{(0)}, \oplus_{v_j \in \mathcal{N}_{\mathcal{H}_{v_i}}} \mathbf{x}_{v_j}^{(0)}), \quad (11)$$

where $\mathcal{N}_{\mathcal{H}_{v_i}}$ is a set containing the 1-hop neighbours of $v_i$ on the hypergraph. Hence, the entropy of the input information utilised to generate $\mathbf{x}_{v_i}^{(1)}$ is $H_2^{v_i^1}$, namely, we have $H_0^{v_i^1} = H_2^{v_i^1}$.

Secondly, for the inductive step, we assume that $H_0^{v_i^k} = H_2^{v_i^k}$. Moreover, we have the feature generation function as:

$$\mathbf{x}_{v_i}^{(k+1)} = \phi_\Theta(\mathbf{x}_{v_i}^{(k)}, \mathbf{x}_{v_i}^{(0)}, \oplus_{v_j \in \mathcal{N}_{\mathcal{H}_{v_i}}} \mathbf{x}_{v_j}^{(k)}). \quad (12)$$

Based on the assumption about $L = k$, for $L = k + 1$, $\oplus_{v_j \in \mathcal{N}_{\mathcal{H}_{v_i}}} \mathbf{x}_{v_j}^{(k)}$ contain the information from the $(k + 1)$-hop neighbours of $v_i$ and $\mathbf{x}_{v_i}^{(k)}$ contain the information from the $k$-hop neighbours of $v_i$. Let $\mathcal{N}_{\mathcal{H}_{v_i}}^l$ denote a set containing $l$-hop neighbours of $v_i$. Based on information theory (Kullback, 1997), we have the information entropy of $\mathcal{N}_{\mathcal{H}_{v_i}}^k$ is zero with the information of $\mathcal{N}_{\mathcal{H}_{v_i}}^{k+1}$ is given, as $\mathcal{N}_{\mathcal{H}_{v_i}}^k \subseteq \mathcal{N}_{\mathcal{H}_{v_i}}^{k+1}$. As a result, based on the feature aggregation function introduced in Eq. (12), we have $H_0^{v_i^{k+1}} = H_2^{v_i^{k+1}}$.

By induction, for $L \in \mathbb{Z}_+$, $H_0^{v_i^L} = H_2^{v_i^L}$. Now, we conduct the proof for the line-based models.

Let $H_L^{v_i^L}$ denote the entropy of the information contained by the $L$-hop neighbours of the line-nodes corresponding to $v_i$. According to previous papers in the GNN literature (Wu et al., 2020), the entropy of information used by an $L$-layer GNN to generate the features of line-nodes corresponding to $v_i$ is $H_L^{v_i^L}$. Based on the discussion in Appendix B and Proposition 3.1, we have $H_0^{v_i^L} = H_L^{v_i^L}$. By definition, the line-nodes corresponding to $v_i$ are only connected to each other and the line-nodes corresponding to the 1-hop neighbours of $v_i$. Therefore, we have $H_L^{v_i^L} = H_2^{v_i^L}$. Accordingly, we have $H_0^{v_i^L} = H_2^{v_i^L}$. $\square$

**Lemma G.2.** *Let $H_1$ denote the entropy of the information that an TF-HNN with an $L$-layer TF-MP-Module utilises to generate features of $v_i$, and $H_2^{v_i^L}$ denote the entropy of information within node $v_i$ and its $L$-hop neighbours on the hypergraph. Then, $H_1^{v_i^L} = H_2^{v_i^L}$.*

*Proof.* For the feature generation of $v_i$, an $L$-layer TF-MP-Module can be directly reformulated as:
$$\hat{\mathbf{x}}_{v_i} = f_{l_0}(\mathbf{x}_{v_i}) + f_{l_1}(\oplus_{v_j \in \mathcal{N}_{\mathcal{H}_{v_i}}^L} \mathbf{x}_{v_j}), \tag{13}$$
where $f_{l_0}$ and $f_{l_1}$ are two linear functions, and $\mathcal{N}_{\mathcal{H}_{v_i}}^L$ is a set containing $L$-hop neighbours of $v_i$ on the hypergraph. Hence, we have the entropy of the input information utilised by an $L$-layer TF-MP-Module for generating features of $v_i$ is $H_2^{v_i^L}$, namely, $H_1^{v_i^L} = H_2^{v_i^L}$. $\square$

After having the Lemmas above, we prove Proposition 4.1 as follows:

*Proof.* Based on Lemma G.1, we have $H_0^{v_i^L} = H_2^{v_i^L}$. Moreover, according to Lemma G.2, we have $H_1^{v_i^L} = H_2^{v_i^L}$. As a result, $H_0^{v_i^L} = H_1^{v_i^L} = H_2^{v_i^L}$. $\square$

# H    PROOF OF PROPOSITION 4.2

Before proving Proposition 4.2, we first prove the following Lemma.

**Lemma H.1.** *Let $\mathcal{H} = \{\mathcal{V}, \mathcal{E}, \mathbf{H}\}$ denote a hypergraph, $\mathcal{G} = \{\mathcal{V}, \mathbf{W}_H\}$ be its clique expansion with edge weights computed by Eq. (5), $\mathbf{L}$ be the graph Laplacian matrix of $\mathcal{G}$ computed by a symmetrically normalised and self-loops added $\mathbf{W}_H$, $\mathbf{X}_{\mathcal{V}}$ represent the input node features, and $\alpha \in (0, 1)$. We set $F(\mathbf{X}) = \mathrm{tr}(\mathbf{X}^\top \mathbf{L} \mathbf{X}) + \frac{\alpha}{1-\alpha} \mathrm{tr}[(\mathbf{X} - \mathbf{X}_{\mathcal{V}})^\top (\mathbf{X} - \mathbf{X}_{\mathcal{V}})]$, and $\mathbf{X}^\star$ as the global minimal point for $F(\mathbf{X})$. Then, we have:*
$$\mathbf{X}^\star = \left(\mathbf{I}_n - (1-\alpha)\tilde{\mathbf{D}}_H^{-1/2}\tilde{\mathbf{W}}_H\tilde{\mathbf{D}}_H^{-1/2}\right)^{-1} \alpha \mathbf{X}_{\mathcal{V}}, \tag{14}$$
*where $\tilde{\mathbf{W}}_H = \mathbf{W}_H + \mathbf{I}_n$ and $\tilde{\mathbf{D}}_H \in \mathbb{R}^{n \times n}$ is the diagonal node degree matrix of $\tilde{\mathbf{W}}_H$.*

*Proof.* Taking the partial derivative of $F$ with the respective of $\mathbf{X}$, we have:
$$\frac{\partial F}{\partial \mathbf{X}} = 2\mathbf{L}\mathbf{X} + \frac{2\alpha}{1-\alpha}(\mathbf{X} - \mathbf{X}_{\mathcal{V}}).$$
By setting $\frac{\partial f}{\partial \mathbf{X}} = 0$, we have:
$$\mathbf{X}^\star = (\mathbf{L} + \frac{\alpha}{1-\alpha})^{-1}\frac{\alpha}{1-\alpha}\mathbf{X}_{\mathcal{V}} = \left(\mathbf{I}_n - (1-\alpha)\tilde{\mathbf{D}}_H^{-1/2}\tilde{\mathbf{W}}_H\tilde{\mathbf{D}}_H^{-1/2}\right)^{-1} \alpha \mathbf{X}_{\mathcal{V}},$$

where $\tilde{\mathbf{W}}_H = \mathbf{W}_H + \mathbf{I}_n$ and $\tilde{\mathbf{D}}_H \in \mathbb{R}^{n \times n}$ is the diagonal node degree matrix of $\tilde{\mathbf{W}}_H$.

Since the second-order derivative of $F$ with the respect of $\mathbf{X}$ is:

$$\frac{\partial^2 f}{\partial^2 \mathbf{X}} = 2\mathbf{L} + \frac{2\alpha}{1 - \alpha}\mathbf{I}_n. \tag{15}$$

Given $\mathbf{L} \in \mathbb{R}^{n \times n}$ is a graph Laplacian matrix, which is positive semi-definite, and, for $\alpha \in (0, 1)$, $\frac{2\alpha}{1-\alpha}\mathbf{I}_n \in \mathbb{R}_+^{n \times n}$ is a diagonal matrix, which is positive definite. Accordingly, we have the righthand side of Eq.(15) is a positive definite matrix for $\alpha \in (0, 1)$. As a result, for $\alpha \in (0, 1)$, $\mathbf{X}^\star$ in Eq. (14) is the global minimal point for $F$. $\qquad\square$

With the Lemma above, we present the proof of Proposition 4.2 as follows.

*Proof.* Let $\mathcal{H} = \{\mathcal{V}, \mathcal{E}, \mathbf{H}\}$ denote a hypergraph and $\mathcal{G} = \{\mathcal{V}, \mathbf{W}_H\}$ be its clique expansion with edge weights computed by Eq. (5). According to Eq. (6), the output of an $L$-layer TF-MP-Module can be reformulated as:

$$\hat{\mathbf{X}}_\mathcal{V} = \Big( (1 - \alpha)^L (\tilde{\mathbf{D}}_H^{-1/2}\tilde{\mathbf{W}}_H\tilde{\mathbf{D}}_H^{-1/2})^L + \alpha \sum_{l=0}^{L-1} (1 - \alpha)^l (\tilde{\mathbf{D}}_H^{-1/2}\tilde{\mathbf{W}}_H\tilde{\mathbf{D}}_H^{-1/2})^l \Big) \mathbf{X}_\mathcal{V},$$

where $\tilde{\mathbf{W}}_H = \mathbf{W}_H + \mathbf{I}_n$ and $\tilde{\mathbf{D}}_H \in \mathbb{R}^{n \times n}$ is the diagonal node degree matrix of $\tilde{\mathbf{W}}_H$. When $L \to +\infty$, the left term tends to $\mathbf{0}$ and the right term becomes a convergent geometric series (Meyer & Stewart, 2023). Hence, for $L \to +\infty$, this equation can be further represented as:

$$\hat{\mathbf{X}}_\mathcal{V} = \Big( \mathbf{I}_n - (1 - \alpha)\tilde{\mathbf{D}}_H^{-1/2}\tilde{\mathbf{W}}_H\tilde{\mathbf{D}}_H^{-1/2} \Big)^{-1} \alpha \mathbf{X}_\mathcal{V}.$$

Based on Lemma H.1, we have $F(\hat{\mathbf{X}}_\mathcal{V}) = F_{min}$. $\qquad\square$

# I    DETAILS OF BENCHMARKING DATASETS

Our benchmark datasets consist of existing six datasets (Cora-CA, DBLP-CA, Citeseer and House from (Chien et al., 2022), Congress and Senate from (Wang et al., 2023a)). In co-citation networks (Citeseer), all documents cited by a particular document are connected by a hyperedge. For co-authorship networks (Cora-CA, DBLP-CA), all documents co-authored by a single author are grouped into one hyperedge. The node features in these co-citation and co-authorship networks are represented using bag-of-words models of the corresponding documents, with node labels corresponding to the paper classes. In the House dataset, each node represents a member of the US House of state-of-the-arts, with hyperedges grouping members of the same committee. Node labels denote the political party of the state-of-the-arts. In the Congress dataset, nodes represent US Congresspersons, with hyperedges linking the sponsor and co-sponsors of bills introduced in either the House or the Senate. In the Senate dataset, nodes also represent US Congresspersons, but hyperedges link the sponsor and co-sponsors of Senate bills only. Nodes in both datasets are labeled by political party affiliation.

# J    ADDITIONAL EXPERIMENTAL RESULTS

Table 9: The hyperlink prediction AUC (%). The best result is in **bold font**.

|  | Cora-CA | DBLP-CA | Citeseer | Congress | House | Senate | Avg. Mean |
|---|---|---|---|---|---|---|---|
| AllSetTransformer | $95.51 \pm 0.46$ | $95.77 \pm 1.01$ | $95.98 \pm 0.73$ | $71.56 \pm 0.07$ | $80.28 \pm 3.33$ | $63.56 \pm 1.15$ | 83.78 |
| PhenomNN | $97.60 \pm 0.74$ | $96.11 \pm 0.86$ | $96.13 \pm 0.92$ | $70.68 \pm 0.34$ | $74.57 \pm 3.02$ | $58.46 \pm 1.98$ | 82.26 |
| ED-HNN | $97.61 \pm 0.83$ | $95.71 \pm 2.10$ | $96.04 \pm 0.85$ | $71.48 \pm 0.12$ | $81.09 \pm 3.47$ | $\mathbf{65.96 \pm 1.79}$ | 84.64 |
| Deep-HGNN | $97.00 \pm 0.50$ | $97.60 \pm 0.18$ | $96.45 \pm 0.55$ | $71.34 \pm 0.28$ | $78.84 \pm 6.20$ | $62.06 \pm 3.13$ | 83.88 |
| TF-HNN (ours) | $\mathbf{97.87 \pm 0.65}$ | $\mathbf{97.96 \pm 2.43}$ | $\mathbf{96.95 \pm 0.97}$ | $\mathbf{72.33 \pm 0.14}$ | $\mathbf{82.18 \pm 3.70}$ | $65.04 \pm 0.97$ | **85.39** |

Table 10: The training time (s) in hyperlink prediction. The best result is in **bold font**.

|  | Cora-CA | DBLP-CA | Citeseer | Congress | House | Senate | Avg. Mean |
|---|---|---|---|---|---|---|---|
| AllSetTransformer | $20.82 \pm 5.27$ | $64.07 \pm 5.75$ | $9.18 \pm 2.75$ | $2086.49 \pm 244.37$ | $51.76 \pm 13.03$ | $33.15 \pm 8.26$ | 377.58 |
| PhenomNN | $14.34 \pm 2.72$ | $365.47 \pm 144.07$ | $25.87 \pm 9.60$ | $1029.77 \pm 200.26$ | $130.82 \pm 18.67$ | $30.89 \pm 3.50$ | 266.19 |
| ED-HNN | $8.03 \pm 1.71$ | $65.84 \pm 8.52$ | $11.28 \pm 2.71$ | $899.00 \pm 162.75$ | $75.01 \pm 10.62$ | $53.81 \pm 8.34$ | 185.50 |
| Deep-HGNN | $12.31 \pm 2.72$ | $107.97 \pm 37.50$ | $8.49 \pm 1.68$ | $795.25 \pm 173.54$ | $36.33 \pm 7.74$ | $7.82 \pm 1.32$ | 161.36 |
| TF-HNN (ours) | $\mathbf{3.22 \pm 1.99}$ | $\mathbf{29.36 \pm 6.31}$ | $\mathbf{2.75 \pm 1.26}$ | $\mathbf{501.66 \pm 111.99}$ | $\mathbf{17.83 \pm 4.98}$ | $\mathbf{3.19 \pm 1.31}$ | **93.00** |

Table 11: The node classification accuracy (%). The best result is in **bold font**.

|  | Cora-CA | DBLP-CA | Citeseer | Congress | House | Senate | Avg. Mean |
|---|---|---|---|---|---|---|---|
| HyperGCL | 83.02 ± 1.36 | 91.12 ± 0.28 | 70.70 ± 1.77 | 94.30 ± 0.75 | 65.25 ± 7.93 | 48.76 ± 4.73 | 75.53 |
| HDS$^{ode}$ | 85.60 ± 1.07 | 91.55 ± 0.33 | 73.68 ± 1.99 | 90.58 ± 1.72 | 71.58 ± 1.60 | 59.72 ± 5.60 | 78.79 |
| TF-HNN (ours) | **86.54 ± 1.32** | **91.80 ± 0.30** | **74.82 ± 1.67** | **95.09 ± 0.89** | **76.29 ± 1.99** | **70.42 ± 2.74** | **82.50** |

Table 12: The training time (s) in node classification. The best result is in **bold font**.

|  | Cora-CA | DBLP-CA | Citeseer | Congress | House | Senate | Avg. Mean |
|---|---|---|---|---|---|---|---|
| HyperGCL | 33.94 ± 3.86 | 722.25 ± 137.08 | 7.94 ± 1.72 | 1384.02 ± 160.35 | 108.08 ± 6.38 | 14.67 ± 1.23 | 378.48 |
| HDS$^{ode}$ | 2.05 ± 0.66 | 531.43 ± 144.96 | 49.70 ± 4.39 | 213.87 ± 0.15 | 5.05 ± 0.99 | 72.23 ± 32.53 | 145.72 |
| TF-HNN (ours) | **0.22 ± 0.12** | **4.39 ± 0.45** | **1.12 ± 0.30** | **0.98 ± 0.35** | **1.01 ± 0.51** | **0.19 ± 0.10** | **1.32** |

## K  NEGATIVE SAMPLING FOR HYPERLINK PREDICTION

We adapt the algorithm from (Chen & Liu, 2023) for this purpose. For each (positive) hyperedge $e \in E$, we generate a corresponding negative hyperedge $f$, where $\alpha \times 100\%$ of the nodes in $f$ are from $e$ and the remaining are from $V e$. Denote the negative hyperlink set as $F$. The number $\alpha$ controls the genuineness of the negative hyperlinks, i.e., higher values of $\alpha$ indicate that the negative hyperlinks are closer to the true ones. Additionally, we define $\beta$ to be the number of times of negative sampling which controls the ratio between positive and negative hyperlinks. In practice, we set $\alpha = 0.5$ and $\beta = 5$.

## L  REPRODUCIBILITY STATEMENT

To ensure the reproducibility of our results, we provide detailed hyperparameter settings used in our experiments. The configurations for the TF-HNN model on the hypergraph node classification task and the hyperlink prediction task are listed below.

Table 13: Hyperparameter settings for TF-HNN in Table 3 and Table 4.

| Dataset | TF-HNN Layers | MLP Layers | MLP Hidden Dimension | Learning Rate | Alpha | Dropout | Weight Decay |
|---|---|---|---|---|---|---|---|
| Cora-CA | 2 | 3 | 1024 | 0.001 | 0.3 | 0.7 | 0.0 |
| DBLP-CA | 2 | 3 | 1024 | 0.0006 | 0.15 | 0.7 | 0.0 |
| Citeseer | 8 | 3 | 1024 | 0.001 | 0.65 | 0.9 | 0.0 |
| House | 16 | 3 | 128 | 0.005 | 0.7 | 0.9 | 0.0 |
| Congress | 1 | 3 | 1024 | 0.0001 | 0.05 | 0.8 | 0.0 |
| Senate | 2 | 7 | 512 | 0.005 | 0.6 | 0.5 | 0.0 |
| Trivago | 64 | 3 | 64 | 0.001 | 0.0 | 0.2 | 0.0 |

In practice, we conduct the hyperparameter search based on grid search. For the hypergraph node classification task listed in Table 3, we apply grid search to the following hyperparameters:

• The number of layers of TF-MP-Module: $\{1, 2, 4, 8, 16\}$.

• The $\alpha$ of TF-MP-Module: $\{0.05, 0.1, 0.15, 0.2, 0.25, 0.3, 0.35, 0.4, 0.45, 0.5, 0.55, 0.6, 0.65, 0.7\}$.

• The number of layers of the node classifier: $\{3, 5, 7\}$.

• The hidden dimension of the node classifier: $\{128, 256, 512, 1024\}$.

• The learning rate for node classifier: $\{1 \times 10^{-4}, 2 \times 10^{-4}, 3 \times 10^{-4}, 4 \times 10^{-4}, 5 \times 10^{-4}, 6 \times 10^{-4}, 7 \times 10^{-4}, 8 \times 10^{-4}, 9 \times 10^{-4}, 1 \times 10^{-3}, 5 \times 10^{-3}\}$.

• The dropout rate for node classifier: $\{0.5, 0.6, 0.7, 0.8, 0.9\}$.

For node classification listed in Table 4, we apply grid search to the following hyperparameters:

• The number of layers of TF-MP-Module: $\{1, 4, 16, 64\}$.

• The $\alpha$ of TF-MP-Module: $\{0.0, 0.1, 0.3, 0.5\}$.

• The number of layers of the node classifier: $\{3\}$.

• The hidden dimension of the node classifier: $\{64\}$.

• The learning rate for node classifier: $\{1 \times 10^{-3}, 1 \times 10^{-4}\}$.

• The dropout rate for node classifier: $\{0.0, 0.2, 0.4\}$.

For hyperlink prediction listed in Figure 3, we apply grid search to the following hyperparameters:

Table 14: Hyperparameter settings for TF-HNN in Figure 3 and Table 5.

| Dataset | TF-HNN Layers | Predictor Layers | Predictor Hidden Dimension | Learning Rate | Alpha | Dropout |
|---------|--------------|------------------|----------------------------|---------------|-------|---------|
| Cora-CA | 64 | 3 | 256 | 0.0005 | 0.5 | 0.0 |
| Citeseer | 2 | 3 | 256 | 0.0005 | 0.5 | 0.0 |
| House | 4 | 3 | 128 | 0.0002 | 0.0 | 0.0 |
| Trivago | 1 | 3 | 64 | 0.0001 | 0.5 | 0.0 |

- The number of layers of TF-MP-Module: $\{1, 2, 4, 8, 16, 32, 64\}$.

- The $\alpha$ of TF-MP-Module: $\{0.0, 0.01, 0.05, 0.1, 0.5\}$.

- The number of layers of the hyperedge predictor: $\{3\}$.

- The hidden dimension of the hyperedge predictor: $\{128, 256\}$.

- The learning rate for node classifier: $\{1 \times 10^{-4}, 2 \times 10^{-4}, 3 \times 10^{-4}, 4 \times 10^{-4}, 5 \times 10^{-4}\}$.

- The dropout rate for node classifier: $\{0.0, 0.2, 0.4, 0.6, 0.8\}$.

For hyperlink prediction listed in Table 5, we apply grid search to the following hyperparameters:

- The number of layers of TF-MP-Module: $\{1, 2, 4\}$.

- The $\alpha$ of TF-MP-Module: $\{0.0, 0.1, 0.3, 0.5\}$.

- The number of layers of the hyperedge predictor: $\{3\}$.

- The hidden dimension of the hyperedge predictor: $\{64\}$.

- The learning rate for node classifier: $\{1 \times 10^{-3}, 1 \times 10^{-4}\}$.

- The dropout rate for node classifier: $\{0.0, 0.2\}$.

The experiments on Trivago are highly time-consuming and need large GPU memory. We used a relatively small search space for hyperparameters in ours and the baselines, which are provided below.

Node classification on Trivago of TF-HNN:

- The number of layers of TF-MP-Module: $\{1, 4, 16, 64\}$.

- The $\alpha$ of TF-MP-Module: $\{0.0, 0.1, 0.3, 0.5\}$.

- The number of layers of the node classifier: $\{3\}$.

- The hidden dimension of the node classifier: $\{64\}$.

- The learning rate for node classifier: $\{1 \times 10^{-4}, 1 \times 10^{-3}\}$.

- The dropout rate for node classifier: $\{0.0, 0.2, 0.4\}$.

Node classification on Trivago of ED-HNN:

- The number of layers of ED-HNN: $\{1, 2, 4\}$ (larger values show the out-of-memory (OOM) issue on our server).

- The number of layers of $\hat{\phi}$, $\hat{\rho}$, and $\hat{\varphi}$, MLPs within message passing: $\{0, 1, 2\}$ (larger values show the out-of-memory (OOM) issue on our server).

- The number of layers of the classifier: $\{3\}$.

- The hidden dimension of $\hat{\phi}$, $\hat{\rho}$, and $\hat{\varphi}$: $\{64, 128\}$ (larger values show the out-of-memory (OOM) issue on our server).

- The hidden dimension of the classifier: $\{64\}$.

- The learning rate for the model: $\{1 \times 10^{-4}, 1 \times 10^{-3}\}$.

- The dropout rate for for the model: $\{0.0, 0.2, 0.4\}$.

Node classification on Trivago of Deep-HGNN:

- The number of message passing layers: $\{1, 4, 16, 64\}$.

- The $\alpha$: $\{0, 0.1, 0.3, 0.5\}$.

- The $\lambda$: $\{0.1\}$.

- The message passing hidden dimension: $\{64, 128, 256\}$ (larger values show the out-of-memory (OOM) issue on our server).

- The number of layers of the classifier: $\{3\}$.

- The hidden dimension of the classifier: $\{64\}$.

- The learning rate for the model: $\{1 \times 10^{-4}, 1 \times 10^{-3}\}$.

- The dropout rate for for the model: $\{0.0, 0.2, 0.4\}$.

Hyperlink prediction on Trivago of ours:

- The number of layers of TF-MP-Module: $\{1, 2, 4\}$.

- The $\alpha$ of TF-MP-Module: $\{0.0, 0.1, 0.3, 0.5\}$.

- The number of layers of the predictor: $\{3\}$.

- The hidden dimension of the predictor: $\{64\}$.

- The learning rate for the predictor: $\{1 \times 10^{-4}, 1 \times 10^{-3}\}$.

- The dropout rate for the predictor: $\{0.0, 0.2\}$.

Hyperlink prediction on Trivago of ED-HNN:

- The number of layers of ED-HNN: $\{1, 2, 4\}$.

- The number of layers of $\hat{\phi}$, $\hat{\rho}$, and $\hat{\varphi}$, MLPs within message passing: $\{0, 1, 2\}$.

- The number of layers of the predictor: $\{3\}$.

- The hidden dimension of $\hat{\phi}$, $\hat{\rho}$, and $\hat{\varphi}$: $\{64, 128\}$.

- The hidden dimension of the predictor: $\{64\}$.

- The learning rate for the model: $\{1 \times 10^{-4}, 1 \times 10^{-3}\}$.

- The dropout rate for for the model: $\{0.0, 0.2\}$.

Hyperlink prediction on Trivago of Deep-HGNN:

- The number of message passing layers: $\{1, 2, 4\}$.

- The $\alpha$: $\{0, 0.1, 0.3, 0.5\}$.

- The $\lambda$: $\{0.1\}$.

- The message passing hidden dimension: $\{64, 128, 256\}$.

- The number of layers of the classifier: $\{3\}$.

- The hidden dimension of the classifier: $\{64\}$.

- The learning rate for the model: $\{1 \times 10^{-4}, 1 \times 10^{-3}\}$.

- The dropout rate for for the model: $\{0.0, 0.2\}$.

## M  EFFICIENCY IMPROVEMENT FROM TF-HNN

In this section, we theoretically discuss that the efficiency improvement provided by TF-HNN is inversely correlated with the complexity of the task-specific module. Specifically, we prove a Lemma.

**Lemma M.1.** *Let $M$ denote the training complexity of the task-specific module and, $J$ denote the training complexity of an HNN. Since an TF-HNN only requires the training of the task-specific module, we approximate its training complexity with the function $t_s(M) = M$. Similarly, we approximate the training complexity of an HNN with the function $t_h(M, J) = M + J$. Furthermore,*

*We quantify the efficiency improvement brought by TF-HNN using the ratio $r(M, J) = \frac{t_h(M,J)}{t_s(M)}$, where a larger $r(M, J)$ indicates a greater efficiency improvement provided by the TF-HNN. Then we have $\frac{\partial r}{\partial M} < 0$.*

*Proof.* We can reformulate the $r(M, J)$ as:

$$r(M, J) = \frac{M + J}{M}.$$

Taking the partial derivative of $r$ with the respective of $M$, we have:

$$\frac{\partial r}{\partial M} = -\frac{J}{M^2} < 0.$$

$\square$

## N    ADDITIONAL DISCUSSION AND RESULTS ABOUT APPNP

The APPNP (Gasteiger et al., 2019) has contributed to the development of more efficient GNN models by shifting learnable parameters to an MLP prior to message passing. In this section, we aim to clarify the differences between the APPNP method proposed in [1] and our TF-HNN in two different perspectives.

• **Methodological Design:** The key difference in the model design is that APPNP requires running the message passing process **during model training**, whereas our TF-HNN performs message passing only **during data preprocessing**. According to Eq. (4) in [1], APPNP employs a message-passing block after a multi-layer perceptron (MLP). The MLP uses the given original node features to generate latent features for the nodes, which then serve as inputs for message passing. This design **requires the message-passing block to be executed during each forward propagation in the training phase**, making it incompatible with preprocessing. In contrast, the training-free message-passing module in TF-HNN directly takes the given original node features as inputs, **enabling it to be fully precomputed during the data preprocessing stage**, thereby eliminating the need for computation during training, which further enhances the training efficiency. Notably, Table 2 below demonstrates that TF-HNN is more training efficient than an APPNP with clique expansion. As a result, we argue the APPNP in [1] does not diminish the core novelty of our TF-HNN, which is the first model to decouple the message-passing operation from the training process for hypergraphs.

• **Practical differences:** In Tables 15 and 16, we empirically demonstrate that, for hypergraph-structured data, TF-HNN is both more training efficient and more effective than APPNP with the clique expansion defined in our Eq. (5). For training efficiency, since APPNP requires message-passing computations during each forward propagation step in training, its training time is **at least 3 times longer than that of TF-HNN.** For effectiveness, our results show that TF-HNN, which performs message passing in the original feature space, achieves better performance compared to APPNP, which projects features into a latent space by an MLP before message passing. We hypothesize that this improvement may be attributed to the MLP in APPNP, which may not be able to fully retain information from the original node features during the projection to the latent space.

Besides the two key differences between APPNP and our TF-HNN above, we also hope to emphasize **the unique theoretical contribution** of our paper for hypergraph machine learning. While the mechanism of graph neural networks (GNNs) is well-studied, the foundational component of HNNs is not clearly identified until our work. In Sec. 3.2, we identify the feature aggregation function as the core component of HNNs, and we show that existing HNNs primarily enhance node features by aggregating features from neighbouring nodes. This insight encompasses various HNN models and provides researchers with a deeper understanding of the behaviour of existing HNNs.

As a result, while APPNP represents a significant contribution to the field, it does not diminish the unique value and contributions of TF-HNN for hypergraphs.

Table 15: The node classification accuracy (%) for TF-HNN, and APPNP+CE. The best result is in **bold font**.

| Methods / Datasets | Cora-CA | Citeseer |
|---|---|---|
| TF-HNN | **86.54 ± 1.32** | **74.82 ± 1.67** |
| APPNP+CE | 86.01 ± 1.35 | 74.32 ± 1.57 |

Table 16: The training time (s) for TF-HNN and APPNP+CE corresponding to results in Table 15. The best result is in **bold font**.

| Methods / Datasets | Cora-CA | Citeseer |
|---|---|---|
| TF-HNN | **0.22 ± 0.12** | **1.12 ± 0.30** |
| APPNP+CE | 0.73 ± 0.16 | 4.61 ± 0.88 |

Table 17: Results for baselines with/without learnable parameters in node classification. The best result on each dataset is in **bold font** ($+LP/-LP$ means the model with/without learnable parameters).

| Dataset | AllDeepSets | | UniGNN | | ED-HNN | | Deep-HGNN | |
|---|---|---|---|---|---|---|---|---|
| | $+LP$ | $-LP$ | $+LP$ | $-LP$ | $+LP$ | $-LP$ | $+LP$ | $-LP$ |
| Cora-CA | 81.97 ± 1.50 | **82.53 ± 0.70** | 83.60 ± 1.14 | **84.82 ± 1.56** | 83.97 ± 1.55 | **85.69 ± 0.96** | 84.89 ± 0.88 | **86.22 ± 1.33** |
| Citeseer | 70.83 ± 1.63 | **71.10 ± 2.33** | 73.05 ± 2.21 | **73.96 ± 1.67** | 73.70 ± 1.38 | **74.08 ± 1.51** | 74.07 ± 1.64 | **74.64 ± 1.52** |

## O    RESULTS ABOUT REMOVING LEARNABLE PARAMETERS FROM BASELINES

In Table 17, we present experiments that remove only the learnable parameters from the node feature generation process in the baselines. In this setup, all the baselines studied in Eq. (4a) to Eq. (4d) (AllDeepSets, UniGNN, ED-HNN, Deep-HGNN) directly apply non-learnable message passing to the original node features. These results confirm that removing the learnable parameters and directly applying message passing on the original given node features can enhance the model performance. These results, together with the ablation study on the weight design for $S$ presented in Table 7 of our submission, demonstrate that both the removal of learnable parameters and the design of our $S$, contribute to the effectiveness of our TF-HNN.

## P    ADDITONAL RESULTS ON YELP

In this section, we conduct experiments on a dataset named Yelp from (Chien et al., 2022), and the results are summarized in Table 18. For HGNN, HCHA, HNHN, UniGCNII, AllDeepSets, and AllSetTransformer, we directly use the accuracy reported in (Chien et al., 2022) and record the training time by running the models with the hyperparameters provided in (Chien et al., 2022) on our RTX 3090 GPUs. For ED-HNN and Deep-HGNN, we report both accuracy and training time based on runs using the optimal hyperparameters we identified. Due to out-of-memory issues with PhenomNN on our RTX 3090 GPUs, we did not include it in the table. The results demonstrate that our model is both effective and efficient on this dataset. Specifically, our model achieves a general second-best performance among the baselines, with accuracy only 0.83% lower than AllSetTransformer, while **being 24 times faster.** These findings highlight the effectiveness and efficiency of our TF-HNN.

## Q    THE ASSUMPTION ABOUT THE STRUCTURE OF HYPERGRAPH

In Sec. 2, we assume that the hypergraph does not contain isolated nodes or empty hyperedges to maintain consistency with prior works Huang & Yang (2021); Chien et al. (2022); Kim et al. (2022); Chen et al. (2022) in the literature. This assumption is based on the observation that most existing hypergraph neural networks (HNNs) use node degree or hyperedge degree as denominators during forward propagation. This inherently presumes these values are nonzero—i.e., there are no isolated nodes or empty hyperedges—since division by zero is undefined. Moreover, we emphasize that this assumption does not compromise the practical applicability of our method. Consistent with prior works, we address scenarios where zero degrees occur by assigning a default value of 1 to such cases.

## R    ADDITIONAL DISCUSSION ABOUT THE SUPERIOR PERFORMANCE, ESPECIALLY ON THE LARGE-SCALE HYPERGRAPH

Generaly, we attribute the superior performance of TF-HNN primarily to its more efficient utilization of node information from the training data compared to baseline models.

Baseline models with training-required message-passing modules process node information by first using a trainable module to project node features into a latent space, and then performing trainable

Table 18: Results on Yelp. The best and second-best results are marked with **bold font** and underlined, respectively.

| | HGNN | HCHA | HNHN | UniGCNII | AllDeepSets | AllSetTransformer | ED-HNN | Deep-HGNN | TF-HNN (ours) |
|---|---|---|---|---|---|---|---|---|---|
| Accuracy (%) | 33.04 ± 0.62 | 30.99 ± 0.72 | 31.65 ± 0.44 | 31.70 ± 0.52 | 30.36 ± 1.57 | **36.89 ± 0.51** | 35.03 ± 0.52 | 35.04 ± 2.64 | 36.06 ± 0.32 |
| Time (s) | 326.15 ± 4.55 | 147.54 ± 0.87 | 70.35 ± 0.52 | 108.21 ± 5.87 | 193.43 ± 2.84 | 158.19 ± 0.66 | 225.64 ± 28.52 | 889.14 ± 40.79 | **6.54 ± 0.96** |

message passing within that space. Learning a latent space that preserves the unique characteristics of individual nodes is extremely challenging, especially for large-scale hypergraphs. This difficulty is compounded by computational constraints: both the projection and message-passing operations require computing gradients for backpropagation, which are extremely resource-intensive. To prevent out-of-memory (OOM) errors during experiments on the large-scale hypergraph Trivago, the hidden dimensions of these baseline models need to be limited to a maximum of 256. This dimension constraint, introduced by GPU memory limitations, makes it difficult for the model to fully preserve information that is helpful for classifying a large number of nodes.

In contrast, TF-HNN performs message passing directly on the original node features during the data pre-processing stage without requiring any trainable parameters. This allows TF-HNN to preserve node-specific information without incurring the expensive memory requirements associated with training-required models. Notably, our experimental results align with previous observations in the Graph Neural Network (GNN) literature. For example, training-free models like SIGN (Frasca et al., 2020) have been shown to outperform training-required models like GCN (Kipf & Welling, 2017) on large-scale graphs such as the Protein-Protein Interaction (PPI) network by a significant margin.

Table 19: Hyperparameter search time for ED-HNN and TF-HNN in node classification on Cora-CA.

| | ED-HNN | TF-HNN | $t_{TF}/t_{ED}$ |
|---|---|---|---|
| Runtime for Completing the Grid Search (hours) | 48.95 | 2.74 | 0.06 |

Table 20: Hyperparameter search time for Deep-HGNN, ED-HNN and our TF-HNN in the node classification task on Trivago (The search time of ED-HNN and TF-HNN on Trivago is shorter than the one on Cora, as the search space used on Trivago is much smaller than the one used on Cora).

| | Deep-HGNN | ED-HNN | TF-HNN |
|---|---|---|---|
| Hyperparameter Searching Time (hours) | 59.61 | 30.42 | 1.04 |

## S HYPERPARAMETER SEARCH

Our hyperparameter tuning process strictly adheres to standard practices using the validation set to determine optimal settings. Following the codes of prior works (Chien et al., 2022; Wang et al., 2023b;a), our code records the best performance on the validation set, the training epoch that achieved the best performance on the validation set and the performance of this recorded training epoch on the test set. The hyperparameters yielding the best results on the validation set were adopted as the final settings for our model. Furthermore, using the node classification task on DBLP-CA as a case study, we conducted a detailed hyperparameter sensitivity analysis. The heatmaps in Figure 5 illustrate validation and test performance across various hyperparameter configurations. The hyperparameters yielding the best results on the validation set highlighted with a green rectangle were adopted as the final settings for our model. Based on these results, we derive three key insights for hyperparameter search in TF-HNN: 1) The model exhibits minimal sensitivity (performance variations within $1\%$) to the following combinations, suggesting they can be tuned independently: MLP Hidden Dimension & Dropout, MLP Hidden Dimension & TF-HNN Layers, Learning Rate & Dropout, Learning Rate & MLP Hidden Dimension, Learning Rate & TF-HNN Layers, TF-HNN Layers & MLP Layers, and TF-HNN Layers & Dropout; 2) The MLP performs well with relatively few layers, reducing the need for deeper architectures; and 3) The following combinations require more careful tuning due to their greater impact on performance: Alpha & MLP Layers, Alpha & Dropout, Alpha & MLP Hidden Dimension, Alpha & Learning Rate, and Alpha & TF-HNN Layers.

Generally, the efficient search for the optimal hyperparameters is challenging within the hypergraph machine learning literature. However, with the fast training speed of TF-HNN, our hyperparameter searching process is still efficient compared with previous methods. Table 19 shows the hyperparameter searching time used for TF-HNN and ED-HNN (Wang et al., 2023a) in node classification on the Cora-CA dataset in our 8 RTX 3090 GPUs server. For TF-HNN, we run the hyperparameter

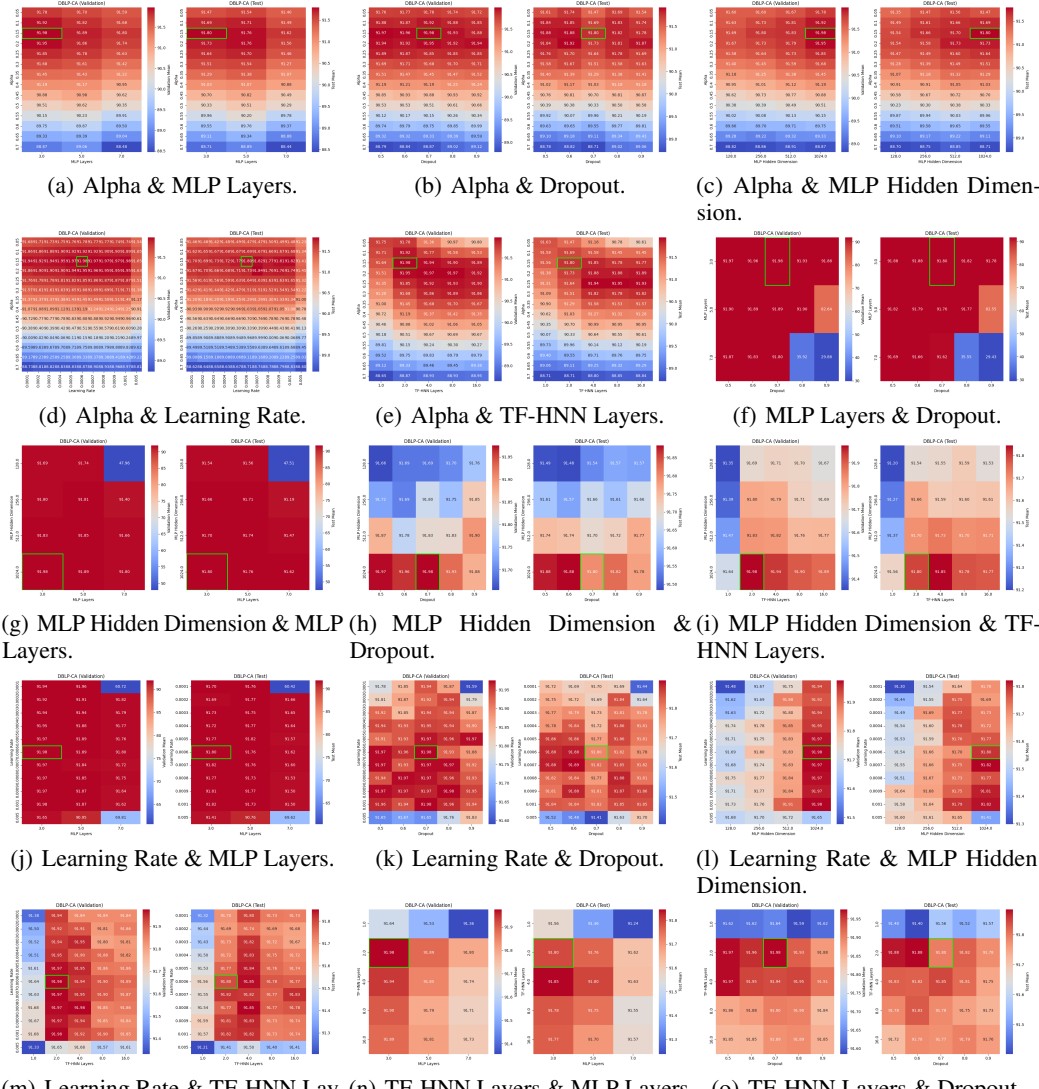

(a) Alpha & MLP Layers.  (b) Alpha & Dropout.  (c) Alpha & MLP Hidden Dimension.

(d) Alpha & Learning Rate.  (e) Alpha & TF-HNN Layers.  (f) MLP Layers & Dropout.

(g) MLP Hidden Dimension & MLP Layers.  (h) MLP Hidden Dimension & Dropout.  (i) MLP Hidden Dimension & TF-HNN Layers.

(j) Learning Rate & MLP Layers.  (k) Learning Rate & Dropout.  (l) Learning Rate & MLP Hidden Dimension.

(m) Learning Rate & TF-HNN Layers.  (n) TF-HNN Layers & MLP Layers.  (o) TF-HNN Layers & Dropout.

Figure 5: Heatmaps of validation and test performance under varying hyperparameter combinations.

combinations mentioned in Appendix L, and for ED-HNN, we run the hyperparameter combination mentioned in Appendix F.3 of their paper. Specifically, according to the discussion and table 6 in Appendix F.3 of Wang et al. (2023a), ED-HNN does grid search for the following hyperparameters:

- The number of layers of ED-HNN, whose search space is $\{1, 2, 4, 6, 8\}$.

- The number of layers of an MLP projector, $\hat{\phi}$, whose search space is $\{0, 1, 2, 4, 6, 8\}$.

- The number of layers of an MLP projector, $\hat{\rho}$, whose search space is $\{0, 1, 2, 4, 6, 8\}$.

- The number of layers of an MLP projector, $\hat{\varphi}$, whose search space is $\{0, 1, 2, 4, 6, 8\}$.

- The number of layers of the node classifier, whose search space is $\{1, 2, 4, 6, 8\}$.

- The hidden dimension of $\hat{\phi}$, $\hat{\varphi}$, and $\hat{\rho}$, whose search space is $\{96, 128, 256, 512\}$.

- The hidden dimension of the node classifier, whose search space is $\{96, 128, 256, 512\}$.

Based on the results presented in Table 19, our hyperparameter search process requires only about 6% of the time required by ED-HNN, demonstrating its efficiency. We treat develop a more efficient

hyperparameter search method for hypergraph neural networks based on the methods like Bayesian Optimisation (Victoria & Maragatham, 2021; Wu et al., 2019b; Nguyen, 2019) as a future work.

# T  DISCUSSION ABOUT PROPOSITION 3.2

We summarise the connection between the clique expansion mentioned in Proposition 3.2 and the clique expansion defined in Eq (5) as the following Lemmas:

**Lemma T.1.** *For both the $W$ of the clique expansion mentioned in Proposition 3.2 and the one defined in Eq (5), we have $W_{ij} > 0$ if and only if $v_i$ and $v_j$ are connected on the given hypergraph and $W_{ij} = 0$ otherwise.*

*Proof.* Here we first prove this lemma for the clique expansion mentioned in Proposition 3.2.

Based on the definition of clique expansion and Lemmas D.1, D.2, D.3, we can directly conclude that, for the weight matrix $W$ of the clique expansion mentioned in Proposition 3.2, $W_{ij} > 0$ if and only if $v_i$ and $v_j$ are connected on the given hypergraph and $W_{ij} = 0$ otherwise.

For the $W$ defined in Eq (5), we have:

$$W_{ij} = \sum_{k=1}^{m} \frac{\delta(v_i, v_j, e_k)}{\mathbf{D}_{\mathcal{H}_{kk}^{\mathcal{E}}}},$$

where $\delta(\cdot)$ is a function that returns 1 if $e_k$ connects $v_i$ and $v_j$ and returns 0 otherwise. Since $\frac{1}{\mathbf{D}_{\mathcal{H}_{kk}^{\mathcal{E}}}} > 0$, we have $W_{ij} = 0$ if and only if, $\forall k \in [1, 2, 3, \cdots, m], \delta(v_i, v_j, e_k) = 0$, namely, $v_i$ and $v_j$ are not connected on the hypergraph. Further, we can have $W_{ij} > 0$ if and only if there exist a $k \in [1, 2, 3, \cdots, m]$ to make $\delta(v_i, v_j, e_k) = 1$, namely, $v_i$ and $v_j$ are connected on the given hypergraph.

$\square$

**Lemma T.2.** *Let $E = \{e_1, e_2, \cdots, e_K\}$ denote the set of hyperedges connecting $v_i$ and $v_j$. Then, for both the clique expansion mentioned in Proposition 3.2 and the clique expansion defined in Eq (5), we have $\frac{\mathrm{d} W_{ij}}{\mathrm{d} |e_k|} < 0, \ \forall e_k \in E$, where $|e_k|$ represents the size of $e_k$.*

*Proof.* Here we first prove this lemma for the clique expansion mentioned in Proposition 3.2.

Based on the discussion in Sec. D, for UniGNN (Huang & Yang, 2021), we have

$$W_{ij} = (1 - \gamma_U) \sum_{k=1}^{m} \frac{\mathbf{H}_{ik}\mathbf{H}_{jk}}{\mathbf{D}_{\mathcal{H}_{ii}^{\mathcal{V}}}^{1/2} \mathbf{D}_{\mathcal{H}_{kk}^{\mathcal{E}}} \tilde{\mathbf{D}}_{\mathcal{H}_{kk}^{\mathcal{E}}}^{1/2}} = \sum_{k=1}^{m} \frac{(1 - \gamma_U) \sum_{k'=1}^{n} \mathbf{H}_{k'k}\mathbf{D}_{\mathcal{H}_{k'k'}^{\mathcal{V}}}}{\mathbf{D}_{\mathcal{H}_{ii}^{\mathcal{V}}}^{1/2}} \frac{\mathbf{H}_{ik}\mathbf{H}_{jk}}{\mathbf{D}_{\mathcal{H}_{kk}^{\mathcal{E}}}^{1/2}} = \sum_{e_k \in E} \frac{\omega_k}{|e_k|^{1/2}},$$

where $\omega_k = \frac{(1 - \gamma_U)s_k}{\mathbf{D}_{\mathcal{H}_{ii}^{\mathcal{V}}}^{1/2}}$ and $s_k$ is the sum of degrees of nodes with $e_k$. Since $\omega_k > 0$ and the design of UniGNN only considers the positive square root, we have:

$$\frac{\mathrm{d} W_{ij}}{\mathrm{d} |e_k|} = -\frac{\omega_k}{2|e_k|^{3/2}} < 0.$$

Based on the discussion in Sec. D, for Deep-HGNN (Chen et al., 2022), we have:

$$W_{ij} = (1 - \gamma_D) \sum_{k=1}^{m} \frac{\mathbf{H}_{ik}\mathbf{H}_{jk}}{\mathbf{D}_{\mathcal{H}_{ii}^{\mathcal{V}}}^{1/2} \mathbf{D}_{\mathcal{H}_{jj}^{\mathcal{V}}}^{1/2} \mathbf{D}_{\mathcal{H}_{kk}^{\mathcal{E}}}} = \frac{(1 - \gamma_D)}{\mathbf{D}_{\mathcal{H}_{ii}^{\mathcal{V}}}^{1/2} \mathbf{D}_{\mathcal{H}_{jj}^{\mathcal{V}}}^{1/2}} \sum_{k=1}^{m} \frac{\mathbf{H}_{ik}\mathbf{H}_{jk}}{\mathbf{D}_{\mathcal{H}_{kk}^{\mathcal{E}}}} = \omega \sum_{e_k \in E} \frac{1}{|e_k|},$$

where $\omega = \frac{(1 - \gamma_D)}{\mathbf{D}_{\mathcal{H}_{ii}^{\mathcal{V}}}^{1/2} \mathbf{D}_{\mathcal{H}_{jj}^{\mathcal{V}}}^{1/2}}$. Since $\omega > 0$, we have:

$$\frac{\mathrm{d} W_{ij}}{\mathrm{d} |e_k|} = -\frac{\omega}{|e_k|^2} < 0.$$

Based on the discussion in Sec. D, for AllDeepSets (Chien et al., 2022) and ED-HNN (Wang et al., 2023a), we have:

$$W_{ij} = \sum_{k=1}^{m} \frac{\mathbf{H}_{ik}\mathbf{H}_{jk}}{\mathbf{D}_{\mathcal{H}_{ii}^{\mathcal{V}}} \mathbf{D}_{\mathcal{H}_{kk}^{\mathcal{E}}}} = \frac{1}{\mathbf{D}_{\mathcal{H}_{ii}^{\mathcal{V}}}} \sum_{k=1}^{m} \frac{\mathbf{H}_{ik}\mathbf{H}_{jk}}{\mathbf{D}_{\mathcal{H}_{kk}^{\mathcal{E}}}} = \omega \sum_{e_k \in E} \frac{1}{|e_k|},$$

where $\omega = \frac{1}{\mathbf{D}_{\mathcal{H}_{ii}^{\mathcal{V}}}}$. Since $\omega > 0$, we have:

$$\frac{\mathrm{d}\,W_{ij}}{\mathrm{d}\,|e_k|} = -\frac{\omega}{|e_k|^2} < 0.$$

Finally, we demonstrate the proof for the clique expansion defined in Eq (5):

$$W_{ij} = \sum_{k=1}^{m} \frac{\delta(v_i, v_j, e_k)}{\mathbf{D}_{\mathcal{H}_{kk}^{\varepsilon}}} = \sum_{e_k \in E} \frac{1}{|e_k|}.$$

Then, we have:

$$\frac{\mathrm{d}\,W_{ij}}{\mathrm{d}\,|e_k|} = -\frac{1}{|e_k|^2} < 0.$$

$\square$

## U    DISCUSSION ABOUT PROPOSITION 4.1

In Proposition 4.1, the "information" refers to the contextual information embedded in the node features, while "information entropy" denotes the entropy of this contextual information. Typically, node features are generated based on specific contextual information. For example, in co-citation datasets where nodes represent research papers, the features of a node are often derived from keywords or sentences in the paper abstract. Consequently, the contextual information embedded in such node features corresponds to these keywords or sentences. According to prior works in Linguistics (Shannon, 1951; Genzel & Charniak, 2002), the information entropy of these keywords or sentences can be computed using the Shannon entropy formula: $H(X) = -\sum_{x \in X} p(x) \log p(x)$, represents the keywords or sentences, $x$ denotes a token within $X$, and $p(x)$ can be defined using various methods, such as an n-gram probabilistic model mentioned in (Genzel & Charniak, 2002).

While we did not compute the exact Shannon entropy here, we use "entropy" in a conceptual way to refer to the amount of information helpful for the downstream task. From this perspective, the key takeaway from the proposition is to highlight that both HNN with an L-layer feature aggregation function and TF-HNN with an L-layer TF-MP-Module can leverage the hypergraph structure to enhance the features of each node by aggregating features containing contextual information from its L-hop neighbouring nodes. For instance, in a co-citation hypergraph, if within L-hop a node is connected to several nodes with features containing contextual information about machine learning and others with features related to biology, the feature aggregation process will incorporate both machine learning and biology related features into the node's features.

## V    ADDITONAL RESULTS ABOUT THE PREPROCESSING TIME

Table 21: The training time (s) and preprocessing time (s) for TF-HNN in node classification.

|  | Cora-CA | DBLP-CA | Citeseer | Congress | House | Senate | Avg. Mean |
|---|---|---|---|---|---|---|---|
| Training | $0.22 \pm 0.12$ | $4.39 \pm 0.45$ | $1.12 \pm 0.30$ | $0.98 \pm 0.35$ | $1.01 \pm 0.51$ | $0.19 \pm 0.10$ | 1.32 |
| Preprocessing | $0.0012 \pm 0.0002$ | $0.0401 \pm 0.0011$ | $0.0220 \pm 0.0012$ | $0.0016 \pm 0.0001$ | $0.0080 \pm 0.0001$ | $0.0009 \pm 0.0001$ | 0.012 |

Let $m$ be the number of edges of the clique expansion used in our TF-MP-Module, then **the theoretical complexity of this module is $O(m)$.** Moreover, unlike the training-required message-passing operators used in existing HNNs, which must be computed during forward propagation in each training epoch and require gradient descent computations for backpropagation, our training-free message-passing operator only needs to be computed once during a single forward propagation in preprocessing. Thus, in practice, the training-free message-passing operator is quite fast. We summarise the runtime in Table 21. These results indicate that the average preprocessing time of our training-free message-passing operator **is about 1% of the average training time for our TF-HNN.**

