# OpenReview forum: "Training-Free Message Passing for Learning on Hypergraphs"
_ICLR.cc/2025/Conference — ICLR 2025 Poster_

### Official Review · Reviewer_9Q7r · 2024-11-03

**Soundness:** 3
**Presentation:** 2
**Contribution:** 3
**Rating:** 8
**Confidence:** 4

**Summary:**

The authors propose a scalable hypergraph neural network (HNN) that reduces the computational cost associated with message passing at each training epoch by decoupling the message passing step.

To achieve this, they (1) establish a general framework that explains popular HNN models, and (2) simplify this framework by removing learnable components and non-linearity, resulting in a single linear operator.

They demonstrate the effectiveness of their approach on both small- and large-scale hypergraphs, showing improvements over several existing HNN models.

**Strengths:**

- S1. Given that real-world hypergraphs are often large, scalable HNNs are necessary in practice.
- S2. Although the model design is quite trivial, the theoretical motivation (i.e., building a general unified framework and simplifying it) of the model design is systematic and interesting.
- S3. Experiments are comprehensive.

**Weaknesses:**

***Major comments.***

- ***W1. Regarding Proposition 3.2.*** My understanding is that the key idea is the existence of a clique expansion that satisfies the property outlined in Proposition 3.2.
In practice, however, the authors use a fixed clique expansion as described in Equation 5.
To what extent does this chosen clique expansion align with the one referenced in Proposition 3.2?
While the exact formulations may differ, it would be helpful to know if the high-level characteristics of these clique-expanded graphs are similar.
This is essential, in my view, as it clarifies whether the theoretical analysis effectively supports the proposed method.

- ***W2. Regarding Proposition 4.1.*** Could you clarify what is meant by the "entropy of information"? Does this refer to mutual information between features and node labels? Further elaboration on this point would help in understanding the key takeaway from this proposition.

- ***W3. Regarding the Initial Message Passing Operator Computation Complexity*** Although the message passing operator incurs a one-time computation cost, the time required for this process should be reported. If this initial computation is substantial and exceeds the typical training time of existing HNNs, it could limit the practical efficiency of the proposed method.

***Minor comments.***
- In Lines 52-53, the text mentions that $n^{k}$ memory is required. While this is accurate for dense tensor formats, typical tensor representations are stored in a sparse format, and sparse operations are well-supported in modern deep-learning libraries. Thus, storing a dense incidence tensor is generally not necessary in practice. It may be helpful to revise this part to reflect real-world scenarios.
- In Lines 79-80, the period "." is missing.
- Please provide clarification on the source of the datasets used.

Please let me know if I have misunderstood any parts. Thank you.

**Questions:**

Refer to the Weakness section.

---

> ### Author Response · Authors · 2024-11-21
> **Response by Authors**
>
> Thank you so much for your thoughtful comments. We have provided detailed, point-by-point responses to your comments below. Additionally, we remain open to further discussions that could help us improve the quality of our paper.
>
> ---
> **_C1. About Proposition 3.2._**
>
> Thank you for the comment. We apologize for any confusion it may have caused. First, we would like to further clarify that **the key takeaway** from Proposition 3.2 is that, through our simplification, each of the selected models can be unified under a general formula involving the adjacency matrix of a clique expansion. Then, we design our TF-MP-Module based on this unified formula. To enhance the clarity of Proposition 3.2, we have revised it as follows. (Note: Due to limitations of the OpenReview system in supporting mathematical representations, we have adjusted some notations. Therefore, the following version differs slightly from the one in our paper.)
>
> _Let_ $X$ _be input node features,_ $X'$ _be the output of the MP-Module of an UniGCNII/AllDeepSets/ED-HNN/Deep-HGNN with $L$ MP layers, and $\alpha \in [0,1)$. Assume that learnable parameters and the non-linearity are removed from the module. For each model, given a hypergraph, there exists a clique expansion $G$={$V,W$} to unify its output as the following formula_ $X'=((1-\alpha)^{L}W^{L}+\alpha\sum_{l=0}^{L-1}(1-\alpha)^{l}W^{l})X.$
>
> With this clarification, we emphasize that the clique expansion used in our TF-HNN shares two key properties with the clique expansions underlying the four schemes in Proposition 3.2, hence connecting our theory and practice:
>
> $\bullet$ **They have the same connectivity pattern.** Specifically, for a given hypergraph, nodes $v_i$ and $v_j$ are connected in the expansion if and only if they are connected on the hypergraph. Mathematically, for the $W$ in both, we have $W_{ij}>0$ if and only if $v_i$ and $v_j$ are connected on the given hypergraph and $W_{ij}=0$ otherwise.
>
> $\bullet$ **Their edge weights exhibit the same general pattern.** Specifically, in both cases, the weights of edges connecting any nodes $v_i$ and $v_j$ are negatively correlated with the sizes of the hyperedges connecting $v_i$ and $v_j$. To demonstrate this mathematically, let $E$ = {$e_1, e_2, \cdots, e_K$} denote the set of hyperedges connecting $v_i$ and $v_j$. Then, for both our clique expansion and the one mentioned in Proposition 3.2, we have $\frac{d W_{ij}}{d |e_k|} < 0$, $\forall e_k\in E$, where $|e_k|$ represents the size of $e_k$.
>
> In the latest version of our submission, we have added a detailed discussion in **Section T** of the appendix and have also refined Section 3.3 in the main manuscript accordingly.

---

> ### Author Response · Authors · 2024-11-21
> **Response by Authors**
>
> **_C2. About Proposition 4.1._**
>
> Thank you for your comment, and we apologize for the confusion caused.
>
> Regarding the meaning of information entropy within Proposition 4.1, **the "information" refers to the contextual information embedded in the node features, while "information entropy" denotes the entropy of this contextual information.** Typically, node features are generated based on specific contextual information. For example, in co-citation datasets where nodes represent research papers, the features of a node are often derived from keywords or sentences in the paper abstract. Consequently, the contextual information embedded in such node features corresponds to these keywords or sentences. According to prior works in Linguistics [1,2], the information entropy of these keywords or sentences can be computed using the Shannon entropy formula: $H(X) = -\sum_{x\in X}p(x)\log p(x)$, represents the keywords or sentences, $x$ denotes a token within $X$, and $p(x)$ can be defined using various methods, such as an n-gram probabilistic model mentioned in [1].
>
> While we did not compute the exact Shannon entropy here, we use ``entropy'' in a conceptual way to refer to the amount of information helpful for the downstream task (we would be happy to rephrase if you suggested so). From this perspective, **the key takeaway** from the proposition is to highlight that both HNN with an L-layer feature aggregation function and TF-HNN with an L-layer TF-MP-Module can leverage the hypergraph structure to enhance the features of each node by aggregating features containing contextual information from its L-hop neighbouring nodes. For instance, in a co-citation hypergraph, if within L-hop a node is connected to several nodes with features containing contextual information about machine learning and others with features related to biology, the feature aggregation process will incorporate both machine learning and biology related features into the node's features.
>
> We have refined the discussion of Proposition 4.1 and included a detailed explanation in **Section U** of the appendix in the latest version of our submission.
>
> [1] Genzel, Dmitriy, and Eugene Charniak. "Entropy rate constancy in text." A. 2002.
>
> [2] Shannon, Claude E. "Prediction and entropy of printed English." Bell system technical journal 30.1 (1951): 50-64.
>
> ---
> **_C3. About the Initial Message Passing Operator Computation Complexity._**
>
> Thank you for this comment. Let $m$ be the number of edges of the clique expansion used in our TF-MP-Module, then **the theoretical complexity of this module is $O(m)$.**  Moreover, unlike the training-required message-passing operators used in existing HNNs, which must be computed during forward propagation in each training epoch and require gradient descent computations for backpropagation, our training-free message-passing operator only needs to be computed once during a single forward propagation in preprocessing. Therefore, in practice, this training-free message-passing operator is quite fast. We have summarised the runtime in Table 1 below. These results indicate that the average preprocessing time of our training-free message-passing operator **is only about 1\% of the average training time for our TF-HNN.** In the latest version of our submission, we have included this discussion in **Section V** of the appendix and referenced this section in Section 6.2 of the main manuscript.
>
> #### Table 1: The training time (s) and preprocessing time (s) for TF-HNN in node classification.
>
> |               | Cora-CA       | DBLP-CA       | Citeseer      | Congress      | House         | Senate        | Avg. Mean   |
> |---------------|---------------|---------------|---------------|---------------|---------------|---------------|-------------|
> | **Training**  | 0.22 ± 0.12   | 4.39 ± 0.45   | 1.12 ± 0.30   | 0.98 ± 0.35   | 1.01 ± 0.51   | 0.19 ± 0.10   | 1.32        |
> | **Preprocessing** | 0.0012 ± 0.0002 | 0.0401 ± 0.0011 | 0.0220 ± 0.0012 | 0.0016 ± 0.0001 | 0.0080 ± 0.0001 | 0.0009 ± 0.0001 | 0.012        |
>
> ---
> **_C4. About using incidence tensor to process hypergraphs._**
>
> Thank you for raising this comment. Indeed, high-dimensional tensors can be stored in a sparse format. However, the practical implementation of incidence tensor-based methods remains challenging due to the requirements of doing multiplication with high-dimensional tensors. Specifically, PyTorch’s sparse tensor functionality supports multiplication only between two 2D sparse tensors, which restricts its direct application to incidence tensors. To clarify this, we have revised **Section 1** in the latest version of our submission accordingly.
>
> ---
> **_C5. The source of the datasets used._**
>
> Thank you for this comment, in the latest version of the submission, we have added the references for our datasets in **Section 6.1**.

---

> ### Comment · Reviewer_9Q7r · 2024-11-22
> **Thank you for your responses**
>
> I appreciate the authors efforts on addressing my concerns. Most of my concerns have been addressed, and therefore, I increase my score $6 \rightarrow 8$.

---

> ### Author Response · Authors · 2024-11-24
> **Thank you again for your valuable feedback**
>
> Dear Reviewer 9Q7r,
>
> Thank you so much for your encouraging response and for considering our efforts to address your concerns!

---

### Official Review · Reviewer_bexR · 2024-11-03

**Soundness:** 2
**Presentation:** 3
**Contribution:** 3
**Rating:** 5
**Confidence:** 3

**Summary:**

This paper introduces an efficient hypergraph learning scheme that performs message passing prior to learning neural network parameters. The proposed framework, called TF-HNN, includes a TF-MP-Module in which training-free message passing is performed. Using the updated features from the TF-MP-Module, TF-HNN learns an MLP model without a heavy computational burden. Despite its high learning efficiency, TF-HNN demonstrates either superior or competitive performance in hypergraph learning tasks. Theoretical analysis supports the design of the proposed method.

**Strengths:**

1. The design of the proposed method is very interesting.
2. The proposed method is both highly efficient and effective.
3. The paper is clearly written and well-organized, making the research easy to follow.

**Weaknesses:**

It appears that there is an issue regarding the hyperparameters. The combinations of hyperparameters used in the experiments shown in Table 13 and Table 14 are quite diverse. For example, the value of alpha ranges from 0.05, 0.15, 0.3, 0.6, 0.65, to 0.7. The learning rate also varies, with values like 0.0006, 0.0001, 0.005, 0.001, and 0.0002. What method was used for hyperparameter search? Additionally, upon reviewing the attached anonymous GitHub link, it appears that the optimal hyperparameters were selected based on performance on the test set rather than the validation set. Were the hyperparameters selected in a fair manner? An analysis of hyperparameter sensitivity should be added.

**Questions:**

Isn’t too much time being spent on hyperparameter search due to the extensive hyperparameter search range?

If the hyperparameters were indeed selected based on the validation set, could you demonstrate this by providing heatmaps of the hyperparameters across the various validation and test sets used in the experiments?

**minor comments**
- line 209: shonw -> shown
- line 144: Ortega et al. (2018) -> (Ortega et al. (2018))

---

> ### Author Response · Authors · 2024-11-21
> **Response by Authors**
>
> Thank you so much for your thoughtful comments. Please find our point-by-point responses below. If our responses address your concerns, we would greatly appreciate it if you could consider raising your score. Additionally, we remain open to further discussions that could help us improve the quality of our paper.
>
> ---
>
> **_C1. Data used for hyperparameter search._**
>
> Thank you for your question, and we apologize for the confusion caused. We would like to clarify that our hyperparameter tuning process strictly adheres to standard practices by **using the validation set, not the test set, to determine optimal settings.** Following the codes of prior works [1-3], our code records the best performance on the validation set, the training epoch that achieved the best performance on the validation set and the performance of this recorded training epoch on the test set. The hyperparameters yielding the best results on the validation set were adopted as the final settings for our model. In the latest version of our submission, we have added these details to Appendix S and referenced this appendix section in Section 6.1 of the main manuscript. Moreover, in response to the reviewer's request, **Figure 5 of Appendix S also presents hyperparameter heatmaps for both the test and validation sets.** The best result on the validation set, along with its corresponding test set result, has been highlighted with green rectangles for clarity.
>
> [1] Wang, Peihao, et al. "Equivariant Hypergraph Diffusion Neural Operators." The Eleventh International Conference on Learning Representations.
>
> [2] Chien, Eli, et al. "You are AllSet: A Multiset Function Framework for Hypergraph Neural Networks." International Conference on Learning Representations.
>
> [3] Wang, Yuxin, et al. "From hypergraph energy functions to hypergraph neural networks." International Conference on Machine Learning. PMLR, 2023.
>
> ---
>
> **_C2. Method used for hyperparameter search._**
>
> Thank you for this question. In practice, we conduct the hyperparameter search based on grid search. For the hypergraph node classification task listed in **Table 13**, we apply grid search to the following hyperparameters:
>
> $\bullet$ The number of layers of TF-MP-Module, whose search space is {$1, 2, 4, 8, 16$}.
>
> $\bullet$ The $\alpha$ of TF-MP-Module, whose search space is {$0.05, 0.1, 0.15, 0.2, 0.25, 0.3, 0.35, 0.4, 0.45, 0.5, 0.55, 0.6, 0.65, 0.7$}.
>
> $\bullet$ The number of layers of the node classifier, whose search space is {$3, 5, 7$}.
>
> $\bullet$ The hidden dimension of the node classifier, whose search space is {$128, 256, 512, 1024$}.
>
> $\bullet$ The learning rate for node classifier, whose search space is {$1\times10^{-4}, 2\times10^{-4}, 3\times10^{-4}, 4\times10^{-4}, 5\times10^{-4}, 6\times10^{-4}, 7\times10^{-4}, 8\times10^{-4}, 9\times10^{-4}, 1\times10^{-3}, 5\times10^{-3}$}.
>
> $\bullet$ The dropout rate for node classifier, whose search space is {$0.5, 0.6, 0.7, 0.8, 0.9$}.
>
> For the hyperlink prediction task listed in **Table 14**, we apply grid search to the following hyperparameters:
>
> $\bullet$ The number of layers of TF-MP-Module, whose search space is {$1, 2, 4, 8, 16, 32, 64$}.
>
> $\bullet$ The $\alpha$ of TF-MP-Module, whose search space is {$0.0, 0.01, 0.05, 0.1, 0.5$}.
>
> $\bullet$ The number of layers of the hyperedge predictor, whose search space is {$3$}.
>
> $\bullet$ The hidden dimension of the hyperedge predictor, whose search space is {$128, 256$}.
>
> $\bullet$ The learning rate for node classifier, whose search space is {$1\times10^{-4}, 2\times10^{-4}, 3\times10^{-4}, 4\times10^{-4}, 5\times10^{-4}$}.
>
> $\bullet$ The dropout rate for node classifier, whose search space is {$0.0, 0.2, 0.4, 0.6, 0.8$}.
>
> In the latest version of our submission, we have included these details in **Appendix L and referenced this appendix in Section 6.1 of the main manuscript**.

---

> ### Author Response · Authors · 2024-11-21
> **Response by Authors**
>
> **_C3. Time spent on hyperparameter search._**
>
> Thank you for this comment. Indeed the efficient search for the optimal hyperparameters is challenging within the hypergraph machine learning literature. However, with the fast training speed of TF-HNN, our hyperparameter searching process is still efficient compared with previous methods. Table 1 below shows the hyperparameter searching time used for TF-HNN and ED-HNN [1] in node classification on the Cora-CA dataset in our 8-RTX-3090-GPUs server. For TF-HNN, we run the hyperparameter combinations mentioned in **_C2_**, and for ED-HNN, we run the hyperparameter combination mentioned in Appendix F.3 of their paper. Specifically, according to the discussion and table 6 in Appendix F.3 of [1], ED-HNN does grid search for the following hyperparameters:
>
> $\bullet$ The number of layers of ED-HNN, whose search space is {$1, 2, 4, 6, 8$}.
>
> $\bullet$ The number of layers of an MLP projector, $\hat{\phi}$, whose search space is {$0, 1, 2, 4, 6, 8$}.
>
> $\bullet$ The number of layers of an MLP projector, $\hat{\rho}$, whose search space is {$0, 1, 2, 4, 6, 8$}.
>
> $\bullet$ The number of layers of an MLP projector, $\hat{\varphi}$, whose search space is {$0, 1, 2, 4, 6, 8$}.
>
> $\bullet$ The number of layers of the node classifier, whose search space is {$1, 2, 4, 6, 8$}.
>
> $\bullet$ The hidden dimension of $\hat{\phi}$, $\hat{\varphi}$, and $\hat{\rho}$, whose search space is {$96, 128, 256, 512$}.
>
> $\bullet$ The hidden dimension of the node classifier, whose search space is {$96, 128, 256, 512$}.
>
> Based on the results presented in Table 1, **our hyperparameter search process requires only about $6$% of the time required by ED-HNN**, demonstrating its efficiency. We have added this discussion to Appendix S and referenced it in Section 6.1 of the main manuscript. Furthermore, we have highlighted in **both Appendix S and Section 7** that developing a more efficient hyperparameter search method for hypergraph neural networks based on the methods like Bayesian Optimisation [2-4] is a meaningful research direction and we leave it for future works.
>
> #### Table 1: Hyperparameter search time for ED-HNN and our TF-HNN in the node classification task on Cora-CA.
>
> | | ED-HNN | TF-HNN | $t_{TF}/t_{ED}$ |
> |------------------------------------------------|--------|--------|------------------|
> | Runtime for Completing the Grid Search (hours) |48.95                                         | 2.74   | 0.06   |
> ---
>
> [1] Wang, Peihao, et al. "Equivariant Hypergraph Diffusion Neural Operators." The Eleventh International Conference on Learning Representations.
>
> [2] Wu, Jia, et al. "Hyperparameter optimization for machine learning models based on Bayesian optimization." Journal of Electronic Science and Technology 17.1 (2019): 26-40.
>
> [3] Victoria, A. Helen, and Ganesh Maragatham. "Automatic tuning of hyperparameters using Bayesian optimization." Evolving Systems 12.1 (2021): 217-223.
>
> [4] Nguyen, Vu. "Bayesian optimization for accelerating hyper-parameter tuning." 2019 IEEE second international conference on artificial intelligence and knowledge engineering (AIKE). IEEE, 2019.

---

> ### Author Response · Authors · 2024-11-24
> **Follow-Up from Authors**
>
> Dear Reviewer bexR,
>
> Thank you once again for your thoughtful comments. As the discussion period is coming to a close, we want to ensure that our rebuttal has sufficiently addressed your concerns. If you have any remaining questions or need further clarification, we would be happy to assist before the deadline.

---

> > ### Comment · Reviewer_bexR · 2024-11-26
> >
> > The search space is extremely large, and I question whether fair comparisons were made by applying hyperparameter search in the same way to other methods. Furthermore, I believe that reporting the hyperparameter search time for Cora, the smallest dataset used, is not meaningful. Considering the complexity, the hyperparameter search time for Trivago, with 172,738 nodes, would likely be excessively long. I will maintain my current rating.

---

> > > ### Author Response · Authors · 2024-12-02
> > > **Response to Reviewer bexR by Authors part 1 (Second Round)**
> > >
> > > Dear Reviewer bexR,
> > >
> > > Thank you for the thoughtful reply. Below, we respond to the points raised in your latest reply, labelled as **_C'_** to distinguish them from the comments in your original review.
> > >
> > > ---
> > >
> > > **_C'1  The Fairness of Our Comparison._**
> > >
> > > Thank you for this comment. However, we emphasize that **we have tried our best to ensure fairness in the comparisons.** We further justify the fairness of our experimental comparison as follows:
> > >
> > > -  For the baseline results, **following previous papers in the literature [1,2]**, we adopted the reported results and optimal hyperparameters from the previous papers for baselines previously evaluated in the same setting. Moreover, for baselines not previously applied in the same setting, we have utilised the search space as used for our method to ensure a fair comparison.
> > >
> > > -  **The search space sizes of existing advanced HNNs are comparable to ours.** Specifically, for node classification on the small-scale hypergraphs presented in Table 3 of our paper, we tested around **46K** hyperparameter combinations. In contrast, under the same setting, [1] tested around **86K** hyperparameter combinations based on Appendix F.3 of [1]. Moreover, while [2] does not explicitly detail its hyperparameter search space, the hyperparameters listed in Appendix D suggest that it tested around **180K** combinations.
> > >
> > > [1] Wang, Peihao, et al. "Equivariant Hypergraph Diffusion Neural Operators." The Eleventh International Conference on Learning Representations.
> > >
> > > [2] Wang, Yuxin, et al. "From hypergraph energy functions to hypergraph neural networks." International Conference on Machine Learning. PMLR, 2023.
> > >
> > > ---
> > >
> > > **_C'2 The Hyperparameter Searching Time on Cora._**
> > >
> > > Thank you for the comment. However, we emphasize that the reported hyperparameter searching time on Cora is meaningful for the following reasons:
> > >
> > > - **The dataset is representative of the literature.** To our knowledge, the Cora dataset is one of the most representative datasets, not only in hypergraph machine learning literature but also in general graph machine learning literature. Therefore, we consider the results on this dataset to be representative.
> > >
> > > - **The ED-HNN [1] is a well-acknowledged strong model in the literature.** To our knowledge, ED-HNN is a very strong baseline that has been acknowledged by many papers. Therefore, we believe that the fact our model can achieve better results on Cora with much less hyperparameter search time than ED-HNN supports the value of our method.
> > >
> > > -  **The insight provided by the results on Cora is transferable to other datasets.** The reduced hyperparameter search time of our method is attributed to its faster training speed compared to ED-HNN and the smaller number of hyperparameter combinations we tested. As a result, **our superior efficiency relative to ED-HNN is transferable to other datasets.** The discussions regarding the hyperparameter search time on the large-scale Trivago dataset are provided in our subsequent responses to **_C'3_**. **Aligned with the insight from the results on Cora, the results on Trivago further highlight that our model requires significantly less hyperparameter search time compared to ED-HNN.**
> > >
> > > [1] Wang, Peihao, et al. "Equivariant Hypergraph Diffusion Neural Operators." The Eleventh International Conference on Learning Representations.

---

> > > ### Author Response · Authors · 2024-12-02
> > > **Response to Reviewer bexR by Authors part 2 (Second Round)**
> > >
> > > **_C'3 The Hyperparameter Searching on the Large-Scale Dataset Trivago._**
> > >
> > > Thank you for the comment. However, we clarify that **on the large-scale dataset Trivago, our method’s hyperparameter searching is efficient. The hyperparameter search time required by our approach is significantly lower compared to the baselines.** A detailed discussion is provided below.
> > >
> > > The experiments on Trivago are highly time-consuming and need large GPU memory. Thus, we used a relatively small search space for hyperparameters in our method and the baselines, which are provided below.
> > >
> > > In node classification:
> > > - Ours:
> > >    - The number of layers of TF-MP-Module: {$1, 4, 16, 64$};
> > >    - The $\alpha$ of TF-MP-Module: {$0.0, 0.1, 0.3, 0.5$}.
> > >    - The number of layers of the classifier: {$3$}.
> > >    - The hidden dimension of the classifier: {$64$}.
> > >    - The learning rate for the classifier: {$1\times10^{-3}, 1\times10^{-4}$}.
> > >    - The dropout rate for the classifier: {$0.0, 0.2, 0.4$}.
> > > - ED-HNN:
> > >     - The number of layers of ED-HNN: {$1, 2, 4$} (larger values show the out-of-memory (OOM) issue on our server).
> > >     - The number of layers of $\hat{\phi}$, an MLP within message passing: {$0, 1, 2$} (larger values show the OOM issue on our server).
> > >     - The number of layers of $\hat{\rho}$, an MLP within message passing: {$0, 1, 2$} (larger values show the OOM issue on our server).
> > >     - The number of layers of $\hat{\varphi}$, an MLP within message passing: {$0, 1, 2$} (larger values show the OOM issue on our server).
> > >     - The number of layers of the classifier: {$3$}.
> > >     - The hidden dimension of $\hat{\phi}$, $\hat{\varphi}$, and $\hat{\rho}$: {$64, 128$} (larger values show the OOM issue on our server).
> > >     - The hidden dimension of the classifier: {$64$}.
> > >     - The learning rate for the model: {$1\times10^{-3}, 1\times10^{-4}$};
> > >     - The dropout rate for the model: {$0.0, 0.2, 0.4$}.
> > > - Deep-HGNN:
> > >     - The number of message passing layers: {$1, 4, 16, 64$}.
> > >     - The $\alpha$: {$0.0, 0.1, 0.3, 0.5$}.
> > >     - The $\lambda$: {$0.1$}.
> > >     - The message passing hidden dimension: {$64, 128, 256$} (larger values show the OOM issue on our server).
> > >     - The number of layers of the classifier: {$3$}.
> > >     - The hidden dimension of the classifier: {$64$}.
> > >     - The learning rate for the model: {$1\times10^{-3}, 1\times10^{-4}$}.
> > >     - The dropout rate for the model: {$0.0, 0.2, 0.4$}.
> > >
> > > In hyperlink prediction:
> > >  - Ours:
> > >     - The number of layers of TF-MP-Module: {$1, 2, 4$}.
> > >     - The $\alpha$: {$0.0, 0.1, 0.3, 0.5$}.
> > >     - The number of layers of the predictor: {$3$}.
> > >     - The hidden dimension of the predictor: {$64$}.
> > >     - The learning rate for the predictor: {$1\times10^{-3}, 1\times10^{-4}$}.
> > >     - The dropout rate for the predictor: {$0.0, 0.2$}.
> > >  - ED-HNN:
> > >     - The number of layers of ED-HNN: {$1, 2, 4$}.
> > >     - The number of layers of $\hat{\phi}$, an MLP within message passing: {$0,1,2$}.
> > >     - The number of layers of $\hat{\rho}$, an MLP within message passing: {$0,1,2$}.
> > >     - The number of layers of $\hat{\varphi}$, an MLP within message passing: {$0,1,2$}.
> > >     - The number of layers of the predictor: {$3$}.
> > >     - The hidden dimension of $\hat{\phi}$, $\hat{\varphi}$, and $\hat{\rho}$: {$64,128$}.
> > >     - The hidden dimension of the predictor: {$64$}.
> > >     - The learning rate for the model: {$1\times10^{-3},1\times10^{-4}$}.
> > >     - The dropout rate for the model: {$0.0,0.2$}.
> > >  - Deep-HGNN:
> > >     - The number of message passing layers: {$1,2,4$}.
> > >     - The $\alpha$ in hyperlink predictor: {$0.0,0.1,0.3,0.5$}.
> > >     - The $\lambda$: {$0.1$}.
> > >     - The message passing hidden dimension: {$64,128,256$}
> > >     - The number of layers of the predictor: {$3$}.
> > >     - The hidden dimension of the predictor: {$64$}.
> > >     - The learning rate for the model: {$1\times10^{-3},1\times10^{-4}$};
> > >     - The dropout rate for the model: {$0.0,0.2$}.
> > >
> > > Due to time constraints, we conducted experiments on node classification to measure the time required to traverse the above hyperparameter search space for our method and the baselines, using the Trivago dataset on our 8-RTX-3090-GPUs server. The results are in Table 1 below.  **Based on these results, the hyperparameter search time for our method is only 1.7% of that required by Deep-HGNN and 3.4% of that required by ED-HNN.**  These results further support the superior efficiency of our method.
> > >
> > > #### Table 1: Hyperparameter search time in node classification on Trivago. (The search time of ED-HNN and TF-HNN on Trivago is shorter than the one on Cora, as shown in the response to your **_C2_**, as the search space used on Trivago is much smaller than the one used on Cora.)
> > > | | Deep-HGNN|   ED-HNN	|  TF-HNN |
> > > |-|-|-|-|
> > > | Hyperparameter Searching  Time (hours)|   59.61	| 30.42   |   1.04  |
> > >
> > > ---
> > >
> > > **We sincerely thank the reviewer once again. We hope that our additional clarifications address the concerns raised, leading to a more positive evaluation of our work and potentially a higher rating.**

---

> > > > ### Comment · Reviewer_bexR · 2024-12-02
> > > >
> > > > Thank you for your response. However, I still have concerns regarding the hyperparameters. Could you clarify why the hyperparameter settings for the Trivago dataset are not included in Table 13? Additionally, while the authors mentioned in their response that a smaller search space was used for the Trivago dataset, this detail does not appear to be specified in the paper. Furthermore, in line 424, it is stated that the experiments were conducted on a single RTX 3090 GPU. Could you confirm whether the experiments were actually conducted using 8 RTX 3090 GPUs instead?

---

> ### Author Response · Authors · 2024-12-02
>
> Dear Reviewer bexR,
>
> Thank you for raising these concerns. We appreciate your feedback and will address them in detail:
>
> - **Hyperparameter Settings for Trivago Dataset:** The omission of hyperparameter settings for the Trivago dataset in Table 13 was due to the fact that we completed the writing close to the submission deadline, leaving insufficient time to carefully check the appendix. Additionally, the experiment for the Trivago dataset is very time-consuming and computationally expensive, which led us to use a relatively smaller search space. We will ensure that the final version of the paper includes all hyperparameter details for the Trivago dataset.
>
> - **Use of GPUs for Experiments:** While we utilized a server with 8 RTX 3090 GPUs for efficient hyperparameter searching, all training time comparisons reported in the paper were conducted on the same single RTX 3090 GPU to ensure fair evaluation across models. We are sorry for any confusion caused by the phrasing in line 424 and will polish this section in the final version.
>
> We sincerely thank you once again for your insightful comments throughout the review and rebuttal process. The additional results and clarifications provided during the rebuttal will be incorporated into the final version of our paper. We hope that our discussion has effectively highlighted the strengths and contributions of our work, encouraging a more positive reevaluation when determining your final score/recommendation.

---

### Official Review · Reviewer_GqRw · 2024-11-04

**Soundness:** 4
**Presentation:** 4
**Contribution:** 4
**Rating:** 10
**Confidence:** 4

**Summary:**

The paper proposes a novel approach called TF-HNN (Training-Free Hypergraph Neural Network) to address the high computational complexity during training in existing hypergraph neural networks (HNNs). The key innovation is a training-free message passing module (TF-MP-Module) that decouples the processing of hypergraph structural information from the model learning stage. The authors first derive a theoretical framework that provides a unified view of existing HNN approaches, identifying the feature aggregation function as the core component processing hypergraph structure. Based on this insight, they remove the learnable parameters and non-linear activations from the feature aggregation functions of four state-of-the-art HNNs to make them training-free. Further, they consolidate the feature aggregation across layers into a single propagation step, resulting in the proposed TF-MP-Module. Extensive experiments on seven real-world datasets for the tasks of node classification and hyper-link prediction demonstrate the competitive performance of TF-HNN, with very less training time. TF-HNN is the first approach to shift the processing of structure to pre-processing stage, which significantly enhances training efficiency.

**Strengths:**

1. The paper makes a significant contribution by addressing the issue of high computational complexity of Hypergraph learning algorithms.
2. The proposed solution, TF-HNN is novel and elegant, which decouples the processing of structural information from the model training stage.
3. Authors provide a strong theoretical foundation for TF-HNN, the unified framework presented in the paper links all the popular HNN approaches, which shows that TF-HNN is designed by keeping many existing methodologies in mind, and hence provides a comprehensive mechanism for efficient training.
4. Extensive experiments on diverse real-world datasets for node classification and hyperedge prediction tasks demonstrate the competitive performance of TF-HNN against state-of-the-art HNN baselines while requiring significantly less training time.
5. The paper is well-written, with a clear motivation, rigorous theoretical analysis, and thorough empirical evaluation supporting the proposed method's effectiveness and efficiency.

**Weaknesses:**

I do not see any weak points in this paper. This is a very well written paper, with significant contributions. Please refer to the questions sections for the questions I have.

**Questions:**

1. An assumption is made about the structure of hypergraph i.e., absence of isolated nodes or empty hyperedges, for the theoretical results. What happens if isolated nodes or empty hyperedges are present? I am not able to see why this assumption is required, and what breaks if it is violated?
2. It is commendable that the proposed TF-HNN performs significantly better than the baselines, but it is also a bit strange to see the baselines performing so poor, particularly on trivago. I understand the boost in training time, but not able to fully understand why there is a 10% improvement, it seems to me  that the learning ability of any SOTA HNN should be similar to TF-HNN. I may have missed something, but curios to hear what the authors have to say on this.

---

> ### Author Response · Authors · 2024-11-21
> **Response by Authors**
>
> Thank you so much for your thoughtful comments. We have provided detailed, point-by-point responses to your comments below. Additionally, we remain open to further discussions that could help us improve the quality of our paper.
>
> ---
>
> **_C1. The assumption about the structure of hypergraph._**
>
> Thank you for your comment. In Section 2, we assume that the hypergraph does not contain isolated nodes or empty hyperedges **to maintain consistency with prior works [1–4] in the literature.** This assumption is based on the observation that most existing hypergraph neural networks (HNNs) use node degree or hyperedge degree as denominators during forward propagation. This inherently presumes these values are nonzero—i.e., there are no isolated nodes or empty hyperedges—since division by zero is undefined. Moreover, we emphasize that this assumption does not compromise the practical applicability of our method. Consistent with prior works, we address scenarios where zero degrees occur by assigning a default value of 1 to such cases. In the latest version of our submission, **we have clarified it in Section Q of the appendix, and we have also referenced this section in Section 2 of the main manuscript**.
>
> [1] Wang, Peihao, et al. "Equivariant Hypergraph Diffusion Neural Operators." The Eleventh International Conference on Learning Representations.
>
> [2] Huang, Jing, and Jie Yang. "UniGNN: a Unified Framework for Graph and Hypergraph Neural Networks." Proceedings of the Thirtieth International Joint Conference on Artificial Intelligence. International Joint Conferences on Artificial Intelligence Organization, 2021.
>
> [3] Chen, Guanzi, et al. "Preventing over-smoothing for hypergraph neural networks." arXiv preprint arXiv:2203.17159 (2022).
>
> [4] Feng, Yifan, et al. "Hypergraph neural networks." Proceedings of the AAAI conference on artificial intelligence. Vol. 33. No. 01. 2019.
>
> ---
> **_C2. The superior performance, especially on the large-scale hypergraph._**
>
> Thank you for the comment. We attribute the superior performance of TF-HNN primarily to its more efficient utilization of node information from the training data compared to baseline models.
>
> Baseline models with training-required message-passing modules process node information by first using a trainable module to project node features into a latent space, and then performing trainable message passing within that space. Learning a latent space that preserves the unique characteristics of individual nodes is extremely challenging, especially for large-scale hypergraphs. This difficulty is compounded by computational constraints: both the projection and message-passing operations require computing gradients for backpropagation, which are extremely resource-intensive. To prevent out-of-memory (OOM) errors during experiments on the large-scale hypergraph Trivago, the hidden dimensions of these baseline models need to be limited to a maximum of 256. This dimension constraint, introduced by GPU memory limitations, **makes it difficult for the model to fully preserve information that is helpful for classifying a large number of nodes.**
>
> In contrast, TF-HNN performs message passing directly on the original node features during the data pre-processing stage without requiring any trainable parameters. This allows TF-HNN to preserve node-specific information without incurring the expensive memory requirements associated with training-required models. Notably, our experimental results align with previous observations in the Graph Neural Network (GNN) literature. For example, training-free models like SIGN [1] have been shown to outperform training-required models like GCN [2] on large-scale graphs such as the Protein-Protein Interaction (PPI) network by a significant margin.
>
> To enhance the understanding of our method, we have included this discussion **in Appendix Section R and referenced it in Section 6.2** of the main manuscript to ensure clarity and accessibility.
>
> [1] Frasca, Fabrizio, et al. "Sign: Scalable inception graph neural networks." arXiv preprint arXiv:2004.11198 (2020).
>
> [2] Kipf, Thomas N., and Max Welling. "Semi-supervised classification with graph convolutional networks." arXiv preprint arXiv:1609.02907 (2016).

---

> > ### Comment · Reviewer_GqRw · 2024-11-25
> > **Thank you for the response**
> >
> > I thank the authors for their responses.
> > I read through all the other reviews, and the responses from authors. I still feel that this paper has significant contribution to Hypergraph Learning literature, and I stand by my score.

---

> > > ### Author Response · Authors · 2024-11-28
> > > **Thank you again for your valuable feedback**
> > >
> > > Dear Reviewer GqRw,
> > >
> > > Thank you so much for your kind comment and strong support of our work! We deeply appreciate your reviews and feedback on both the theoretical and empirical aspects of our work.

---

### Official Review · Reviewer_uQFu · 2024-11-06

**Soundness:** 2
**Presentation:** 3
**Contribution:** 1
**Rating:** 3
**Confidence:** 4

**Summary:**

The paper proposes TF-HNN, a training-free hypergraph neural network that removes the need for computationally intensive message passing during training. By shifting hypergraph structural processing to the preprocessing stage, TF-HNN reduces computational complexity. The model achieves efficient, robust node feature generation without oversmoothing, utilizing as much structural information as traditional HNNs. Experiments show that TF-HNN outperforms state-of-the-art HNNs in accuracy and training speed, especially on large-scale benchmarks.

**Strengths:**

1. The paper is well-structured and easy to follow.
2. The summary of hypergraph neural networks is comprehensive, particularly with the insights provided in Table 1 and the related analysis.

**Weaknesses:**

Major weaknesses:
1  Proposition 4.2 shows the similarity between the proposed method and APPNP [1], yet the paper does not cite APPNP. Specifically, Eq (6) and Eq (14) appear to apply APPNP after performing clique expansion on the hypergraph. This connection should be discussed more thoroughly, referencing APPNP's Eq (7) and Eq (8) to clarify the relationship and implications of this similarity in the context of hypergraph learning. While the experimental results demonstrate superiority, this oversight is a significant limitation on the paper's originality.

2  Equations (4a) to (4d) result from removing learnable parameters from several baselines, corresponding to the different scenarios in the authors' proposed framework. While these modifications reasonably reduce training time, there is insufficient ablation study to explain the performance improvements. The authors' comparison of the impact from the weighted $S$ of TF-HNN in node classification is commendable. However, they should also present the performance of these modified baselines to clearly demonstrate the impact of removing the learnable parameters.

Minor weakness:

Chien et al.'s work [2] provides the challenging YELP dataset, where many baseline methods yield unsatisfactory performance. Including results on this dataset would enhance the paper's quality and offer a more comprehensive evaluation of the proposed method.
[1] You are allset: A multiset function framework for hypergraph neural networks. Chien  et al. ICLR 2022

[2] Predict then Propagate: Graph Neural Networks meet Personalized PageRank
Gasteiger et al. ICLR 2019

**Questions:**

See weaknesses.

---

> ### Author Response · Authors · 2024-11-21
> **Response by Authors**
>
> Thank you so much for your thoughtful comments. Please find our point-by-point responses below. If our responses address your concerns, we would greatly appreciate it if you could consider raising your score. Additionally, we remain open to further discussions that could help us improve the quality of our paper.
>
> ---
> **_C1. Comparison with APPNP._**
>
> Thank you for bringing the interesting paper [1] to our attention and we are sorry for missing it in the original version of our submission. In the latest version of our submission, we have appropriately cited and discussed APPNP in **Sections 1 and 3.1** of the main manuscript.
>
> Indeed, the APPNP proposed in [1] has contributed to the development of more efficient GNN models by shifting learnable parameters to an MLP prior to message passing. However, we aim to clarify the differences between the APPNP method proposed in [1] and our TF-HNN in two different perspectives.
>
> $\bullet$ **Methodological Design:** The key difference in the model design is that APPNP requires running the message passing process **during model training**, whereas our TF-HNN performs message passing only **during data preprocessing**. According to Eq. (4) in [1], APPNP employs a message-passing block after a multi-layer perceptron (MLP). The MLP uses the given original node features to generate latent features for the nodes, which then serve as inputs for message passing. This design **requires the message-passing block to be executed during each forward propagation in the training phase**, making it incompatible with preprocessing. In contrast, the training-free message-passing module in TF-HNN directly takes the given original node features as inputs, **enabling it to be fully precomputed during the data preprocessing stage**, thereby eliminating the need for computation during training, which further enhances the training efficiency. Notably, Table 2 below demonstrates that TF-HNN is more training efficient than an APPNP with clique expansion. As a result, we argue the APPNP in [1] does not diminish the core novelty of our TF-HNN, which is the first model to decouple the message-passing operation from the training process for hypergraphs.
>
> $\bullet$ **Practical differences:** In Tables 1 and 2 below, we empirically demonstrate that, for hypergraph-structured data, TF-HNN is both more training efficient and more effective than APPNP with the clique expansion **defined in our Eq. (5)**. For training efficiency, since APPNP requires message-passing computations during each forward propagation step in training, its training time is **at least 3 times longer than that of TF-HNN.** For effectiveness, our results show that TF-HNN, which performs message passing in the original feature space, achieves better performance compared to APPNP, which projects features into a latent space by an MLP before message passing. We hypothesize that this improvement may be attributed to the MLP in APPNP which may potentially not be able to fully retain information from the original node features during the projection to the latent space.
>
> Table 1: The node classification accuracy (\%) for TF-HNN and APPNP+CE. The best result on each dataset is in **bold font**.
>
> | Methods / Datasets | Cora-CA             | Citeseer            |
> |---------------------|---------------------|---------------------|
> | TF-HNN             | **86.54 ± 1.32**   | **74.82 ± 1.67**   |
> | APPNP+CE           | 86.01 ± 1.35      | 74.32 ± 1.57       |
>
> ---
>
> Table 2: The training time (s) for TF-HNN and APPNP+CE corresponding to results in Table 1. The best result on each dataset is in **bold font**.
>
> | Methods / Datasets | Cora-CA             | Citeseer            |
> |---------------------|---------------------|---------------------|
> | TF-HNN             | **0.22 ± 0.12**    | **1.12 ± 0.30**    |
> | APPNP+CE           | 0.73 ± 0.16       | 4.61 ± 0.88       |
>
> Besides the two key differences between APPNP and our TF-HNN above, we also hope to emphasize **the unique theoretical contribution** of our paper for hypergraph machine learning. While the mechanism of graph neural networks (GNNs) is well-studied, the foundational component of HNNs is still underexplored. In Section 3.2, we identify the feature aggregation function as the core component of HNNs, and we show that existing HNNs primarily enhance node features by aggregating features from neighbouring nodes. This insight encompasses various HNN models and provides researchers with a deeper understanding of the behaviour of existing HNNs.
>
> As a result, while APPNP represents a significant contribution to the field, it does not diminish the unique value and contributions of TF-HNN for hypergraphs. The detailed discussion provided in this response has been incorporated into **Section N of the appendix**.
>
> [1] Gasteiger, Johannes, Aleksandar Bojchevski, and Stephan Günnemann. "Predict then Propagate: Graph Neural Networks meet Personalized PageRank." ICLR 2018.

---

> ### Author Response · Authors · 2024-11-21
> **Response by Authors**
>
> **_C2. Remove only the learnable parameters from the node feature generation process of baselines._**
>
> Thank you for this comment. In Table 3 below, we present experiments that remove only the learnable parameters from the node feature generation process in the baselines. In this setup, all the baselines studied in Eq.(4a) to Eq.(4d) (AllDeepSets, UniGNN, ED-HNN, Deep-HGNN) directly apply non-learnable message passing to the original node features. These results confirm that **removing the learnable parameters and directly applying message passing on the original given node features can enhance the model performance.** These results, together with the ablation study on the weight design for $S$ presented in Table 7 of our submission, demonstrate that both the removal of learnable parameters and the design of our $S$, contribute to the effectiveness of our TF-HNN. In the latest version of our submission, we have included this discussion into **Section O of the appendix** and referenced this section in Section 6.2 of the main manuscript. We believe this discussion can improve our manuscript by helping readers to better understand the effectiveness of our TF-HNN.
>
>
> Table 3: Results for baselines with/without learnable parameters in node classification. The best result on each dataset is highlighted in **bold font**. Here $+LP$/$-LP$ means the model with/without learnable parameters.
>
> | Dataset   | AllDeepSets $+LP$       | AllDeepSets $-LP$       | UniGNN $+LP$          | UniGNN $-LP$          | ED-HNN $+LP$          | ED-HNN $-LP$          | Deep-HGNN $+LP$       | Deep-HGNN $-LP$       |
> |-----------|-------------------------|-------------------------|-----------------------|-----------------------|-----------------------|-----------------------|-----------------------|-----------------------|
> | Cora-CA   | 81.97 ± 1.50           | **82.53 ± 0.70**       | 83.60 ± 1.14          | **84.82 ± 1.56**      | 83.97 ± 1.55          | **85.69 ± 0.96**      | 84.89 ± 0.88          | **86.22 ± 1.33**      |
> | Citeseer  | 70.83 ± 1.63           | **71.10 ± 2.33**       | 73.05 ± 2.21          | **73.96 ± 1.67**      | 73.70 ± 1.38          | **74.08 ± 1.51**      | 74.07 ± 1.64          | **74.64 ± 1.52**      |
>
> ---
> **_C3. Experiments on Yelp._**
>
> Thank you for highlighting the valuable Yelp dataset introduced by [1]. We have conducted experiments on this dataset, and the results are summarized in Table 4 below. For HGNN, HCHA, HNHN, UniGCNII, AllDeepSets, and AllSetTransformer, we directly use the accuracy reported in [1] and record the training time by running the models with the hyperparameters provided in [1] on our RTX 3090 GPUs. For ED-HNN and Deep-HGNN, which do not have results reported in [1], we report both accuracy and training time based on runs using the optimal hyperparameters we identified. Due to out-of-memory issues with PhenomNN on our RTX 3090 GPUs, we did not include it in the table. The results demonstrate that our model is both effective and efficient on this dataset. Specifically, our model achieves a general second-best performance among the baselines, with accuracy only 0.83\% lower than AllSetTransformer, while **being 24 times faster.** These findings further highlight the effectiveness and efficiency of our proposed TF-HNN. In the latest version of our submission, We have included this discussion in **Section P of the appendix** and referenced this section in Section 6.2 of the main manuscript. We believe this discussion can improve our manuscript by helping the readers to better understand the efficiency and effectiveness of our TF-HNN.
>
> Table 4: Results for HNNs and TF-HNN on Yelp. The best results are highlighted in **bold font**. The second-best results are in _Italics_.
>
> | Metric           | HGNN          | HCHA          | HNHN          | UniGCNII       | AllDeepSets    | AllSetTransformer   | ED-HNN         | Deep-HGNN       | TF-HNN (ours)  |
> |-------------------|---------------|---------------|---------------|----------------|----------------|---------------------|----------------|----------------|----------------|
> | Accuracy (\%)  | 33.04 ± 0.62  | 30.99 ± 0.72  | 31.65 ± 0.44  | 31.70 ± 0.52   | 30.36 ± 1.57   | **36.89 ± 0.51**    | 35.03 ± 0.52   | 35.04 ± 2.64   | _36.06 ± 0.32_ |
> | Time (s)      | 326.15 ± 4.55 | 147.54 ± 0.87 | _70.35 ± 0.52_| 108.21 ± 5.87  | 193.43 ± 2.84  | 158.19 ± 0.66       | 225.64 ± 28.52 | 889.14 ± 40.79 | **6.54 ± 0.96**|
>
> [1] Chien, Eli, et al. "You are allset: A multiset function framework for hypergraph neural networks." 10th International Conference on Learning Representations, ICLR 2022.

---

> ### Author Response · Authors · 2024-11-24
> **Follow-Up from Authors**
>
> Dear Reviewer uQFu,
>
> Thank you once again for your thoughtful comments. As the discussion period is coming to a close, we want to ensure that our rebuttal has sufficiently addressed your concerns. If you have any remaining questions or need further clarification, we would be happy to assist before the deadline.

---

> > ### Comment · Reviewer_uQFu · 2024-11-25
> >
> > Thank you for the response. However, I still think the contribution of the paper is quite incremental, which just adapts a simple and almost trivial decoupling idea that is well-known in the literature. For APPNP, it is easy to be implemented as train-free: just move the mlp to the end, and empirically this also works well. So the explanation in the response is not sufficient to justify the contribution of TF-HNN compared with APPNP.

---

> ### Author Response · Authors · 2024-11-28
> **Response to Reviewer uQFu by Authors part 1 (Second Round)**
>
> Dear Reviewer uQFu,
>
> Thank you for your response. We very much appreciate your feedback on our work, at the same time we respectfully disagree with your view that our work is trivial compared to APPNP [1]. While we acknowledge APPNP as a significant contribution to the graph learning literature, we believe it does not diminish the unique value and contributions of TF-HNN within the context of hypergraph learning. Below, we respond to the points raised in your latest reply, labelled as **_C'_** to distinguish them from the comments in your original review.
>
> [1] Gasteiger, Johannes, Aleksandar Bojchevski, and Stephan Günnemann. "Predict then Propagate: Graph Neural Networks meet Personalized PageRank." International Conference on Learning Representations. 2018.
>
> ---
>
> **_C'1 The contribution of the paper is quite incremental._**
>
> The key contribution of our work is **the first training-free message passing for hypergraphs.** To our knowledge, our work is the first to decouple message passing from training in the context of hypergraph learning, which significantly reduces computational overhead without sacrificing performance. For instance, as shown in **Table 4** of our paper, on the large-scale benchmark Trivago, TF-HNN outperforms the best baseline in node classification accuracy by 10%, with just 1% of the training time of that baseline. Since many real-world hypergraphs are large-scale, training efficiency is a critical concern. **Enhancing the training efficiency of hypergraph neural networks (HNNs) is not a marginal improvement but a necessary step to address real-world scalability.** The efficiency gain brought by our model has enhanced the practical usability of HNNs for real-world applications with large-scale hypergraphs, establishing its importance in advancing the field.
>
> ---
>
> **_C'2 This paper just adapts a simple and almost trivial decoupling idea that is well-known in the literature._**
>
> We emphasize that **employing the decoupling idea to hypergraph learning is non-trivial.** This decoupling is supported by novel and sound theoretical analysis. In **Section 3.1**, we identify the feature aggregation function as the core mechanism of HNNs through a theoretical analysis, which, to our knowledge, provides the first unified view of the key component within the message-passing module of HNNs in the literature. Building on this insight, **Section 3.2** systematically removes redundancies from existing HNN designs and decouples the message passing from training. Moreover, **Proposition 4.1** demonstrates that this decoupling preserves the capacity to generate informative node features comparable to more complex HNNs. To our knowledge, **no prior work in the hypergraph domain has provided comparable theoretical results, especially when removing learnable parameters and nonlinear functions.** We believe these theoretical insights can serve as a starting point for future advancements in the HNN design.

---

> ### Author Response · Authors · 2024-11-28
> **Response to Reviewer uQFu by Authors part 2 (Second Round)**
>
> **_C'3 This work is a trivial generalization of APPNP._**
>
> Firstly, we hope to emphasise three key differences between our work and APPNP [1].
>
> $\bullet$ **The objectives of the two works are different.** We focus on designing a more efficient method for processing hypergraphs, whereas APPNP primarily aims to capture long-range dependencies while preserving local information within a graph. Therefore, the key challenge in our paper is to simplify the formula for processing hypergraphs as much as possible, while the core challenge of APPNP lies in designing an operator that balances long-range dependencies and local information within a graph.
>
> $\bullet$ **The designs of the two works are different.** We enhance the efficiency of HNNs by systematically identifying key components and eliminating redundancies to decouple message passing from training. APPNP uses long-range dependencies while preserving local information within a graph, leveraging the relationship between the PageRank algorithm [2] and the GCN model [3]. Notably, while APPNP decouples learnable parameters within the message-passing process, it still requires performing message passing during each forward propagation in training. In contrast, our method entirely shifts the message passing from the training to the preprocessing, as elaborated below.
>
> $\bullet$ **The implementations of the two works are different.** It is crucial to note that, although APPNP decouples the neural network from information propagation on the graph, Eq. (4) of APPNP [1] shows that message passing is still required during each forward propagation in training. Therefore, **the message passing in APPNP is not fully decoupled from training and cannot be completed during the data preprocessing.** In contrast, as shown in Section 3 of our paper, our TF-HNN fully shifts the message-passing process from training to preprocessing. Consequently, APPNP with the $W$ matrix used by TF-HNN **requires at least three times more training time than our TF-HNN**, as shown in Tables 1 and 2 of our response to Reviewer uQFu’s **_C1_**.
>
> While a hypothetical modification of APPNP — removing the MLP before message passing and using the $W$ matrix used by TF-HNN — may resemble our approach, we argue that it does not diminish the contribution of our work for the following reasons:
>
> $\bullet$ **Novelty:** To our knowledge, no prior work has theoretically or empirically explored such a hypothetical modification, making it unexplored.
>
> $\bullet$ **Theoretical results:** Simply modifying APPNP fails to provide the theoretical depth and guarantees that underpin our work. To ensure the effectiveness of TF-HNN for hypergraph learning, we provide a comprehensive theoretical framework. This includes an in-depth analysis of existing HNNs (Sections 3.1, 3.2, and 3.3), a discussion of the $W$ utilized by TF-HNN (Section 3.3), and a theoretical analysis of the model behaviour (Section 4). This depth of analysis and the resulting insights into hypergraph-specific mechanisms are absent in APPNP and the hypothetical variant, solidifying the originality and impact of our contributions.
>
> $\bullet$ **Simple ≠ Trivial**: Even if we ignore all the theoretical results supporting the design of TF-HNN and purely treat it as a modified version of APPNP, we argue that this model is still not a trivial generalization. Generally, a simple modification does not necessarily mean a trivial modification. For example, GCNII [4], a significant contribution to graph machine learning, modifies GCN [3] with simple, well-known techniques (initial residuals studied by [1] and identity mapping studied by [5]). Despite its simplicity, this modification is remarkable because it enabled deep GCNs to leverage initial node features and address the oversmoothing issue effectively. Similarly, the modification that removes the MLP before the message passing within APPNP and makes it use the $W$ matrix used by TF-HNN is non-trivial as: 1) it can largely improve the model training efficiency; 2) it can improve the model effectiveness by doing message passing in the original feature space; and 3) it enables the graph-based model to process hypergraphs.
>
> Consequently, our work is not a trivial generalization of APPNP and represents a significant theoretical and practical advancement in hypergraph learning.
>
> [1] J. Gasteiger, et al. "Predict then Propagate: Graph Neural Networks meet Personalized PageRank." ICLR. 2018.
>
> [2] L. Page. The PageRank citation ranking: Bringing order to the web. Technical Report, 1999.
>
> [3] TN Kipf, et al. "Semi-Supervised Classification with Graph Convolutional Networks." ICLR. 2017.
>
> [4] M Chen, et al. "Simple and deep graph convolutional networks."ICML. 2020.
>
> [5] M Hardt, et al. "Identity Matters in Deep Learning." ICLR. 2017.
>
>   ---
>
> **Once again, we sincerely thank the reviewer and are more than happy to discuss any additional improvements that could help the reviewer evaluate our work more positively.**

---

### Author Response · Authors · 2024-11-21
**Overall Response by Authors**

We sincerely thank the reviewers for their valuable feedback and insights. We are pleased that they recognized the value of our theoretical analysis of existing hypergraph neural networks (HNNs) (uQFu, GqRw, and 9Q7r) and the unique contribution that TF-HNN brings to the hypergraph machine learning literature (GqRw, bexR, 9Q7r). We also appreciate their acknowledgement of the efficiency and effectiveness of our proposed TF-HNN (GqRw, bexR, 9Q7r), the comprehensiveness of our experiments (GqRw, 9Q7r), and the quality of our writing (uQFu, GqRw, bexR).

In response to the reviewers' comments, we have provided detailed, point-by-point responses and have revised and submitted a new version of our paper. The revised parts are highlighted in blue in the updated version. The main enhancements include:

$\bullet$ Providing a comparison between TF-HNN and the previously developed graph neural network named APPNP (see Section 1, Section 3.1 and Appendix N).

$\bullet$ Adding additional results and discussion to further analyse the superior performance of TF-HNN (see Appendix O and Appendix R, referenced in Section 6.2).

$\bullet$ Providing new results on the Yelp benchmark dataset, further demonstrating the effectiveness and efficiency of TF-HNN (see Appendix P referenced in Section 6.2).

$\bullet$ Providing more details about hyperparameter tuning, making the experiment more comprehensive (see Appendix L and Appendix S referenced in Section 6.1).

$\bullet$ Clarifying the descriptions of our propositions to enhance their accessibility for readers (see Section 2, Section 3.3, Appendix T and Appendix U).

$\bullet$ Including the runtime of our preprocessing stage, highlighting the efficiency of TF-HNN (see Appendix V referenced in Section 6.2).

$\bullet$ Correcting all typos mentioned by the reviewers.

Please see below for our point-by-point responses to the reviewers' feedback. Thank you for your time and consideration.

---

### Meta-Review · Area_Chair_31wc · 2024-12-21

**Metareview:**

The authors propose a hypergraph neural network that decouples message passing from training by precomputing message passing during preprocessing. To achieve this, the authors unify popular hypergraph neural network designs into a single theoretical framework and systematically remove learnable parameters. The experimental results demonstrate that this method significantly reduces training overhead with little to no loss in performance.

The reviewers recognized the paper's practical significance, theoretical contributions, and the comprehensiveness of its experimental evaluation. However, opinions were divided on the novelty of the work. Some reviewers pointed out notable similarities between the proposed model and decoupled GNN architectures, but the authors clearly clarified differences in their responses.

The overall reviews leaned towards acceptance, and the strengths of the work, particularly as it introduces the first decoupled approach in hypergraph neural networks, justify its acceptance.

**Additional Comments On Reviewer Discussion:**

During the rebuttal phase, the authors addressed most of the concerns raised by the reviewers. Regarding novelty, they clarified the differences between the proposed method and decoupled GNN architectures; however, the positive and negative effects (e.g., in terms of expressive power) of these differences require further exploration, which can be included in the camera-ready version.

---

### Decision · Program_Chairs · 2025-01-22

Accept (Poster)